# GNAS/PKA signaling promotes aberrant osteochondral differentiation of *Gli1*⁺ tendon sheath progenitors

Lijun Chen [1,2,3,7], Chao Peng [2,7], Lanyi Chai[2,7], Renjie Zhang[1,7], Chenghang Zhu[2], Hailin Wang[2], Qirong Cheng[2], Yan Yan [4], Cailiang Shen[1], Hong Zheng [2], Jiazhao Yang [5,8✉], Haitao Fan [6,8✉] & Chen Kan [1,2,4,8✉]

## Abstract

Tendon injury promotes aberrant osteochondral differentiation of tendon stem cells (TSCs) and results in disability. However, the cellular subsets within the osteochondral lineage involved in this process and associated mechanisms remain unclear. Here, we found that, following Achilles tenotomy, murine *Gli1*⁺ tendon sheath cells expanded rapidly, transitioning into tenogenic and osteochondrogenic cells. Lineage tracing, together with single-cell RNA sequencing, revealed that osteochondrogenic *Gli1*⁺ tendon sheath cells originate from Scx⁺ tendon stem/progenitor cells, preferentially differentiate into osteochondral lineage tendon progenitors at 7 dpi, subsequently undergoing aberrant chondrogenesis and osteogenesis at 21dpi and 63dpi, respectively. In addition, *Acvr1*^{R206H/+} robustly accelerates osteochondral differentiation in *Gli1*⁺ tendon sheath progenitors. Furthermore, GNAS/PKA signaling was significantly activated in osteochondral differentiation of *Gli1*⁺ tendon sheath progenitors. Alternatively, treatment with the Gₛₐ antagonist, NF449, or genetic inhibition of the PKA subunit, *Prkaca*, in *Gli1*⁺ sheath progenitors significantly alleviated aberrant osteochondral differentiation. NF449 also prevented osteochondral differentiation of human tendon stem cells. These findings identify *Gli1*⁺ tendon sheath progenitors with osteochondral differentiation capacity during heterotopic ossification via activation of GNAS/PKA signaling, suggesting PKA as a potentially effective therapeutic target to treat tendon ossification.

**Keywords** Heterotopic Ossification; Tendon Sheath Progenitors; Glioma-associated Oncogene Homolog 1 (*Gli1*); Guanine Nucleotide-Binding Protein (s) Alpha (*Gnas*); Protein Kinase A (PKA)
**Subject Categories** Development; Signal Transduction; Stem Cells & Regenerative Medicine

## Introduction

Mesenchymal stem cells (MSCs), also known as multipotent stromal cells, can differentiate into various connective cell lineages and participate in tissue development, maintenance, and regeneration (Soliman et al, 2021). However, aberrant osteochondral differentiation of MSCs in soft tissues leads to incorrect tissue regeneration into heterotopic ossification (HO), a devastating disease that limits the range of motion in joints and throughout the body, as well as neuropathic pain (Juan et al, 2024). Tissue-specific MSCs contribute to HO development in diverse tissues (Dey et al, 2016b). Tendon stem cells preferentially give rise to osteochondrogenic lineages in injured tendon, which subsequently form tendon HO (Dey et al, 2016a; Feng et al, 2020). In fibrodysplasia ossificans progressiva (FOP), a genetic HO disorder, a mutation in Activin A receptor type 1 (*Acvr1*) promotes aberrant osteochondral differentiation in muscle resident MSCs, also called fibro-adipogenic progenitors (FAP) (Lees-Shepard et al, 2022). Ligament stem cells have been considered the cellular source for ossification of posterior longitudinal ligaments (OPLL) (Wang et al, 2024). However, the diversity of tissue-specific MSCs, and their developmental trajectories during HO formation, have not been characterized.

The hedgehog signaling pathway is a highly regulated, coordinated signal cascade involving extracellular ligands, receptor proteins, cytoplasmic signaling molecules, transcription factors, co-regulators, and target genes. Activation of this signaling cascade is required for tissue development and homeostasis, and also plays a major role in maintaining pluripotent and somatic stem cell populations (Yunxiao & Philip A, 2023). One hedgehog pathway transcription factor in particular, *Gli1*, is a signature gene for several tissue-specific stem cells, including tendon enthesis progenitors (Fang et al, 2022), incisor stem cells (Tingwei et al, 2024), perivascular MSCs (Kramann et al, 2015), FAPs (Yao et al, 2021), skeletal stem cells (Jeffery et al, 2022), hepatocyte stem cells

[1]Department of Orthopedics, The First Affiliated Hospital of Anhui Medical University, 218 Jixi Road, 230032 Hefei, China. [2]Department of Pathophysiology, School of Basic Medical Sciences, Anhui Medical University, 81 Meishan Road, 230022 Hefei, China. [3]Department of Pathology, the Second People's Hospital of Hefei, Hefei Hospital Affiliated to Anhui Medical University, Guangde Street, 230011 Hefei, China. [4]Laboratory Animal Research Center, School of Basic Medical Sciences, Anhui Medical University, He Fei, China. [5]Department of Orthopedics, The First Affiliated Hospital of USTC, 17 Lujiang Road, 230001 Hefei, China. [6]Department of Orthopedics, The First Affiliated Hospital of Ningbo University, 59 Liuting Street, 315010 Ningbo, China. [7]These authors contributed equally as first authors: Lijun Chen, Chao Peng, Lanyi Chai, Renjie Zhang. [8]These authors contributed equally as senior authors: Jiazhao Yang, Haitao Fan, Chen Kan.✉E-mail: yangjiazhao@ustc.edu.cn; fyyfanhaitao@nbu.edu.cn; chenkan@ahmu.edu.cn

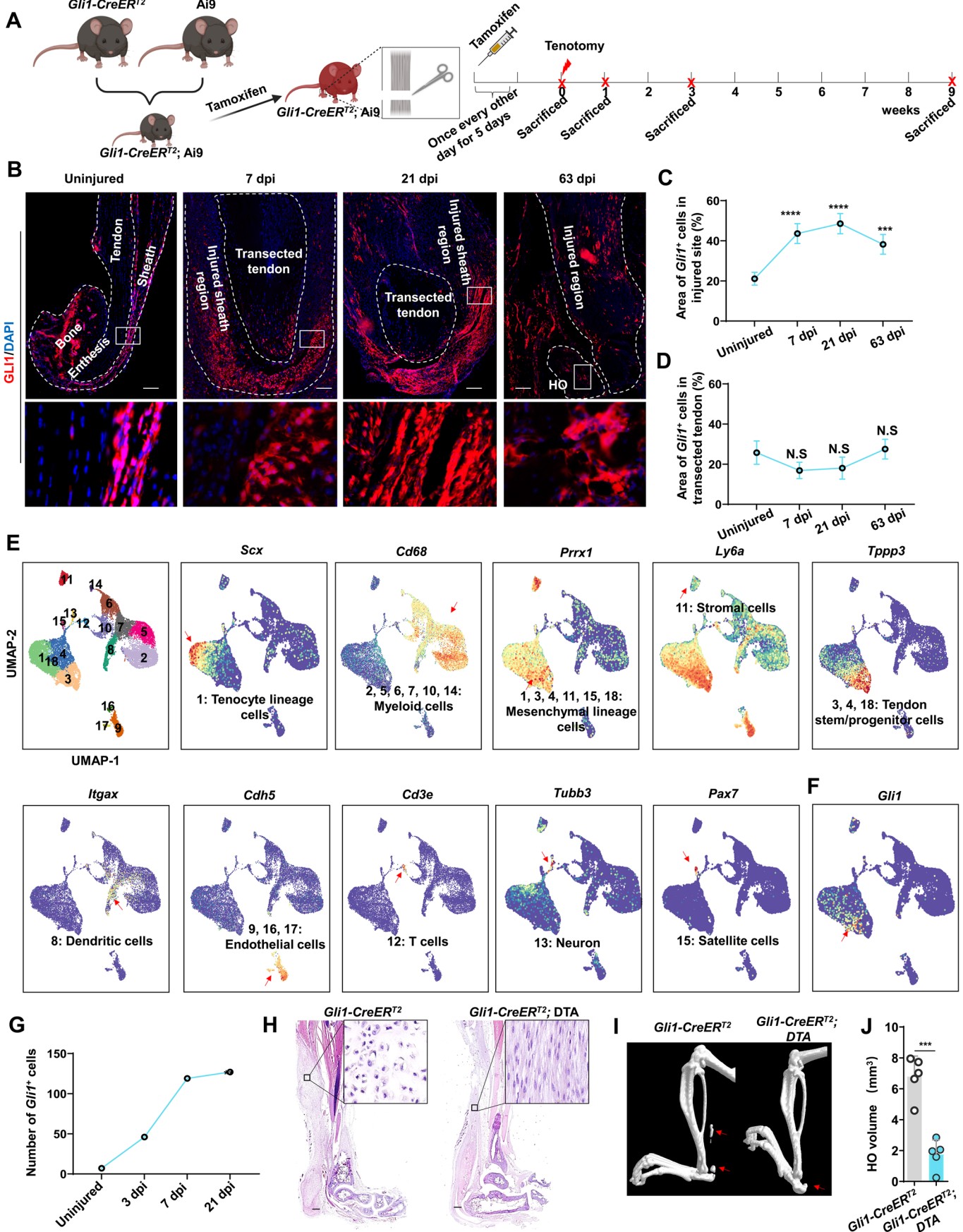

**Figure 1.** *Gli1*⁺ tendon sheath progenitors contribute to traumatic HO formation.

(A) Schematic diagram depicting the expression pattern of *Gli1*⁺ tendon sheath cells in injured Achilles tendon during HO formation using genetic lineage tracing in vivo (*Gli1-CreER$^{T2}$*; Ai9 mice). (B, C) Representative immunofluorescence images (B) and statistical analysis (C) of *Gli1*⁺ cells (genetically traced by *Gli1-CreER$^{T2}$*;Ai9 system) in uninjured and injured Achilles tendon sites at 7, 21, and 63 dpi ($n = 5$ per group). 7 dpi vs uninjured ****$P = 2.63e-5$, 21 dpi vs uninjured ****$P = 6.84e-6$, 63 dpi vs uninjured ***$P = 1.76e-4$. Scale bar, 200 μm. (D) Statistical analysis of *Gli1*⁺ cells in uninjured tendon and transected tendon ($n = 5$ per group). 7 dpi *vs* uninjured $P = 9.59e-2$, 21 dpi *vs* uninjured $P = 1.71e-1$, 63 dpi *vs* uninjured $P = 7.08e-1$. N.S. indicated no significance. (E) Feature plot of indicated marker gene expression of each cell type in uninjured and injured site of Achilles tendon at different time point post injury using scRNA-seq. (F, G) Feature plot of *Gli1* expression (F) and the number (G) of *Gli1*⁺ cells in uninjured and injured site of Achilles tendon at different time point post injury using scRNA-seq. (H) Representative HE images of injured Achilles tendon in *Gli1-CreER$^{T2}$* and *Gli1-CreER$^{T2}$*; DTA at 21 dpi. Scale bar, 200 μm. (I, J) Representative microCT (I) and statistical analysis (J) of HO volume in *Gli1-CreER$^{T2}$* and *Gli1-CreER$^{T2}$*; DTA at 63 dpi ($n = 5$ per group). ***$P = 1.28e-4$. Data are presented as mean ± SD. All *P* values were determined by unpaired Student's *t* test.

(Peng et al, 2022), sentinel muscle stem cells (Peng et al, 2023), neural stem cells (Luo et al, 2023) and other niche supportive MSCs involved in tissue fibrosis or cancer progression (Braeuer et al, 2021; Sobecki et al, 2022). We previously reported that *Gli1*⁺ cells are essential for osteochondral differentiation during BMP4-dependent HO formation in muscle (Kan et al, 2018a; Kan et al, 2019). However, the identity of *Gli1*⁺ cells in Achilles tendon, and their contribution to tendon HO, remains unclear.

Several signaling pathways reportedly participate in regulating aberrant osteochondral differentiation of MSCs during HO formation (Kan et al, 2018b). Gain-of-function mutations in *Acvr1* can activate the phosphorylation of mothers against decapentaplegic homolog 1/5/8 (Smad 1/5/8), subsequently promoting the development of HO (Kaplan *et al*, 2024). Transforming growth factor beta (TGF-β) signaling is also essential for tendon HO formation (Wang et al, 2018). However, targeting BMP/SMAD or TGF-β signaling confers relatively limited therapeutic effects in HO treatment (Aykul et al, 2022), suggesting that other signaling pathways might also regulate HO formation. To explore such pathways that may serve as effective therapeutic targets for HO treatment, it is necessary to investigate the dynamic changes in signaling that occur during the elaborate process of fate determination in tissue-specific MSCs.

Guanine nucleotide-binding protein G(s) subunit alpha (*GNAS*) isoforms function as the stimulatory subunit of G protein involved in activating adenylyl cyclase (AC), which increases levels of the ubiquitous signal molecule, cyclic adenosine monophosphate (cAMP), and subsequent activation of protein kinase A (PKA) (Coles et al, 2020). PKA can activate several substrates, such as the cyclic AMP response element-binding protein (CREB), which regulate cell proliferation, morphology, autophagy, and cell death. Deficiency for *GNAS* has been shown to promote heterotopic intramembranous ossification in progressive osseous heteroplasia (POH) (Shore et al, 2002), although little is known about its role in heterotopic endochondral ossification, such as tendon HO.

In this study, lineage tracing and single-cell RNA sequencing (scRNA-seq) identified *Gli1*⁺ tendon sheath progenitors as a subpopulation of tendon stem/progenitor cells that serve as a cellular source for tendon HO formation in both mice and humans. *Gli1*⁺ tendon sheath progenitors initially differentiate into tendon lineage progenitors that show enhanced capacity for osteochondral differentiation, terminating as chondrocytes and osteocytes. Moreover, our results show that GNAS/PKA signaling is essential for generating these tendon lineage progenitors, while suppressing this signaling pathway can effectively inhibit HO formation, thus providing a therapeutic target for strategies aimed at alleviating tendon HO formation.

# Results

## *Gli1*⁺ tendon sheath progenitors expand and contribute to HO after Achilles tenotomy

To better understand the contribution of *Gli1*-expressing (*Gli1*⁺) tendon cells to the development of HO, we established a *Gli1*-labeled (*Gli1-CreER$^{T2}$*; Ai9) mouse line by crossing *Gli1-CreER$^{T2}$* mice with Ai9 reporter mice. Tenotomy surgery was then performed in these mice to observe the distribution of *Gli1*⁺ cells during HO formation (Fig. 1A). After inducing HO in the Achilles tendon, samples were collected at 0-, 7-, 21-, and 63-days post injury (dpi), respectively corresponding to the uninjured, fibroproliferative, chondrogenic, and osteogenic stages of HO (Fig. 1B; Appendix Fig. S1A). In uninjured tendon, we found that *Gli1*⁺ cells were mainly distributed in tendon sheath, with relatively few such cells in tendon enthesis (Fig. 1B). At 7 dpi following tenotomy, *Gli1*⁺ cells were expanded at the injury site in the tendon sheath region, but not the transection region of the Achilles tenotomy (Fig. 1B,C). Moreover, the abundance of *Gli1*⁺ cells was significantly increased in the injury site at 21 and 63 dpi, compared to the uninjured group (Fig. 1B,C). However, the *Gli1*⁺ cells in the transected tendon were not significantly increased following tenotomy injury, compared to the uninjured tendon enthesis region (Fig. 1B,D). These results suggested the possible involvement of *Gli1*⁺ tendon sheath cells in HO formation following tendon injury.

To further confirm that *Gli1*⁺ cells contribute to HO in tendon, we quantified *Gli1*⁺ cells during HO formation in a previously published scRNA-seq dataset spanning four time points (i.e., uninjured, 3, 7, and 21 dpi) in Achilles tendon HO development (Sorkin et al, 2020). Analysis of various cell type-specific markers from the Immgen database, together with previously published markers, identified 18 clusters spanning nine distinct cell types, including tenocytes (cluster 1, *Scx*^high/*Tppp3*⁻/*Ly6a*^low), myeloid cells (clusters 2, 5, 6, 7, 10, and 14, *Cd68*⁺), tendon stem/progenitor cells (clusters 3, 4, and 18, *Tppp3*^high/*Prrx1*⁺/*Ly6a*⁺/*Scx*^low), dendritic cells (cluster 8, *Itgax*⁺), endothelial cells (cluster 9, 16, and 17, *Cdh5*⁺), stromal cells (cluster 11, *Prrx1*⁺/*Ly6a*⁻), T cells (cluster 12, *Cd3e*⁺), neuron (cluster 13, *Tubb3*⁺) and satellite cells (cluster 15, *Pax7*⁺) (Fig. 1E; Appendix Fig. S1B). We then examined *Gli1* expression across all these cell types and found that its expression was exclusively detected in tenocytes and tendon stem/progenitor cells in HO samples (Fig. 1F), and moreover, its expression was obviously strikingly higher in tendon stem/progenitor cells compared to that in tenocyte-lineage cells (Fig. 1F). In addition, *Gli1*⁺ cell abundance progressively increased following tenotomy (Fig. 1G).

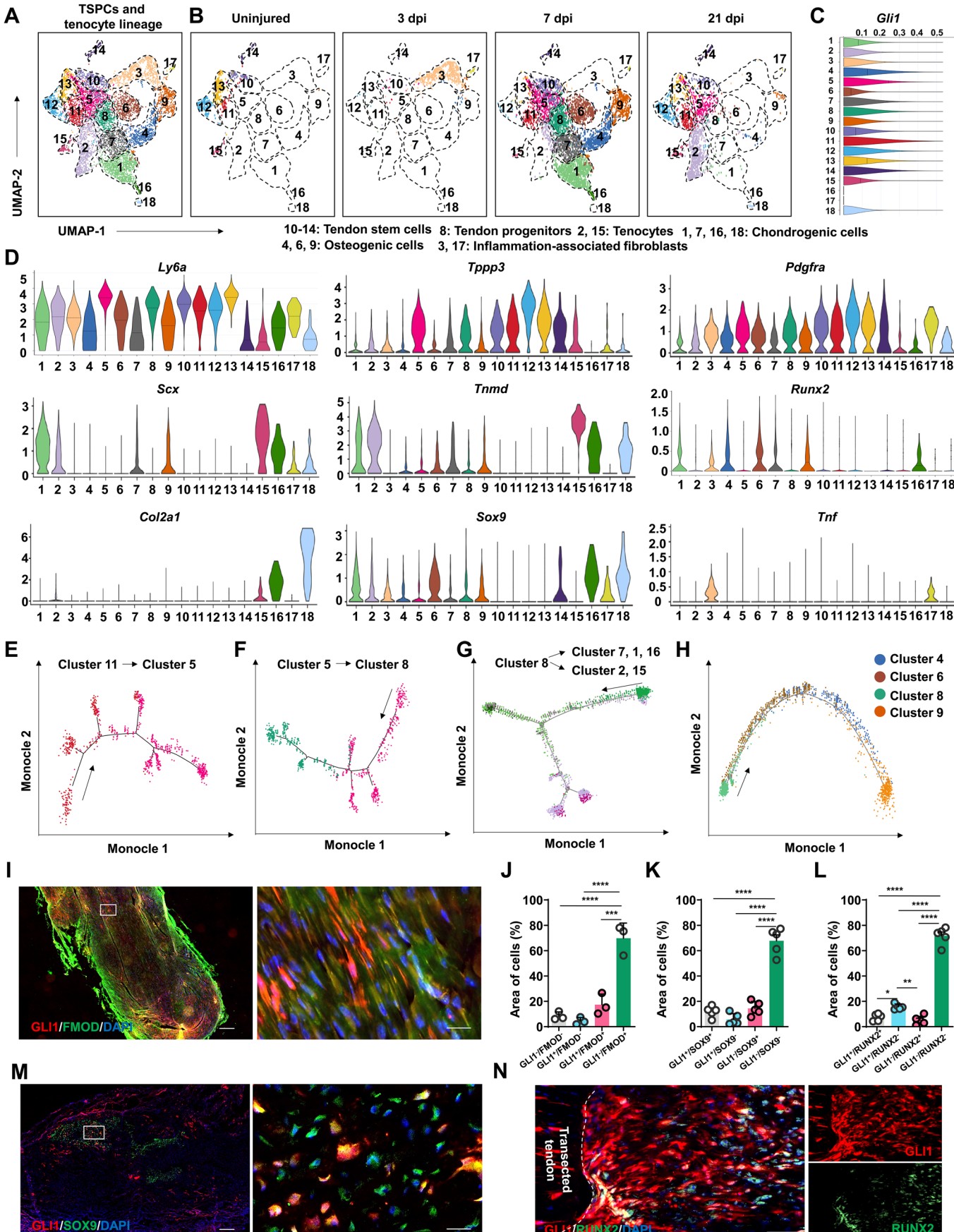

**Figure 2. *Gli1*+ tendon sheath progenitors give rise to tendon and osteochondral lineage progenitors.**

(A, B) Total (A) and separated (B) UMAP visualization of tenocytes and tendon stem/progenitors derived from uninjured and injured Achilles tendon in mice. (C) Violin plot of *Gli1* expression in each cluster cell derived from uninjured and injured mouse Achilles tendon of WT mice ($n = 3$ per group) at different time point (3, 7, and 21 dpi). (D) Violin plot of marker genes for identifying diverse mesenchymal and tendon cell types during HO formation ($n = 3$ per group). (E) Pseudotemporal trajectories of cluster 11 cells and cluster 5 cells in tenotomy site of WT mice using Monocle 2 analysis. (F) Pseudotemporal trajectories of cluster 5 cells and cluster 8 cells in tenotomy site of WT mice using Monocle 2 analysis. (G) Pseudotemporal trajectories of clusters 1, 2, 7, 8, 15, and 16 cells in tenotomy site of WT mice using Monocle 2 analysis. (H) Pseudotemporal trajectories of clusters 4, 6, 8 and 9 in tenotomy site of WT mice using Monocle 2 analysis. (I) Representative immunofluorescence images of GLI1 and FMOD in the injured site at 7 dpi. Scale bar, 200 μm. (J) Statistical analysis of frequency of GLI1⁻/FMOD⁻, GLI1⁺/FMOD⁺, GLI1⁻/FMOD⁺, GLI1⁺/FMOD⁻ cells in the injured site at 7 dpi ($n = 3$ per group). *P* values from left to right: ****$P = 5.39e-5$, ****$P = 3.39e-5$, ***$P = 1.75e-4$. (K) Statistical analysis of frequency of GLI1⁻/ SOX9⁻, GLI1⁺/ SOX9⁺, GLI1⁻/ SOX9⁺, GLI1⁺/ SOX9⁻ cells in injured site at 7 dpi ($n = 5$ per group). *P* values from left to right: ****$P = 2.78e-09$, ****$P = 6.06e-10$, ****$P = 5.55e-9$ (L) Statistical analysis of frequency of GLI1⁻/ RUNX2⁻, GLI1⁺/ RUNX2⁺, GLI1⁻/ RUNX2⁺, GLI1⁺/ RUNX2⁻ cells in injured site at 7 dpi ($n = 5$ per group). *P* values from left to right: *$P = 3.07e-2$, ****$P = 4.41e-13$, **$P = 5.88e-3$. ****$P = 3.36e-12$, ****$P = 2.58e-13$. (M) Representative immunofluorescence images of GLI1 and SOX9 in injured site at 7 dpi. Scale bar, 100 μm. (N) Representative immunofluorescence images of GLI1 and RUNX2 in injured site at 7 dpi. Scale bar, 100 μm. Data are presented as mean ± SD. All *P* values were determined by one-way ANOVA with Bonferroni post hoc test.

To establish whether *Gli1*+ tendon sheath progenitors contribute to HO formation, we generated mice with *Cre*-dependent *Gli1*+ cell depletion by crossing *Gli1-CreER^T2* with diphtheria toxin subunit A (DTA) strains. Tamoxifen administration to *Gli1-CreER^T2*; DTA mice induced conditional ablation of *Gli1*+ cells. To verify depletion efficacy, *Gli1-CreER^T2*; Ai9; DTA mice showed a significant reduction in *Gli1*+ cells relative to controls at 21 dpi (Fig. EV1A,B), confirming successful DTA-mediated ablation. This ablation substantially diminished the TPPP3⁺ tendon progenitor pool (Fig. EV1C,D) and correspondingly reduced osteochondral lineage populations, including SOX9⁺ chondrogenic progenitors, RUNX2⁺ osteochondrogenic progenitors, COL2⁺ mature chondrocytes, and OPN⁺ osteoblasts (Fig. EV1E–L). Histochemical staining with hematoxylin and eosin (HE), together with microCT analysis, showed that osteochondral differentiation was significantly inhibited in *Gli1-CreER^T2*; DTA mice, compared to that in *Gli1-CreER^T2* mice (Fig. 1H–J). Collectively, these results suggested that *Gli1*+ tendon sheath progenitors contribute to HO formation.

## *Gli1*+ tendon sheath progenitors transitioned into tenogenic and osteochondrogenic cells after Achilles tenotomy

As *Gli1* expression was restricted to tenocytes and tendon stem/progenitors in our above data (Fig. 1F), we next examined the diversity of subpopulations within these two cell lineages during HO development. UMAP analysis based on expression of signature genes in only the tenocytes and tendon stem/progenitors from our above scRNA-seq results identified 18 clusters of tenocytes and tendon stem/progenitor subpopulations and *Gli1* expression was elevated in clusters 4, 5, 7, 8, and 11-14 compared to other clusters (Fig. 2A–C; Appendix Fig. S2A). Among these clusters, only 10, 11, 12, 13, 14, and 15 could be detected in uninjured tendon samples (0 dpi) (Fig. 2B). Except for cluster 15, these subtypes expressed characteristically elevated levels of *Ly6a*, *Tppp3*, and *Pdgfra*, but not *Scx*, and were thus considered tendon stem cells (Fig. 2D). By contrast, cluster 15 showed high expressions of *Scx* and *Tnmd*, indicating these were tenocytes (Fig. 2D).

At 3 dpi, the inflammatory stage, clusters 3, 5 and 17 could be detected at the injury site, suggesting a likely role in immunoregulation (Fig. 2B). Cluster 5 also showed higher expression of *Tppp3* and *Pdgfra* than other groups, and were therefore designated as tendon progenitors, which likely transitioned from tendon stem

cells (clusters 10–14). Clusters 3 and 17 exhibited a *Tnf*+/*Pdgfra*^high/*Tppp3*^low/*Scx*^low, which suggested that these clusters were inflammation-associated fibroblasts (Fig. 2B,D) (Paolo et al, 2024). At 7 dpi, we found that clusters 1, 2, 4, 5, 6, 7, 8, 9, and 16 were enriched at the injury site (Fig. 2B). Similar to cluster 5 cells, clusters 7 and 8 expressed higher level of *Ly6a*, *Tppp3* and *Pdgfra*, characteristic of tendon progenitors. Cluster 2 exhibited an *Scx*^high/*Tnmd*^high/*Sox9*^mid/*Runx2*^low/*Tppp3*^low/*Pdgfra*^low phenotype, suggesting that these were tenocyte lineages (Fig. 2D). Clusters 1 and 16 exhibited a *Sox9*^high/*Runx2*^mid/*Scx*^high/*Tnmd*^high/*Ly6a*^low/*Tppp3*^low/*Pdgfra*^low, indicating these were chondrogenic lineage cells; the *Pdgfra*^mid/*Tppp3*^low/*Scx*^low/*Tnmd*^low/*Runx2*^high/Sox9^low phenotype of clusters 4, 6 and 9 suggested these cells could represent tendon stem cell-derived osteogenic lineage cells. Therefore, *Gli1*^high cells could represent tendon stem/progenitor cells and osteochondrogenic progenitors.

As cluster 11 comprised tendon stem cells, while cluster 5 represented progenitor cells that were the first to emerge following tenotomy, we speculated that cluster 5 may have originated from cluster 11 at 3 dpi. Monocle 2 analysis confirmed this potential fate trajectory from clusters 11–5 (Fig. 2E). In addition, cluster 8, which had been defined above as tendon progenitors based on their signature gene expression, did not emerge until after cluster 5 was detected. Further monocle 2 analysis suggested that cluster 5 cells could transition to the cluster 8 phenotype (Fig. 2F). We also noted that clusters 1, 2, 7, and 16 emerged at approximately the same time, suggesting that these clusters could represent possible differentiation fates of cluster 8. Indeed, monocle 2 analysis showed two states of descent from cluster 8, including state 1 (cluster 2) and state 2 (clusters 1, 7, and 16) (Fig. 2G). Furthermore, monocle 2 trajectory analysis demonstrated that chondrogenic tendon progenitors (cluster 7) could differentiate into premature chondrocytes (cluster 1) and ultimately mature chondrocytes (clusters 16 and 18, *Scx*^mid/*Sox9*^high/*Col2a1* ^high) at 21 dpi (Fig. 2D; Appendix Fig. S2B).

We speculated that cluster 8 tendon progenitors might also differentiate into osteogenic cell types, including clusters 4, 6, and 9. Monocle 2 analysis supported a scenario in which cluster 8 cells transitioned into cluster 6, then cluster 4 (defined as tendon stem cell-derived fibroblasts above), and finally cluster 9 (Fig. 2H), suggesting that cluster 9 could differentiate into bone tissue following chondrocyte maturation. These scRNA-seq analyses thus suggested that *Gli1*+ tendon sheath progenitors initially

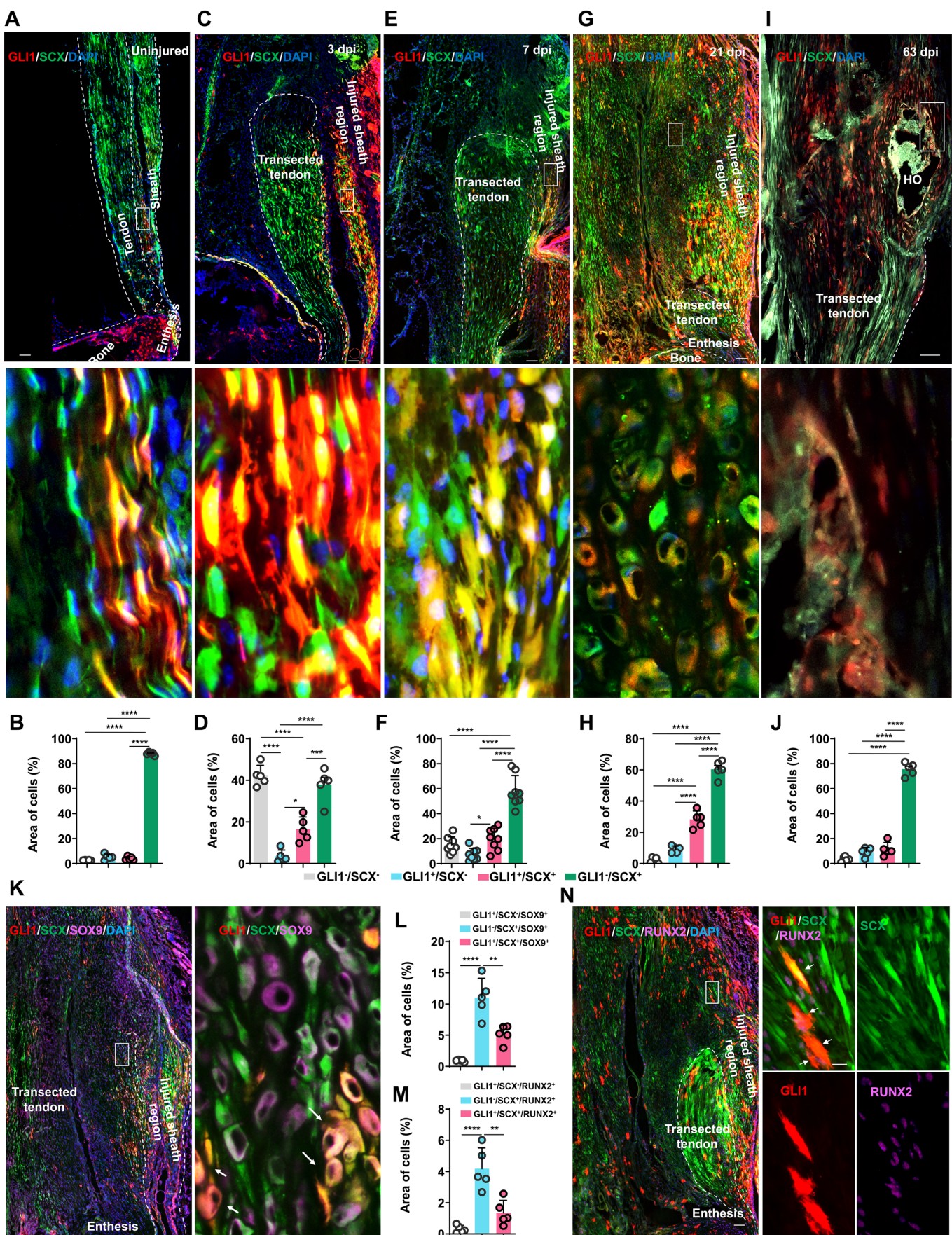

**Figure 3. Gli1⁺/Scx⁺ tendon sheath progenitors undergo osteochondrogenic differentiation.**

(A, B) Representative immunofluorescence images (A) and statistical analysis (B) of *Gli1⁺* and *Scx⁺* cells in uninjured Achilles tendon of *Gli1-CreER^{T2}*; Ai9; *Scx*-GFP mice (n = 5 per group). *P* values from left to right: ****P = 3.37e-15, ****P = 1.47e-12, ****P = 1.15e-13. Scale bar, 200 μm. (C, D) Representative immunofluorescence images (C) and statistical analysis (D) of *Gli1⁺* and *Scx⁺* cells in the injured Achilles tendon of *Gli1-CreER^{T2}*; Ai9; *Scx*-GFP mice at 3 dpi (n = 5 per group). *P* values from left to right: ****P = 5.07e-8, ****P = 1.17e-5, *P = 1.09e-2, ****P = 2.71e-7, ***P = 1.06e-4, Scale bar, 200 μm. (E, F) Representative immunofluorescence images (E) and statistical analysis (F) of *Gli1⁺* and *Scx⁺* cells in the injured Achilles tendon of *Gli1-CreER^{T2}*; Ai9; *Scx*-GFP mice at 7 dpi (n = 8 per group). *P* values from left to right: ****P = 7.09e-10, *P = 4.90e-2, ****P = 1.65e-11, ****P = 8.12e-9, Scale bar, 200 μm. (G, H) Representative immunofluorescence images (G) and statistical analysis (H) of *Gli1⁺* and *Scx⁺* cells in the injured Achilles tendon of *Gli1-CreER^{T2}*; Ai9; *Scx*-GFP mice at 21 dpi (n = 5 per group). *P* values from left to right: ****P = 1.28e-7, ****P = 7.97e-13, ****P = 8.13e-6, ****P = 5.73e-12, **P = 6.49e-9, Scale bar, 200 μm. (I, J) Representative immunofluorescence images (I) and statistical analysis (J) of *Gli1⁺* and *Scx⁺* cells in the injured Achilles tendon of *Gli1-CreER^{T2}*; Ai9; *Scx*-GFP mice at 63 dpi (n = 5 per group). *P* values from left to right: ****P = 6.0e-14, ****P = 2.20e-13, ****P = 3.65e-13, Scale bar, 200 μm. (K, L) Representative immunofluorescence images (K) and statistical analysis (L) of *Gli1⁺*, *Scx⁺* and *Sox9⁺* cells in the injured Achilles tendon of *Gli1-CreER^{T2}*; Ai9; *Scx*-GFP mice at 21 dpi (n = 5 per group). ****P = 9.35e-6, **P = 1.54e-3, Scale bar, 200 μm. (M, N) Representative immunofluorescence images (M) and statistical analysis (N) of *Gli1⁺*, *Scx⁺* and *Runx2⁺* cells in the injured Achilles tendon of *Gli1-CreER^{T2}*; Ai9; *Scx*-GFP mice at 21 dpi (n = 5 per group). ****P = 6.04e-5, **P = 1.03e-3, Scale bar, 200 μm. Data are presented as mean ± SD. All *P* values were determined by one-way ANOVA with Bonferroni post hoc test.

transitioned into tendon progenitors, then further differentiated into separate tenocyte and osteochondrogenic lineage subpopulations. Consistently, Slingshot analysis revealed that cluster 8 cells, originating from cluster 5 cells, could differentiate into tenogenic $(5 \rightarrow 8 \rightarrow 2 \rightarrow 15)$, chondrogenic $(5 \rightarrow 8 \rightarrow 7 \rightarrow 1 \rightarrow 16 \rightarrow 18)$ and osteogenic $(5 \rightarrow 8 \rightarrow 4 \rightarrow 9 \rightarrow 17)$ lineage cells (Appendix Fig. S2C,D).

To confirm that *Gli1⁺* tendon sheath progenitors do, in fact, give rise to tenocytes and osteochondrogenic lineages, we again induced HO by tenotomy in *Gli1*-labeled *Gli1-CreER^{T2}*; Ai9 mice and analyzed tissues at 7 and 21 dpi. Injured and uninjured tendons underwent fluorescence-activated cell sorting (FACS) to isolate *Gli1⁺* and *Gli1⁻* populations, followed by scRNA-seq. UMAP-based dimensionality reduction and graph-based clustering revealed 11 transcriptionally distinct cell clusters, which we annotated as: tendon stem cells (clusters 3, 4 and 8; *Tppp3^{high}/ Ly6a^{High}/Scx^{low}*), tendon progenitors (clusters 1, 2, 6 and 9; *Tppp3^{mid}/Ly6a^{high}/Scx^{high}*), Tenocytes (cluster 7; *Tppp3^{low}/ Ly6a^{low}/Scx^{high}/Fmod^{high}*) and osteochondral lineages (cluster 5: Fibrochondrocytes, *Tppp3^{low}/Sox9⁺/ Acan⁻/Tnfaip6^{high}*; cluster 11, mature chondrocytes, *Tppp3^{low}/ Col2a1^{high}/Acan⁻/Tnfaip6^{low}*; cluster 10, osteogenic progenitors, *Tppp3^{mid}/Ly6a^{high}/Scx^{high}/Runx2^{high}*) (Fig. EV2A,B). RNA velocity analysis demonstrated a hierarchical differentiation trajectory from tendon stem cells→tendon progenitors→tenocytes/ osteochondral lineages (Fig. EV2C). *Gli1⁻* cells also contribute to tenogenic and osteochondral cells during tendon repairing process (Appendix Fig. S3).

In addition, immunofluorescent staining revealed that *Gli1⁺* cells expressed the tenocyte marker, FMOD, chondrogenic marker, SOX9 and osteogenic marker, RUNX2, in *Gli1-CreER^{T2}*; Ai9 mice at 7 dpi following tenotomy. (Fig. 2I–N). At 21 dpi, *Gli1⁺* cells clearly exhibited COL2 expression, suggesting they had differentiated into chondrocytes (Appendix Fig. S4A,B). We noted that *Gli1⁺* cells expressed OCN at 63 dpi (Appendix Fig. S4C,D). Taken together, these results indicated that *Gli1⁺* tendon sheath progenitors could differentiate into tenocytes, chondrocytes, osteocytes, and participate in HO development in vivo.

## Osteochondrogenic *Gli1⁺* tendon sheath progenitors originate from *Scx⁺* tendon stem/progenitor cells

Tendon sheath comprised of tendon stem/progenitor cells, FAPs, and other fibroblasts (Ashley L and Michael T, 2019). Scleraxis

(*Scx*) is expressed in tendon stem/progenitor cells and their progeny cells (Guak-Kim et al, 2020), and we noted that terminating tenocytes (cluster 2) and osteochondral cells (clusters 16 and 9) expressed higher level of *Scx* (Fig. 2D), suggesting that *Gli1⁺* tendon sheath progenitors could be a subpopulation of tendon stem/progenitor cells. To investigate this, we traced the developmental distribution of *Gli1⁺* cells in the Achilles tendon at embryonic day 14.5 and postnatal days 0, 3, and 14 – key stages of tendon development. Lineage tracing analysis (*Gli1-CreER^{T2}*; ZsGreen) demonstrates that *Gli1⁺* cells were initially localized to the tendon sheath at E14.5, subsequently expanding throughout the tendon tissue to encompass both the sheath and mid-substance regions by P0, P3, and P14 (Appendix Fig. S5). These findings provide definitive evidence that *Gli1⁺* cells function as bona fide tendon stem/progenitor cells under homeostatic conditions.

To further confirm that tenogenic *Gli1⁺* tendon sheath progenitors, rather than other *Gli1⁺* lineages, contributed to HO formation, we generated *Gli1-CreER^{T2}*; Ai9; *Scx*-GFP triple-mutant transgenic mice to examine GLI1 co-localization with SCX. Fluorescence imaging showed that the majority of *Gli1⁺* cells expressed SCX in uninjured tendon sheath region (Fig. 3A,B). At 3 dpi, *Gli1⁺* cells in the injured region were also observed to express SCX in tenotomized mice (Fig. 3C,D), with *Gli1⁺/Scx⁺* cells representing the dominant population among total *Gli1⁺* cells at 7 dpi (Fig. 3E,F). At 21 dpi, we found that GLI1 indeed colocalized with SCX in the chondrocyte region, supporting that *Gli1⁺* tenocyte-lineage cells give rise to chondrocytes (Fig. 3G,H). Consistent with this finding, *Gli1⁺* cells also expressed SCX in the heterotopic osteoid region at 63 dpi (Fig. 3I,J).

To further confirm that *Gli1⁺/Scx⁺* tendon stem/progenitor cells occupied the capacity for osteochondral differentiation, we examined the expression of chondrogenic differentiation marker, SOX9, osteogenic marker, RUNX2, in *Gli1-CreER^{T2}*; Ai9; *Scx*-GFP mice at 21 dpi following tenotomy. Indeed, among *Gli1⁺* populations, *Gli1⁺/Scx⁺* cells expressed SOX9, with rarely detectable *Gli1⁺/Scx⁻/SOX9⁺* cells in injured tendon region (Fig. 3K,L). Consistently, Only *Gli1⁺/Scx⁺* cells expressed RUNX2, but not *Gli1⁺/Scx⁻* cells (Fig. 3M,N). Taken together, these results demonstrated that *Gli1⁺* tendon sheath progenitors, as a population of tendon stem/progenitor cells, could differentiate into chondrocytes or osteocytes following tendon injury in mice in vivo.

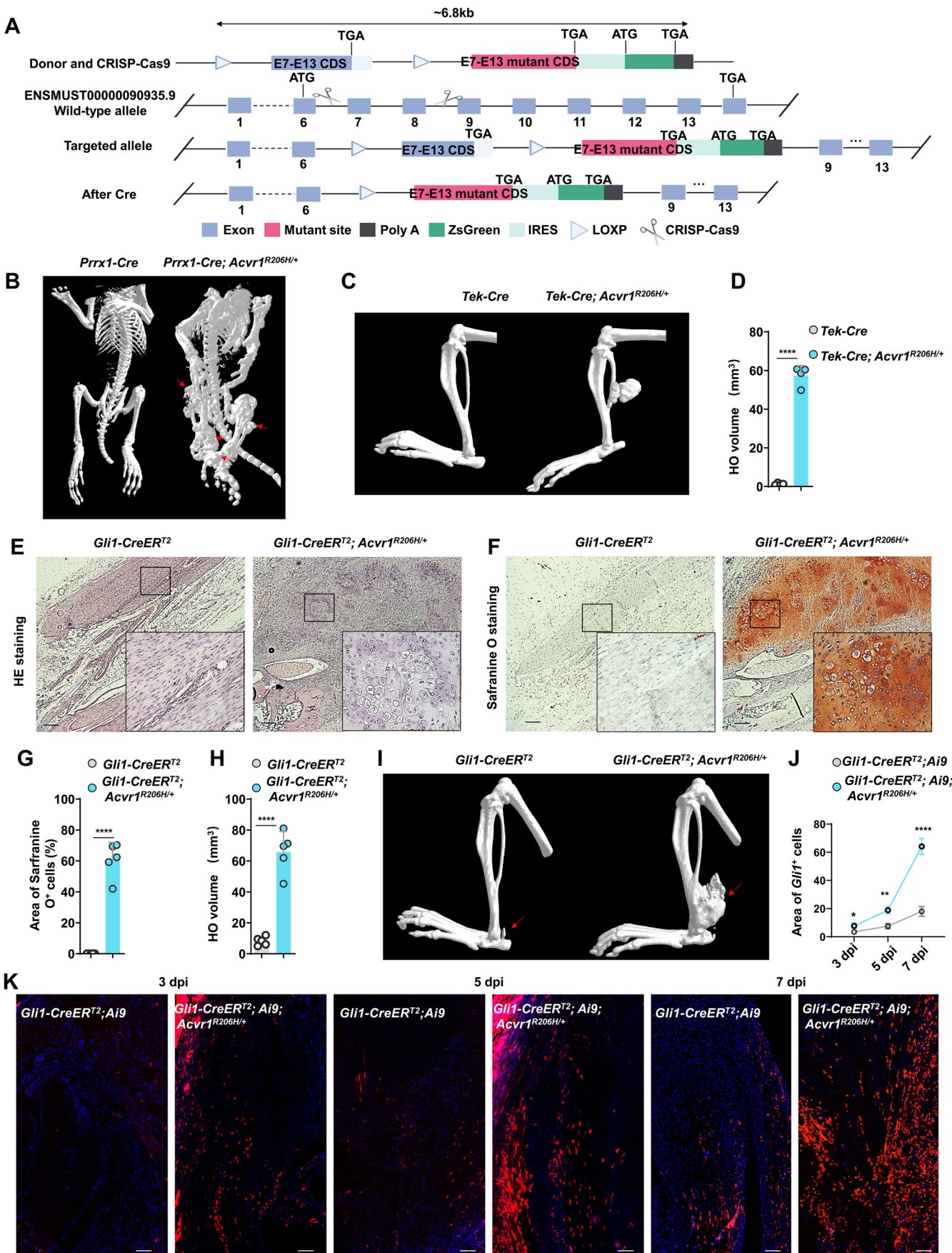

**Figure 4.   *Gli1*⁺ tendon sheath progenitors initiate tendon HO formation in FOP model mice.**

(A) Schematic diagram depicting the generation of *Acvr1^R206H/+^ flox* mice using *CRISPR-Cas9* technology. (B) microCT showing spontaneous HO formation around the joints of *Prrx1-Cre; Acvr1^R206H/+^* mice, but not *Prrx1-Cre* mice. (C, D) Representative microCT image (C) and statistical analysis (D) of HO formation in the tibial muscle of *Tek-Cre; Acvr1^R206H/+^* mice, but not *Tek-Cre* mice after CTX injury (n = 4 per group). ****P = 6.05e-7. (E, F) Representative HE staining (E) and safranine O staining (F) images of the injured tendon of *Gli1-CreER^T2^* and *Gli1-CreER^T2^; Acvr1^R206H/+^* mice at 7 dpi. Scale bar, 200 μm. (G) Statistical analysis of Safranine O⁺ region in injured tendon of *Gli1-CreER^T2^* and *Gli1-CreER^T2^; Acvr1^R206H/+^* mice at 7 dpi (n = 5 per group). ****P = 2.28e-6. (H) Statistical analysis of HO volume in the injured tendon of *Gli1-CreER^T2^* and *Gli1-CreER^T2^; Acvr1^R206H/+^* mice at 56 dpi (n = 5 per group). ****P = 1.29e-5. (I) Representative microCT images of HO in the injured tendon of *Gli1-CreER^T2^* and *Gli1-CreER^T2^; Acvr1^R206H/+^* mice at 56 dpi. (J, K) Statistical analysis (J) and representative immunofluorescence images (K) of *Gli1*⁺ cells in the injured tendon of *Gli1-CreER^T2^; Ai9* and *Gli1-CreER^T2^; Ai9; Acvr1^R206H/+^* mice at 3, 5, and 7 dpi (n = 5 per group). *P = 3.86e-2, **P = 3.82e-3, ****P = 2.73e-4. Scale bar, 100 μm. Data are represented as the mean ± SD. All P values were determined by unpaired Student's *t* test.

## FOP-associated *Acvr1^R206H/+^* mutation accelerates osteochondral differentiation of *Gli1*⁺ tendon sheath progenitors

To examine whether aberrant osteochondral differentiation of *Gli1*⁺ tendon sheath progenitors lead to HO after tendon injury in FOP model mice, we established a conditional knock-in FOP model mouse harboring the *Acvr1^R206H/+^* mutation. For this purpose, we used the *CRISPR-Cas9* system with donor RNAs targeting floxed endogenous *Acvr1* exons 7–13 to introduce the mutant exons 7–13 fused to a luciferase-based *ZsGreen* reporter in germ cells of *Acvr1^R206H/+^* mice (Fig. 4A). This modification preserved the integrity of the *Acvr1* exon locus in the absence of *Cre*, thereby avoiding deleterious effects of *Acvr1* deficiency on mouse development. The wild-type (WT) *Acvr1* exons 7–13 were then excised and mutant *Acvr1* exons were expressed upon induction of *Cre* expression and traced by ZsGreen signal (Fig. 4A). To confirm the pathological phenotype of this FOP model mice, we crossed mice harboring limb-specific *Prrx1-cre* with *Acvr1^R206H/+^* mice. MicroCT examination showed that *Prrx1-Cre; Acvr1^R206H/+^* mice, rather than *Prrx1-Cre* mice, exhibited spontaneous HO around the joints at the juvenile stage (Fig. 4B). In addition, we also introduced the *Acvr1^R206H/+^* genotype in fibro-adipogenic progenitors (FAPs) using *Tek-Cre* transgenic mice and found that cardiotoxin (CTX) injury in muscle similarly resulted in HO development in these *Tek-Cre; Acvr1^R206H/+^* mice (Fig. 4C,D).

Next, we generated *Gli1-CreER^T2^; Acvr1^R206H/+^* mice by crossing *Gli1-CreER^T2^* mice with *Acvr1^R206H/+^* mice. We performed tenotomy after inducing *Gli1* promoter-dependent *Cre* expression by tamoxifen treatment in *Gli1-CreER^T2^; Acvr1^R206H/+^* mice for 1 week. To confirm the conditional *Acvr1^R206H/+^* mutation occurred after *Gli1-Cre* expression following tenotomy injury, we first performed immunofluorescence comparing ACVR1 expression patterns in injured tendons of *Gli1-CreER^T2^; Ai9* versus *Gli1-CreER^T2^; Acvr1^R206H/+^; Ai9* mice at 5 dpi, with particular focus on spatial co-localization of ACVR1 within *Gli1*⁺ lineage cells. Critically, no significant difference in ACVR1 expression was observed between groups, excluding the hypothesis that elevated ACVR1 expression (rather than the *Acvr1^R206H/+^* mutation) drives HO in tendons (Fig. EV3A,B). The *Acvr1^R206H/+^* mutation classically triggers p-SMAD1/5 pathway activation. To functionally validate pathway activation, we performed phospho-SMAD1/5 (p-SMAD1/5) immunofluorescence comparing signaling intensity in injury-induced HO niches versus physiological repair zones. Strikingly, the majority of *Gli1*⁺ cells in *Gli1-CreER^T2^; Acvr1^R206H/+^; Ai9* tendons exhibited robust p-SMAD1/5 activation, whereas *Gli1*⁺ cells in control (*Gli1-CreER^T2^; Ai9*) tendons showed negligible signaling (Fig. EV3C,D).

Complementing this, fluorescence-activated cell sorting (FACS) of *Gli1*⁺ (tdTomato⁺) cells from injured tendons (5 dpi) of both *Gli1-CreER^T2^; Ai9* and *Gli1-CreER^T2^; Acvr1^R206H/+^; Ai9* mice. Sanger sequencing of sorted cells confirmed successful disruption of the STOP cassette and expression of the mutant allele in *Gli1-CreER^T2^; Acvr1^R206H/+^* mice (Fig. EV3E; Appendix Fig. S6), enabling quantification of mutant allele burden. These results collectively confirmed successful *Acvr1^R206H/+^* mutant receptor expression and BMP pathway hyperactivation, further validating our FOP mouse model establishment.

Hematoxylin and Eosin staining and Safranine O staining showed that *Gli1-CreER^T2^; Acvr1^R206H/+^* mice displayed rapid chondrocytes emerging at 7 dpi (Fig. 4E–G). These findings sharply contrasted from the relatively slow formation of endochondral bone at the tenotomy site of *Gli1-CreER^T2^* mice over 9 weeks post injury, i.e., we used microCT imaging to confirm the role of *Gli1*⁺ cells in aberrant osteogenesis of FOP model mice following tendon injury and found that, at 8 weeks after tenotomy, *Gli1-CreER^T2^; Acvr1^R206H/+^* mice exhibited significantly greater osteogenic differentiation of *Gli1*⁺ tendon sheath progenitors compared to that in *Gli1-CreER^T2^* mice (Fig. 4H,I), thus highlighting the pathological role of heterozygous *Acvr1^R206H^* mutation in accelerating *Gli1*⁺ tendon sheath progenitor differentiation in HO.

In addition, we traced the distribution of *Gli1*⁺ cells during HO formation in injured Achilles tendon of FOP model mice (*Gli1-CreER^T2^; Acvr1^R206H/+^; Ai9*). This fluorescence microscopy imaging showed that the number of *Gli1*⁺ tendon sheath progenitors at inflammatory, fibroproliferative and chondrogenic stages occurred at 3, 5 and 7 dpi, respectively, in *Gli1-CreER^T2^; Acvr1^R206H/+^; Ai9* mice, was significantly higher than those in *Gli1-CreER^T2^; Ai9* mice (Fig. 4J,K). This rapid expansion of *Gli1*⁺ sheath cells with *Acvr1^R206H/+^* mutation could be the underlying mechanism of acceleration of HO formation in FOP. Overall, these findings demonstrated that *Gli1*⁺ tendon sheath progenitors undergo osteochondral differentiation faster and formed tendon HO to a greater extent in the *Acvr1^R206H/+^* background of FOP model mice.

## *Gli1*⁺ tendon sheath progenitors differentiate into tenocyte and osteochondral lineage progenitors during endochondral ossification in FOP mice

Since *Gli1*⁺ tendon sheath progenitors initially transitioned into tenocyte and osteochondral lineages in traumatic tendon HO, we next examined whether *Gli1*⁺ tendon sheath progenitors expressed FMOD at the injury site in vivo in tenotomized FOP model mice (*Gli1-CreER^T2^; Acvr1^R206H/+^; Ai9*). At 3 days post-tenotomy, immunofluorescence staining showed that GLI1⁺/FMOD⁺ cells

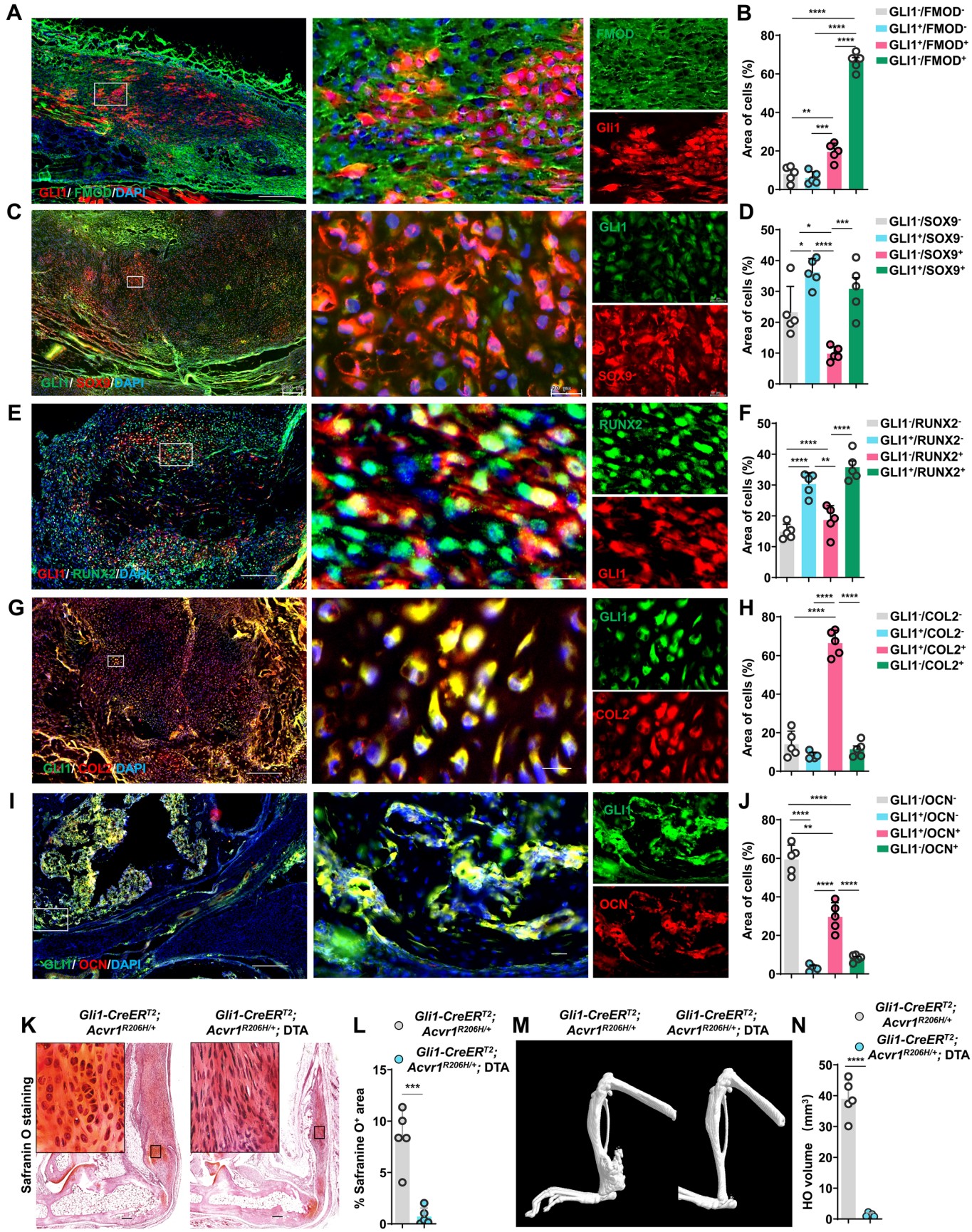

◄ **Figure 5.** *Acvr1$^{R206H/+}$* mutation accelerates osteochondral differentiation of *Gli1$^+$* tendon sheath progenitors.

(A, B) Representative immunofluorescence images (A) and statistical analysis of frequency (B) of GLI1$^-$/ FMOD$^-$, GLI1$^+$/ FMOD$^-$, GLI1$^-$/ FMOD$^+$, GLI1$^-$/ FMOD$^+$ cells in the injured site of FOP model mice at 3 dpi ($n = 5$ per group). *P* values from left to right: **$P = 2.27$e-3, ****$P = 1.17$e-12, ***$P = 4.59$e-4, ****$P = 6.94$e-13, ****$P = 2.49$e-11. Scale bar, 200 or 20 μm. (C, D) Representative immunofluorescence images (C) and statistical analysis of frequency (D) of GLI1$^-$/ SOX9$^-$, GLI1$^+$/SOX9$^-$, GLI1$^-$/SOX9$^+$, GLI1$^+$/ SOX9$^+$ cells in the injured site of FOP model mice at 5 dpi ($n = 5$ per group). *P* values from left to right: *$P = 2.52$e-2, *$P = 1.85$e-2, ****$P = 3.43$e-5, ***$P = 4.18$e-4. Scale bar, 200 or 20 μm. (E, F) Representative immunofluorescence images (E) and statistical analysis of frequency (F) of GLI1$^-$/RUNX2$^-$, GLI1$^+$/RUNX2$^-$, GLI1$^-$/RUNX2$^+$, GLI1$^+$/RUNX2$^+$ cells in the injured site of FOP model mice at 5 dpi ($n = 5$ per group). *P* values from left to right: ****$P = 5.56$e-5, ****$P = 1.32$e-6, **$P = 1.12$e-3, ****$P = 1.73$e-5. Scale bar, 200 or 20 μm. (G, H) Representative immunofluorescence images (G) and statistical analysis of frequency (H) of GLI1$^-$/COL2$^-$, GLI1$^+$/ COL2$^-$, GLI1$^+$/ COL2$^+$, GLI1$^-$/ COL2$^+$ cells in the injured site of FOP model mice at 7 dpi ($n = 5$ per group). *P* values from left to right: ****$P = 1.52$e-10, ****$P = 2.90$e-11, ****$P = 6.99$e-11. Scale bar, 200 or 20 μm. (I) Representative immunofluorescence images showing *Gli1$^+$* tendon sheath progenitors differentiate into osteocytes at 14 dpi in FOP model mice. Scale bar, 200 or 20 μm. (J) Statistical analysis of the frequency of GLI1$^-$/OCN$^-$, GLI1$^+$/OCN$^+$, GLI1$^-$/OCN$^+$, GLI1$^+$/OCN$^-$ cells in the injured site of FOP model mice at 14 dpi ($n = 5$ per group). *P* values from left to right: ****$P = 8.87$e-11, ****$P = 9.49$e-7, ****$P = 4.35$e-10, ****$P = 3.85$e-6, ****$P = 6.35$e-5. (K, L) Representative Safranine O staining images (K) and statistical analysis (L) of chondrocyte region in the injured tendon of *Gli1-CreER$^{T2}$*; *Acvr1$^{R206H/+}$* and *Gli1-CreER$^{T2}$*; *Acvr1$^{R206H/+}$*; DTA mice ($n = 5$ per group). ***$P = 3.27$e-4. Scale bar, 200 μm. (M, N) Representative microCT images (M) and statistical analysis (N) of HO in the injured tendon of *Gli1-Cre$^{ERT2}$*; *Acvr1$^{R206H/+}$* and *Gli1-Cre$^{ERT2}$*; *Acvr1$^{R206H/+}$*; DTA mice at 63 dpi ($n = 5$ per group). ****$P = 7.86$e-7. Data are presented as mean ± SD. All *P* values were determined by One-way ANOVA with Bonferroni post hoc test (B, D, F, H, J) and unpaired Student's *t* test (L, N).

were significantly more abundant than GLI1$^+$/FMOD$^-$ cells at the injury site (Fig. 5A,B), suggesting that the majority of *Gli1$^+$* cells give rise to tenocyte-lineage cells. Since GLI1$^+$/FMOD$^+$ cells represented tendon progenitor cells with the capacity for osteochondral differentiation in traumatic HO, we therefore subsequently examined SOX9 and RUNX2 expression in *Gli1$^+$* cells at 5 dpi and found that ~30.8% of *Gli1$^+$* cells (stained by the ZsGreen Ab) expressed SOX9 (Fig. 5C,D), while 35.7% cells (Ai9$^+$) expressed RUNX2, suggesting their potential to contribute to HO via endochondral ossification (Fig. 5E,F).

To further trace the differentiation of *Gli1$^+$* lineage cells into chondrocytes, we examined expression of the chondrocyte maturation marker, COL2, at the injury site of tenotomized FOP mice at 7 dpi, corresponding to the chondrogenic stage. Immunofluorescence staining of injured tendon sections indicated that ~66.5% *Gli1$^+$* cells (stained by the ZsGreen Ab) also expressed COL2, thus demonstrating that *Gli1$^+$* cells could differentiate into chondrocytes after differentiating into the tenocyte lineage (Fig. 5G,H).

We then examined osteogenic differentiation of *Gli1$^+$* cells into osteocytes by immunofluorescence staining for the osteocyte differentiation marker, OCN, in injured tendon of FOP at 14 dpi, which corresponds to the osteogenesis stage. This analysis revealed that ~29.6% *Gli1$^+$* cells expressed OCN, which indicated that *Gli1$^+$* cells could differentiate into osteocytes following aberrant chondrogenic differentiation (Fig. 5I,J). In addition, depletion of *Gli1$^+$* cells by tamoxifen treatment in *Gli1-CreER$^{T2}$*; *Acvr1$^{R206H/+}$*; DTA mice, as above, resulted in significantly suppressing formation of both heterotopic cartilage and bone tissue in the injured tendon (Fig. 5K–N). Overall, these results revealed that *Gli1$^+$* tendon sheath progenitors differentiate into chondrocytes and osteocytes in FOP mice.

## GNAS/PKA signaling is activated in the osteochondral lineage of *Gli1$^+$* tendon sheath progenitors

Since cluster 8 tendon progenitors showed greater capacity for osteochondral differentiation than cluster 5 and represented the cellular source for tenotomy-related HO, we performed Gene Ontology (GO) analysis of differentially expressed genes (DEGs) between clusters 8 and 5 from our above scRNA-seq. Notably, DEGs annotated with "osteoblast differentiation" were enriched in

cluster 8, further supporting their osteochondral progenitor identity during HO formation (Fig. 6A). We noted that five ossification-associated genes in the Top 20 DEGs, including *Ptn*, *Dlk1*, *Mdk*, *Igf2* and *Gnas*, were expressed at remarkably higher levels in cluster 8 compared to cluster 5 (Fig. 6B). In particular, *Gnas* had the highest enrichment score, although the relationship between elevated *Gnas* expression and aberrant osteochondral differentiation in tendon progenitors has not been documented. We therefore examined the role of *Gnas* in tendon HO formation by examining its expression across cell types in our scRNA-seq data and found that *Gnas* was highly expressed in cluster 8 tendon progenitors, but not in tendon stem cells (clusters 10–14) or tendon lineage cells (clusters 1, 2, 7, 9, 16, and 18) (Fig. 6C). These results suggested that *Gnas* may be essential for differentiation of osteochondral progenitors, but not terminal differentiation or maintenance of quiescent stem cells.

We next examined the co-expression of GNAS and GLI1 in our traumatic HO model by immunofluorescence staining of injured and uninjured tendon at different time points in *Gli1-CreER$^{T2}$*; Ai9 reporter mice. Image analysis showed that GNAS expression was lowest in uninjured tendon and highest in injured tendon at 7 dpi (Fig. 6D,E), i.e., very few *Gli1$^+$* cells also expressed GNAS in uninjured tendon of *Gli1-CreER$^{T2}$*; Ai9 mice. By contrast, GLI1$^+$/GNAS$^+$ cells could be detected at significantly increased frequency at 7 dpi compared to the uninjured group. However, the abundance of GLI1$^+$/GNAS$^+$ cells was significantly decreased in 21 dpi and 63 dpi tendon sections, but still significantly higher than that in uninjured tendon (Fig. 6D,F), further supporting GNAS as a key regulator of osteochondral differentiation in tendon progenitors. Previous studies have shown that GNAS overexpression leads to increased cAMP production and PKA activation, the latter of which is known to regulate cell proliferation, differentiation, and other biological processes (Krushna C et al, 2018). Therefore, we next examined PKA activation in *Gli1$^+$* progenitors during HO formation by immunofluorescence staining for the activation marker, phospho (p)-PKA substrate. We observed a significantly elevated p-PKA substrate signal in injured tendons at 7 dpi compared with uninjured controls (Fig. 6G,I,J), but like GNAS, it was significantly decreased at 21 and 63 dpi without returning to pre-injury baseline levels (Fig. 6G,I,K). Moreover, we also observed GLI1$^+$/p-PKA substrate$^+$ cells were significantly increased at 7 dpi compared to the corresponding population in uninjured tendon

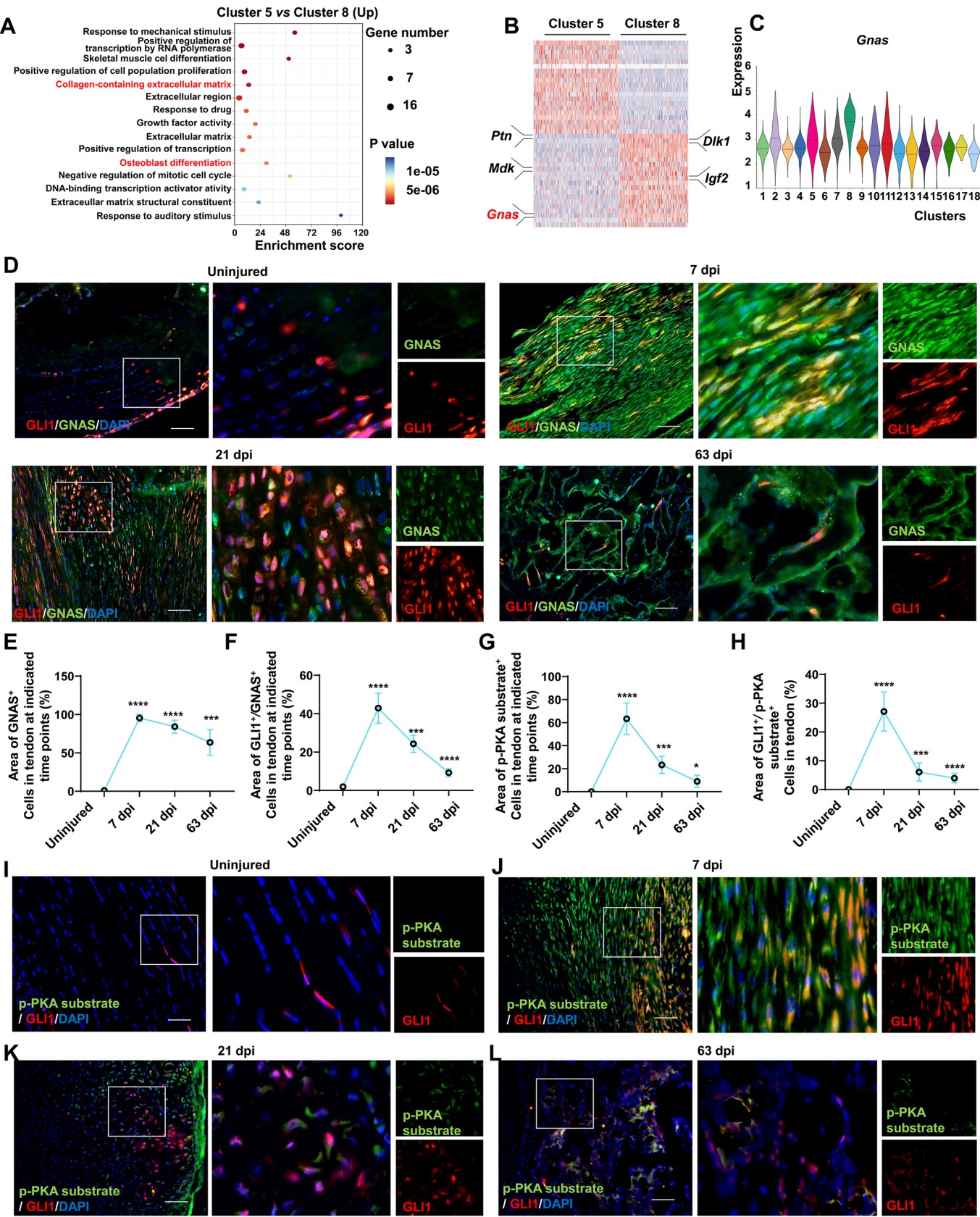

**Figure 6.   GNAS/PKA signaling pathway was activated in osteochondral progenitors during HO formation.**

(A) GO analysis of DEG between clusters 5 and 8. (B) Heatmap of DEG in clusters 5 and 8. (C) Feature plot showing the expression of *Gnas* in mesenchymal and tendon lineage cells. (D) Representative immunofluorescence images of GNAS expression in uninjured tendon and injured site of *Gli1-CreERT2*; Ai9 mice at 7, 21, and 63 dpi. Scale bar, 50 μm. (E, F) Statistical analysis of GNAS⁺ region (E) and GNAS⁺/GLI1⁺ region (F) in uninjured tendon and injured site of *Gli1-CreERT2*; Ai9 mice at 7, 21, and 63 dpi (*n* = 5 per group). *P* values in (E): 7 dpi vs uninjured ****$P = 9.47e-11$. 21 dpi vs uninjured ****$P = 2.03e-8$, 63 dpi vs uninjured ****$P = 3.21e-5$. *P* values in (F): 7 dpi vs uninjured ****$P = 2.64e-6$, 21 dpi vs uninjured ****$P = 3.64e-6$, 63 dpi vs uninjured ****$P = 9.58e-5$. (G, H) Statistical analysis of p-PKA substrate⁺ region (G) and p-PKA substrate⁺/GLI1⁺ region (H) in uninjured tendon and injured site of *Gli1-CreERT2*; Ai9 mice at 7, 21, and 63 dpi (*n* = 5 per group). *P* values in (G): 7 dpi vs uninjured ****$P = 6.33e-6$, 21 dpi vs uninjured ***$P = 1.15e-4$, 63 dpi vs uninjured *$P = 1.17e-2$. *P* values in (H): 7 dpi vs uninjured ****$P = 1.96e-5$, 21 dpi vs uninjured ***$P = 2.63e-3$, 63 dpi vs uninjured ****$P = 6.44e-4$. (I–L) Representative immunofluorescence images of p-PKA substrate expression in uninjured tendon (I) and injured site of *Gli1-CreERT2*; Ai9 mice at 7 (J), 21 (K), and 63 dpi (L). Scale bar, 50 μm. Data are presented as mean ± SD. All *P* values were determined by unpaired Student's *t* test.

(Fig. 6H,I,J). Subsequently, these dual-positive cells underwent a significant reduction in samples collected at 21 dpi and later time points (Fig. 6H,I,K,L). Moreover, both GNAS⁺/GLI1⁺ and GNAS⁺/GLI1⁻ subpopulations comprised osteochondrogenic progenitors, mature chondrocytes, and osteoblasts. Similarly, p-PKA substrate⁺/GLI1⁺ and p-PKA substrate⁺/GLI1⁻ cells exhibited identical differentiation potential, representing the same osteochondral lineages (Appendix Figs. S7 and S8), suggesting that GNAS/PKA pathway is critically involved in driving aberrant osteochondrogenic differentiation during HO pathogenesis.

We next investigated whether GNAS/PKA signaling was also activated during aberrant osteochondral differentiation by *Gli1*⁺ tendon sheath progenitors in FOP model mice (*Gli1-CreERT2*; *Acvr1R206H/+*). At 5 dpi, immunofluorescence staining showed that both GNAS and p-PKA substrate expression at the tendon injury site was significantly increased in FOP model mice tendon compared to that in uninjured tendon, then subsequently decreased at 7 and 14 dpi (Fig. EV4A–F), which was consistent with observations in traumatic HO mice. Furthermore, levels of GLI1⁺/GNAS⁺ cells and GLI1⁺/p-PKA substrate⁺ cells were significantly higher at 5, 7 and 14 dpi than in uninjured tendons of FOP model mice (Fig. EV4G–L). These collective results revealed that GNAS/PKA signaling was activated during aberrant osteochondral differentiation of *Gli1*⁺ tendon progenitors in both traumatic and genetic HO model mice.

## Inhibiting GNAS/PKA signaling prevents tendon HO

To investigate the function of GNAS/PKA signaling in tendon HO formation, we performed tenotomy in WT mice, then treated the injured mice with a selective Gsα antagonist, NF449. Since *Gnas* was highly expressed in *Gli1*⁺ osteochondral progenitors (cluster 8) at 7 dpi, we intraperitoneally injected NF449 for 7 days following tenotomy, and sacrificed mice at 7, 21, and 63 dpi. At 7 dpi, SOX9 and RUNX2 expression at the tenotomy site was significantly lower in NF449-treated mice compared to the vehicle controls (Fig. 7A–D). Moreover, the populations of GLI1⁺/SOX9⁺ and GLI1⁺/RUNX2⁺ cells in injured tendons of *Gli1-CreERT2*; Ai9 mice were both significantly decreased after NF449 treatment (Appendix Fig. S9A–D). Safranine O staining at 21 dpi in NF449- and vehicle-treated tenotomized mice revealed that mice in the NF449 group had a significantly smaller chondrocyte region than vehicle control mice (Fig. 7E,F). Consistent with this effect, NF449 also significantly inhibited HO volume at 63 dpi (Fig. 7G,H).

To further explore PKA regulation of osteochondral differentiation in *Gli1*⁺ tendon sheath progenitors, we generated *Prkacaflox/flox* mice,

which we then crossed with *Gli1-CreERT2* mice to abrogate expression of *Prkaca*, a catalytic subunit of PKA. We found that *Gli1-CreERT2*; *Prkacaflox/flox* mice had significantly reduced safranine O⁺ area and HO volume compared to that in *Gli1-CreERT2* mice (Fig. 7I–L). In addition, populations of GLI1⁺/SOX9⁺ and GLI1⁺/RUNX2⁺ cells in injured tendons of *Gli1-CreERT2*; *Prkacaflox/flox*; Ai9 mice showed significant reductions relative to *Gli1-CreERT2*; Ai9 mice, suggesting that *Prkaca* inhibition impaired osteochondral differentiation in *Gli1*⁺ tendon sheath progenitors (Appendix Fig. S9E–H).

We also examined the effect of Gsα inhibitor, NF449, on tendon HO formation in FOP mice. For this experiment, we immediately treated *Gli1-CreERT2*; *Acvr1R206H/+* mice with NF449 for 7 days following tenotomy, then sacrificed the mice at 7 and 14 dpi. We observed that NF449 could significantly suppress heterotopic endochondral bone formation in these FOP mice (Fig. 7M–P). Consistent with these results, *Prkaca* ablation in *Gli1*⁺ cells also resulted in significantly reduced HO (Fig. 7Q–T). CREB, a downstream target of PKA, also regulates the osteochondral differentiation of *Gli1*⁺ cells, and inhibition of CREB significantly reduces HO formation (Fig. EV5). Cumulatively, these results confirmed that the GNAS/PKA signaling pathway indeed positively regulated aberrant osteochondral differentiation of *Gli1*⁺ cells during HO formation.

## The spatial density of GNAS and p-PKA substrate is associated with HO in humans

To further evaluate the contribution of GLI1 and associated GNAS/PKA signaling to HO in human tendon, we recruited 6 patients with either destructive fracture (*n* = 3) or ossification of the posterior longitudinal ligament (*n* = 3) and respectively collected uninjured tendon and HO samples from these patients. Immunofluorescent staining showed that GLI1⁺ cells were distributed in the tendon sheath and expanded significantly in human HO lesions (Fig. 8A–E). Consistently, both GNAS⁺ cells and p-PKA substrate⁺ cells exhibited significantly elevated densities in HO lesions compared to uninjured tendon controls (Appendix Fig. S10A–D).

Moreover, after inducing osteochondral differentiation for 5 days in CD140a⁺ human tendon multipotent cells (Adrian R et al, 2020), further immunofluorescent imaging showed that GNAS and p-PKA substrate signal were significantly higher than that in vehicle-treated cells in vitro (Fig. 8F–I; Appendix Fig. S10E–G). Moreover, we found that GNAS and p-PKA substrate expression was significantly lower in NF449-treated human tendon stem cells compared to vehicle controls (Fig. 8F–I). In addition, qPCR assay revealed that NF449 significantly inhibited the expression of *SOX9* and *RUNX2* during chondrogenic

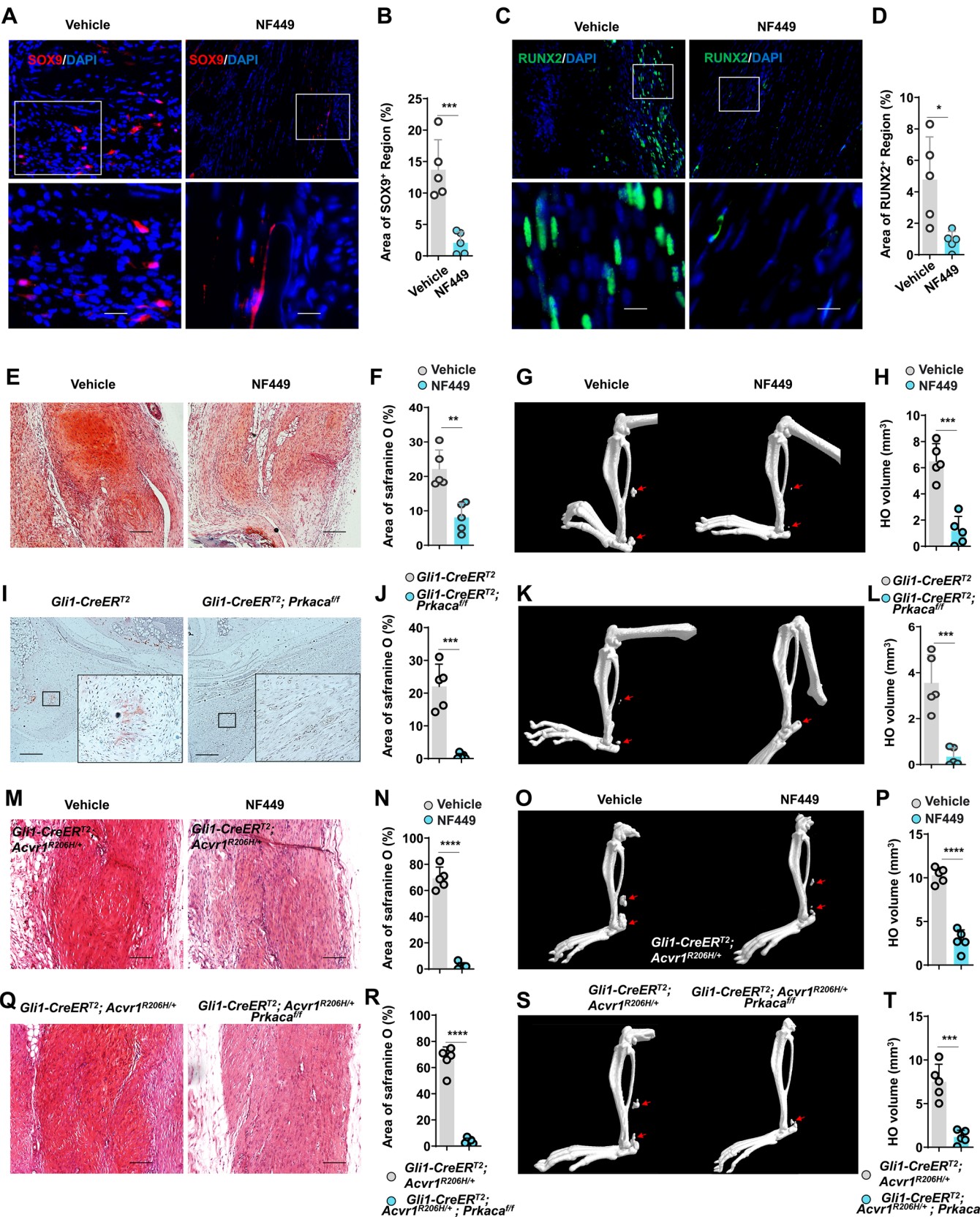

**Figure 7. Inhibition of GNAS/PKA signaling prevents HO formation.**

(A, B) Representative immunofluorescence images (A) and statistical analysis (B) of $SOX9^+$ region in injured site from tenotomized mice following vehicle and NF449 treatment ($n = 5$ per group). ***$P = 8.94$e-4. Scale bar, 10 μm. (C, D) Representative immunofluorescence images (C) and statistical analysis (D) of $RUNX2^+$ region in injured site from tenotomized mice following vehicle and NF449 treatment ($n = 5$ per group). *$P = 1.37$e-2. Scale bar, 10 μm. (E, F) Representative Safranine O staining images (E) and statistical analysis (F) of Safranine $O^+$ region in injured site from tenotomized mice following vehicle and NF449 treatment ($n = 5$ per group). **$P = 1.85$e-3. Scale bar, 100 μm. (G, H) Representative microCT images (G) and statistical analysis (H) of HO volume in injured site from tenotomized mice following vehicle and NF449 treatment ($n = 5$ per group). ***$P = 1.39$e-4. (I, J) Representative Safranine O staining images (I) and statistical analysis (J) of Safranine $O^+$ region in injured site from tenotomized $Gli1\text{-}CreER^{T2}$ and $Gli1\text{-}CreER^{T2}$; $Prkaca^{I/I}$ mice ($n = 5$ per group). ***$P = 1.18$e-4. Scale bar, 100 μm. (K, L) Representative microCT images (K) and statistical analysis (L) of HO volume in injured site from tenotomized $Gli1\text{-}CreER^{T2}$ and $Gli1\text{-}CreER^{T2}$; $Prkaca^{I/I}$ mice ($n = 5$ per group). ***$P = 5.11$e-4. (M, N) Representative Safranine O staining images (M) and statistical analysis (N) of Safranine $O^+$ region in injured site from tenotomized $Gli1\text{-}CreER^{T2}$; $Acvr1^{R206H/+}$ mice following vehicle and NF449 treatment ($n = 5$ per group). **** $P = 1.5$e-7. Scale bar, 100 μm. (O, P) Representative microCT images (O) and statistical analysis (P) of HO volume in injured site from tenotomized $Gli1\text{-}CreER^{T2}$; $Acvr1^{R206H/+}$ mice following vehicle and NF449 treatment ($n = 5$ per group). ****$P = 3.87$e-6. (Q, R) Representative Safranine O staining images (Q) and statistical analysis (R) of Safranine $O^+$ region in injured site from tenotomized $Gli1\text{-}CreER^{T2}$; $Acvr1^{R206H/+}$ and $Gli1\text{-}CreER^{T2}$; $Acvr1^{R206H/+}$; $Prkaca^{I/I}$ mice at 7 dpi ($n = 5$ per group). ****$P = 6.2$e-7. Scale bar, 100 μm. (S, T) Representative microCT images (S) and statistical analysis (T) of HO volume in injured site from tenotomized $Gli1\text{-}CreER^{T2}$; $Acvr1^{R206H/+}$ and $Gli1\text{-}CreER^{T2}$; $Acvr1^{R206H/+}$; $Prkaca^{I/I}$ mice at 14 dpi ($n = 5$ per group). ***$P = 1.79$e-4. Data are presented as mean ± SD. All $P$ values were determined by unpaired Student's $t$ test.

and osteogenic differentiation of $CD140a^+$ human tendon multipotent cells, respectively (Fig. 8J,K), indicating that suppressing Gsα might block human tendon HO. Taken together, these results revealed that GNAS/PKA signaling activation was closely associated with HO formation in human tendon.

## Discussion

In this study, we discovered a subpopulation of $Gli1^+$ tendon sheath progenitors that function as tendon stem/progenitors and characterized their developmental trajectory in HO formation following tenotomy in mice. These $Gli1^+$ sheath progenitors can initially differentiate into normal or osteochondral tendon lineage cells, with the latter subsequently giving rise to chondrocytes and osteocytes in the development of tendon HO. Moreover, we found that the GNAS/PKA signaling pathway is activated during osteochondral differentiation and subsequently confirmed that targeted suppression of GNAS/PKA signaling could effectively reduce the formation of tendon HO.

Tendon sheath, distinct from mid-substance of tendon, consist of vasculature and neuron, could be the niche of tendon stem cells (Harvey et al, 2019). $Tppp3^+/Pdgfra^+$ sheath cells were proved to be a subpopulation of tendon stem cells, which could serve to treat tendon injury. Gli1 marks several populations of stem cells across tissues in mice, highlighting the central role of hedgehog signaling in tissue homeostasis and pathogenesis after injury (Kramann et al, 2015; Yao et al, 2021). Gli1 labels tendon enthesis progenitor (Fang et al, 2022), whereas its contribution to tendon HO remains unclear. We used genetic tracing analysis to reveal that, in addition to tendon enthesis, $Gli1^+$ cells distributed in uninjured tendon sheath. These $Gli1^+$ tendon sheath cell, rather than enthesis cells, rapidly expanded after injury and distributed at each stage following HO development, hinting the potential contribution of $Gli1^+$ tendon sheath progenitors to HO. To further confirm the contribution of $Gli1^+$ tendon sheath progenitors to HO, we generated an FOP model mouse harboring the $Acvr1^{R206H/+}$ mutation in $Gli1^+$ cells ($Gli1\text{-}CreER^{T2}$; $Acvr1^{R206H/+}$). After induction of Cre with tamoxifen, these mice rapidly developed HO after tenotomy injury, robustly evidencing the decisive role of $Gli1^+$ tendon sheath progenitors in tendon HO formation. We also observed that Acvr1 gain-of-function mutation could greatly

accelerate tendon HO formation, suggesting that BMP signaling was also essential to tendon HO formation.

An integrated analysis combining scRNA-seq and genetic lineage tracing analysis subsequently uncovered the developmental trajectory of $Gli1^+$ tendon sheath progenitors after tenotomy injury, including tenogenic and osteochondral progenitors; these progenitor populations retained characteristics of tenocytes, supporting the existence of tissue-specific stem cells. In addition, we observed the potential contribution of $Gli1^-$ progenitor subpopulations to osteochondral lineage differentiation, which aligns with emerging evidence of functional heterogeneity within tendon stem/progenitor cell compartments. $GLI1^-//SOX9^+$ cells numerically dominate their $GLI1^+$ counterparts, our experimental evidence demonstrating that $Gli1^+$ cell ablation reduces HO suggests a dual-pathway regulatory mechanism: (1) Direct depletion of osteochondrogenic $GLI1^+$ progenitors, and (2) Secondary disruption of niche-dependent maintenance in $Gli1^-$ subpopulations through intercellular crosstalk. $Tppp3^+$ tendon sheath progenitors also contribute to HO formation (Yea et al, 2023). We showed that $Gli1^+$ cell depletion resulted in a significant reduction of the total $TPPP3^+$ tendon progenitor population. This was accompanied by corresponding decreases in $SOX9^+$ chondrogenic progenitors, $RUNX2^+$ osteochondrogenic progenitors, $COL2^+$ mature chondrocytes and $OPN^+$ osteoblasts. These findings collectively establish $Gli1^+$ cells as crucial regulators of both cell-autonomous differentiation and non-autonomous progenitor maintenance. However, at the molecular level, these two tendon sheath progenitor subpopulations exhibit both similarities and heterogeneities that warrant future exploration.

We subsequently identified that Gnas was highly expressed during osteochondral differentiation of $Gli1^+$ tendon sheath progenitors. Loss of Gnas promotes osteogenic differentiation of MSC and promotes POH development via activating YAP/Hedgehog signaling pathway (Cong et al, 2021). This bidirectional dysregulation highlights a dual role for Gnas in skeletal biology: while its loss accelerates pathological ossification, intact yet dysregulated GNAS signaling exacerbates aberrant osteochondrogenesis. We hypothesized that this discrepancy could be associated with the distinct transcriptome of tissue-specific stem cells. However, to avoid complete suppression of Gnas expression, we selected NF449 to antagonize Gsα. Indeed, the osteochondral differentiation was inhibited, and no subcutaneous HO was found. To avoid the unexpected osteogenic effect of NF449, we intended to find the downstream targets of Gnas. Gnas-encoding

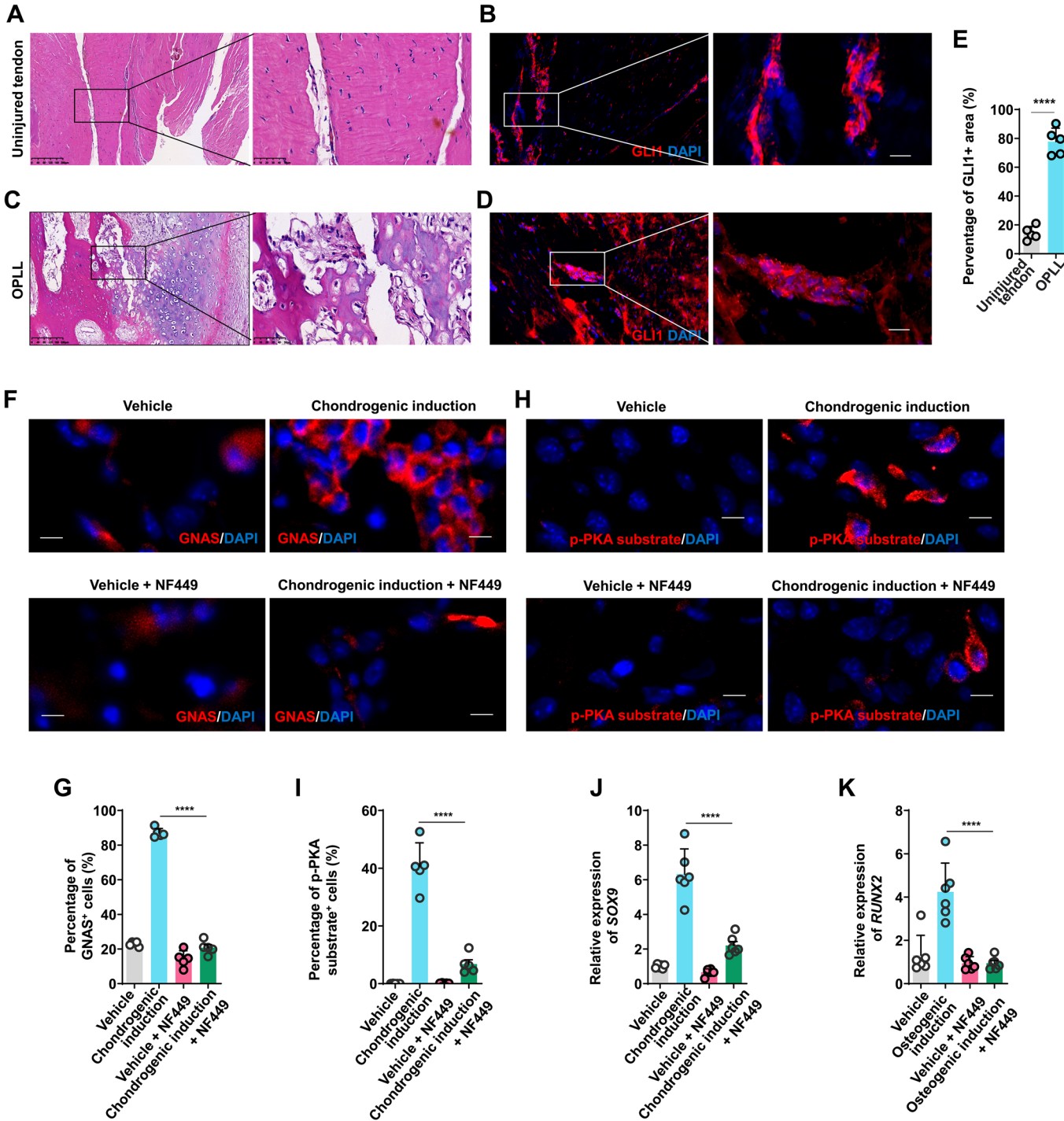

**Figure 8. GLI1, GNAS, and p-PKA substrate expression was higher in human tendon HO.**

(A) Representative HE staining showing the morphology of human tendon. Scale bar, 200 or 50 μm. (B) Representative immunofluorescence staining of GLI1 in human tendon. Scale bar, 10 μm. (C) Representative HE staining showing the morphology of HO from Ossification of the Posterior Longitudinal Ligament patients. Scale bar, 200 or 50 μm. (D) Representative immunofluorescence staining of GLI1 in HO from Ossification of the Posterior Longitudinal Ligament patients. Scale bar, 10 μm. (E) Statistical analysis of the GLI1+ cells in human normal tendon and HO lesion ($n = 3$ per group). ****$P = 7.88$e-7. (F, G) Representative immunofluorescence staining (F) and statistical analysis (G) of GNAS in naive or chondrogenic tendon stem cells with or without NF449 treatment. ($n = 5$ per group). ****$P = 2.8$e-14. Scale bar, 10 μm. (H, I) Representative immunofluorescence staining (H) and statistical analysis (I) of p-PKA substrate in naive or chondrogenic tendon stem cells with or without NF449 treatment. ($n = 5$ per group). Data are represented as the mean ± SD. ****$P = 9.77$e-9. Scale bar, 10 μm. (J, K) Statistical analysis of *SOX9* (J, ****$P = 8.91$e-8) and *RUNX2* (K, ****$P = 6.62$e-6) in chondrogenic or osteogenic tendon stem cells with or without NF449 treatment. ($n = 6$ per group). Data is presented as mean ± SD. All P values were determined by unpaired Student's t test (E) and one-way ANOVA with Bonferroni post hoc test (G, I, J, K).

protein Gsα binds and activates the membrane-bound adenylyl cyclase enzyme, which in turn stimulates the conversion of adenosine triphosphate (ATP) to cyclic adenosine monophosphate (cAMP). cAMP is an important intracellular second messenger that activates different signaling proteins, including protein kinase A, leading to a chain of cellular events that eventually result in various physiological responses. We observed that NF449, as an antagonist of Gsα could inhibit the phosphorylation of PKA and expression of GNAS; this could be caused by positive feedback of GNAS/PKA loop, a mechanism supported by the reduction in HO observed following CREB inhibition. Moreover, we examined the activation of PKA and found that PKA substrate was highly expressed by osteochondral progenitors at 7 dpi. Therefore, targeting PKA through inhibiting PKA activity and expression effectively inhibits tendon HO formation.

In summary, we revealed that *Gli1*+ tendon sheath progenitors are a subpopulation of tendon stem cells that harbors osteochondral differentiation capacity to form HO. Further fate determination tracing analysis of *Gli1*+ tendon sheath progenitors revealed a bifurcate trajectory into tenogenic and osteochondrogenic preference, providing new insights into the fate determination of tendon stem cells during tendon injury. In addition, we unraveled that GNAS/PKA pathway promoted the production of osteochondral progenitors from stem cells, which could be used to treat or prevent HO.

# Methods

### Reagents and tools table

| Reagent/resource | Reference or source | Identifier or catalog number |
|---|---|---|
| **Experimental models** | | |
| *Gli1-CreER^T2* | The Jackson Laboratory | *Gli1tm3(cre/ERT2)Alj/J* |
| SCX-GFP | Brian A et al, 2007 | N/A |
| *Prrx1-Cre* | The Jackson Laboratory | B6.Cg-Tg(Prrx1-cre)1Cjt/J |
| *Tek-Cre* | The Jackson Laboratory | B6.Cg-Tg(Tek-cre)12Flv/J |
| Ai9 | The Jackson Laboratory | B6.Cg-Gt(ROSA)26Sortm9(CAG-tdTomato)Hze/J |
| DTA | The Jackson Laboratory | B6.129P2-Gt(ROSA)26Sortm1(DTA)Lky/J |
| ZsGreen | The Jackson Laboratory | B6.Cg-Gt(ROSA)26Sortm6(CAG-ZsGreen1)Hze/J |
| *Prkaca*-flox | GemPharmatech | T013119 |
| *Acvr1^R206H/+ flox* | This study | N/A |
| **Recombinant DNA** | | |
| N/A | | |
| **Antibodies** | | |
| Anti-human CD140a-APC | Biolegend | Cat#323511 |
| Anti-human CD45-FITC | Biolegend | Cat#982316 |
| Anti-mouse SOX9 | Invitrogen | Cat#702016 |

| Reagent/resource | Reference or source | Identifier or catalog number |
|---|---|---|
| Anti-human RUNX2 | R&D | Cat#AF2006 |
| Anti-mouse Collagen II | Thermo | Cat#PA5-99159 |
| Anti-mouse FMOD | Proteintech | Cat#60108-1-Ig |
| Anti-mouse OCN | Proteintech | Cat#23418-1-AP |
| Anti-mouse GNAS | Proteintech | Cat#10150-2-AP |
| Anti-Phospho-PKA substrate | CST | Cat#9624 |
| Anti-mouse ACAN | Proteintech | Cat#13880-1-AP |
| Anti-mouse TPPP3 | Proteintech | Cat#15057-1-AP |
| Anti-mouse ACVR1 | Proteintech | Cat#67417-1-Ig |
| Anti-mouse p-SMAD1/5 | CST | Cat#9516S |
| Anti-mouse OPN | Proteintech | Cat#22952-1-AP |
| Anti-mouse ZsGreen | FRONTIER INSTITUTE | Cat#MSFR106440 |
| Anti-mouse GLI1 | Abcam | Cat#ab151796 |
| **Oligonucleotides and other sequence-based reagents** | | |
| PCR primers | This study | Table EV1 |
| **Chemicals, enzymes, and other reagents** | | |
| Tamoxifen | Sigma | Cat#T5648 |
| NF449 | MedChemExpress | Cat#HY-112461 |
| 666-15 | MedChemExpress | Cat#HY-101120 |
| TGF-β3 | Sigma | Cat#939250 |
| BMP6 | MedChemExpress | Cat#HY-P700029AF |
| Ascorbate-2-phosphate | Sigma | Cat#49752 |
| Dexamethasone | Sigma | Cat#D4902 |
| β-glycerol phosphate | Sigma | Cat#50020 |
| **Software** | | |
| Loupe Brower 8 | 10X Genomics | 10xgenomics.com |
| TissueFAXS Viewer | TissueGnostics | tissueGnostics.com |
| Prism 9.0 | GraphPad | graphpad.com |
| **Other** | | |

## Animals

*Gli1-CreER^T2* (*Gli1^tm3(cre/ERT2)Alj*/J) mice were provided by Dr. Qingfeng Wu at the Chinese Academy of Sciences (Beijing, China). SCX-GFP mice (Brian A et al, 2007) were provided by Dr. Ronen Schweitzer at Oregon Health & Science University (Portland, USA). *Prrx1-Cre* (B6.Cg-Tg(Prrx1-cre)1Cjt/J), *Tek-Cre* (B6.Cg-Tg(Tek-cre)12Flv/J), Ai9 (B6.Cg-Gt(ROSA)26Sortm9(CAG-tdTomato) Hze/J), DTA (B6.129P2-Gt(ROSA)26Sortm1(DTA)Lky/J) and ZsGreen (B6.Cg-Gt(ROSA)26Sortm6(CAG-ZsGreen1)Hze/J) mice were purchased from The Jackson Laboratory. *Prkaca*-flox mice were purchased from GemPharmatech Co., Ltd (Nanjing, China). All mice were housed in the same animal facility on a 12-hour

reverse light/dark cycle and provided with food and water ad libitum. All procedures and husbandry were performed according to protocols approved by the Institutional Animal Care and Use Committee at Anhui Medical University.

## Generation of *Acvr1*<sup>R206H/+</sup> *flox* mice

Since the *Acvr1*-202(ENSMUST00000090935.9) transcript contains 13 exons, with a translation start site A T G at exon 5 and a translation termination site T G A at exon 13, encoding 509 amino acids, R206H mutation was produced in Exon 8 of *Acvr1*-202. The corresponding 206-position codon is mutated from C G C to C A C, and the amino acid is mutated from arginine (R) to histidine (H). We insert *Loxp*-E7 to E13 coding sequence-*Stop-Loxp*-E7 to E13 Mutant coding sequence-*IRES-ZsGreen polyA* between exon 6 and exon 9 using CRISPR (clustered regularly interspaced short palindromic repeats)/Cas technology. The fragment, with an insert length of about 6.8 kb, was expressed under the regulation of the endogenous *Acvr1* gene. Donor vector repairs the DSB through homologous recombination and obtained positive F0 generation mice. The F0 positive mice were mated with C57BL/6JGpt mice, the pups will be genotyped by PCR, followed by sequence analysis. Owing to the larger insert transcripts, the ZsGreen expression was rarely detected. Subsequent staining by anti-ZsGreen antibody (MSFR106440, FRONTIER INSTITUTE co., Ltd) is used to identify the *Acvr1*<sup>R206H/+</sup> mutant cells in tissue section of FOP model mice. Genotyping PCR primers are shown in the Reagents and Tools Table. Sanger sequencing was performed to confirm the deletion of STOP codons in tendon cells derived from *Gli1-CreER*<sup>T2</sup>; *Acvr1*<sup>R206H/+</sup>; Ai9 and *Acvr1*<sup>R206H/+</sup> mice.

## Human tendon and ossification of the posterior longitudinal ligament (OPLL) sample collection

Three destructive fracture patients and three OPLL patients (both genders were included) were recruited in this study. Human tendons (1 cm-diameter) were derived from the abandoned tissues form destructive fracture patients with amputation surgery (The First Affiliated Hospital of University of Science and Technology of China). Additionally, derelict HO samples from OPLL patients (The First Affiliated Hospital of Anhui Medical University) were collected for GLI1 staining. This study was approved by the ethics committee of The First Affiliated Hospital of University of Science and Technology of China and Anhui Medical University. The informed consent was obtained from all subjects and that the experiments conformed to the principles set out in the WMA Declaration of Helsinki and the Department of Health and Human Services Belmont Report.

## HO induction

Tendon HO model was induced as reported previously (Kan et al, 2024a). In brief, mice were anesthetized with 1% pentobarbital sodium. The skin was incised to expose the Achilles tendon. The Achilles tendon received 20 times of repeated clamping by the hemostatic forceps and then was cut by the scissors. Finally, the skin was closed with sutures.

## Genetic lineage tracing and depletion analysis

*Gli1-CreER*<sup>T2</sup>; Ai9 mice were generated by crossing *Gli1-CreER*<sup>T2</sup> mice and Ai9 reporter mouse to label *Gli1*<sup>+</sup> cells (including stem cells and their progeny). To induce Cre expression, *Gli1-CreER*<sup>T2</sup>; Ai9 mice received three intraperitoneal (i.p.) injections of tamoxifen (3 mg per mouse) administered every other day over a 5-day period. After an additional week, *Gli1-CreER*<sup>T2</sup>; Ai9 mice were tenotomized and sacrificed at 3, 5, 7, 14, 21, and 63 dpi in indicated experiments.

DTA is a mouse line useful for conditional depletion of specific subpopulations of cells. Once bred to mice that express *Gli1-CreER*<sup>T2</sup> recombinase after tamoxifen induction, the loxP and stop sequence are removed, and diphtheria toxin subunit A (DTA) expression is activated, resulting in the ablation of *Gli1-CreER*<sup>+</sup> cells.

## Single-cell RNA sequencing analysis

We applied the Single-cell RNA sequencing data (GSE126060) from Dr Bejamin Levi laboratory for developmental trajectory analysis of tendon stem cells. Also, *Gli1-CreER*<sup>+</sup> and *Gli1-CreER*<sup>-</sup> cells from uninjured and injured tendons of *Gli1-CreER*<sup>T2</sup>; Ai9 mice were isolated for scRNA-seq. The UMI count matrix was then analyzed using Seurat (version 4.0.0) R package. To remove low-quality cells and likely multiplet captures, a set of criteria were conducted: Cells were filtered by (1) gene numbers (gene numbers <200), (2) UMI (UMI < 1000), (3) log10GenesPerUMI (log10GenesPerUMI < 0.7), (4) percentage of mitochondrial RNA UMIs (proportion of UMIs mapped to mitochondrial genes > 5%) and (5) percentage of hemoglobin RNA UMIs (proportion of UMIs mapped to hemoglobin genes >5%). Subsequently, the Doublet-tFinder package (version 2.0.3) was used to identify potential doublets. To obtain the normalized gene expression data, library size normalization was processed using the NormalizeData function. Specifically, the global-scaling normalization method "LogNormalize" normalized the gene expression measurements for each cell by the total expression, multiplied by a scaling factor (10,000 by default), and log-transformed the results.

Top 2000 highly variable genes (HVGs) were calculated using the Seurat function FindVariableGenes (mean.function = FastExp-Mean, dispersion.function = FastLogVMR). Principal-component analysis (PCA) was performed to reduce the dimensionality with RunPCA function. Graph-based clustering was performed to cluster cells according to their gene expression profile with the FindClusters function. Cells were visualized using a 2-dimensional Uniform Manifold Approximation and Projection (UMAP) algorithm with the RunUMAP function. The FindAllMarkers function (test.use = presto) was used to identify marker genes of each cluster. Differentially expressed genes (DEGs) were selected using the function FindMarkers (test.use = presto). $P$ value < 0.05 and |log2foldchange| > 0.58 was set as the threshold for significantly differential expression. GO enrichment and KEGG pathway enrichment analysis of DEGs were, respectively, performed using R (version 4.0.3) based on the hypergeometric distribution.

## Pseudotime analysis

The developmental pseudotime of tendon cells from GSE126060, including *Gli1-CreER*<sup>+</sup>, and *Gli1-CreER*<sup>-</sup> cells, was reconstructed using Monocle2 (version 2.9.0), Slingshot, and RNA velocity analysis. For Monocle 2 analysis, the raw count was first converted from Seurat object into CellDataSet object with the importCDS

function in Monocle. The differentialGeneTest function of the Monocle2 package was used to select ordering genes (qval <0.01) which were likely to be informative in the ordering of cells along the pseudotime trajectory. The dimensional reduction clustering analysis was performed with the reduceDimension function, followed by trajectory inference with the orderCells function using default parameters. Gene expression was plotted with the plot_genes_in_pseudotime function to track changes over pseudotime. Spliced/unspliced counts were quantified with velocyto. High-dimensional RNA velocity vectors were inferred using the dynamical model in scVelo, accounting for cluster-specific kinetics and adjusting for latent time. Additionally, Post-clustering (Seurat), lineage trajectories were modeled using Slingshot. The minimum spanning tree defined branch points from user-specified start clusters (e.g., progenitor states), and pseudotime was computed along principal curves.

## Immunofluorescence and immunohistochemistry

Immunostaining for different markers was performed as previously described (Kan et al, 2024a). Briefly, sections were prefixed with 4% paraformaldehyde in PBS. Nonspecific binding was blocked by incubation for 1 h at room temperature with 10% goat serum (C2065, Beyotime, China) and 0.25% Triton X-100 (9036-19-5, Sigma, USA). The sections were then incubated with primary antibodies diluted with 1% BSA + 0.25% Triton X-100 at 4 °C overnight. After washing, the sections were incubated with the appropriate secondary antibodies (Alexa Fluor 488- and Alexa Fluor 594-conjugated antibodies that were diluted with 1% BSA + 0.25% Triton X-100) in the dark at room temperature for 2 h; counterstaining was performed with 4,6-diamidino-2-phenylindole (1:4000). All fluorescently stained images were captured using a ZEISS Axio Observer (Carl Zeiss, Germany). Signals from all channels were collected individually and overlaid in Adobe Photoshop. Detailed data for all antibodies in this study are shown in the Reagents and Tools Table.

## MicroCT

Mouse hindlimbs were harvested and imaged after injury at different time points. To quantitatively measure the HO volume and bone parameters, microCT (Skyscan 1176, Bruker) was used with the setting parameters of 180° rotation, constant 90 kV voltage, and a voxel size of 72 μm. 3D images were reconstructed with SkyScan software. The HO region was first outlined by the ROI module and then quantified by individual 3D object analysis.

## Histology analysis

Hindlimbs from uninjured and tenotomized WT and FOP mice were harvested at designated time points and processed for specified staining. Briefly, hindlimbs were fixed at 4 °C in 4% paraformaldehyde. Samples were then decalcified in 20% (m/v) EDTA solution for 4–6 weeks at 4 °C until deformable manually. Bones were paraffin-embedded, and 10 μm sections were cut and stored at −20 °C. Paraffin sections were selected for dewaxing and then HE staining (HE staining Kit, G1120, Solarbio), and Safranine O staining (G1371, Solarbio, China) assays were performed according to the manufacturer's instructions.

## Treatment with antagonist of Gsα and CREB inhibitors

WT and FOP mice (C57BL/6 background) that underwent tenotomy surgery were immediately treated with vehicle (saline), NF449 (MCE, 50 mg/kg), or 666-15 (MCE, 20 mg/kg) through *i.p.* injection every other day for 1, 2, or 4 weeks in the indicated experiments.

## Human tendon stem/progenitor cell isolation

Fresh human tendons were cut into pieces. MACS separation columns (Miltenyi Biotec Inc.) were used to harvest CD140a$^+$/CD45$^-$ cells into the cell incubator. These cells were observed after 3 days of incubation without any disturbance. After 7 days of culture, these tendon stem cells showed confluent growth and then were considered for osteochondral differentiation assay.

## RNA extraction and quantitative PCR analysis

An RNAiso Plus kit (Takara) was used according to the manufacturer's instructions for the extraction of total RNA from MSCs. In addition, a PrimeScript RT reagent kit (Takara) was used to synthesize first-strand cDNA. The expression of various genes was quantified by real-time PCR mixture assays (Takara). Human *GAPDH* was used as internal control. All primer sequences and product sizes are listed in the Reagents and Tools Table.

## Osteochondral differentiation assay of tendon stem cells

Osteochondral differentiation of stem cells was reported previously (Kan et al, 2024b). For osteochondral differentiation, tendon stem cells were counted and seeded in a 24-well plate with osteogenic medium (α-MEM supplemented with 10% (vol/vol) FBS, $10^{-7}$ M (0.1 μM) dexamethasone, 10 mM β-glycerol phosphate and 50 μM ascorbate-2-phosphate) or chondrogenic medium (α-MEM supplemented with 10% (vol/vol) FBS, TGF-β3 (10 ng/ml), and BMP6 (500 ng/ml)). The medium was changed every three days, and the cells were maintained in culture for 2–4 weeks. Immunofluorescence staining with indicated antibodies and qPCR assay were used to examine the capacity of differentiation.

## Study approval

All the mice in the experiments of this study were approved by the Animal Care and Use Committees at Anhui Medical University (Protocol: LLSC20210972 and LLSC20240938). Additionally, the experiments about HO samples were approved by the Biomedical Ethics Committee of Anhui Medical University (Protocol: 83240167) and experiments associated with human tendon were approved by the ethics committee of Anhui Provincial Hospital, The First Affiliated Hospital of University of Science and Technology of China (Protocol: 2024-BE(H)-128).

## Statistics

All experiments were repeated at least three times. Data are reported as the mean ± SD. Statistical analyses between two groups were performed using unpaired two-tailed Student's *t* tests. One-way ANOVA is applied into multiple comparisons with a single

factor. All statistical analyses were performed using Prism 8 (GraphPad) software. $*P < 0.05$, $**P < 0.01$, $***P < 0.001$ or $****P < 0.0001$ was considered statistically significant in this study.

## Graphics

Graphics were created with BioRender.com (some of the Fig. 1A or synopsis).

## Data availability

The data that support the findings of this study are openly available in GEO database, reference number 126060 and 298748. Source data have been deposited in BioStudies under accession number S-BSST2091.

The source data of this paper are collected in the following database record: biostudies:S-SCDT-10_1038-S44318-025-00553-7.

## Peer review information

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

## Acknowledgements

We are grateful to Drs. Xiaoling Zhang (Shanghai Jiao Tong University, China) and Xiao Chen (Zhejiang University, China) for their assistance with transferring the SCX-GFP mice. This work was supported by the National Natural Science Foundation of China (82102573 and 82472531), the Postdoctoral Fellowship Program of CPSF under Grant Number GZC20240010, the China Postdoctoral Science Foundation - Anhui Joint Support Program under Grant Number 2024T026AH, the Young Science and Technology Talent Lifting Program of Anhui Provincial Association for Science and Technology (RCTJ202428), Anhui Provincial Postdoctoral Research Program Funding (2025A1038), the Basic and Clinical Cooperative Research Incubation Project of Anhui Medical University for The Third Affiliated Hospital (2023sfy016), the Key Project of Ningbo Municipal Natural Science Foundation (2024J031), Anhui Provincial Key Research Program in Natural Sciences for Higher Education Institutions (2023AH053407) and the Natural Science Foundation of Anhui Province (2408085MH197).

## Author contributions

**Lijun Chen**: Conceptualization; Data curation; Supervision; Funding acquisition; Investigation; Methodology; Writing—original draft; Project administration; Writing—review and editing. **Chao Peng**: Data curation; Investigation; Methodology. **Lanyi Chai**: Data curation; Investigation. **Renjie Zhang**: Resources; Investigation; Methodology. **Chenghang Zhu**: Resources; Investigation; Methodology. **Hailin Wang**: Investigation; Methodology. **Qirong Cheng**: Investigation; Methodology. **Yan Yan**: Investigation; Methodology. **Cailiang Shen**: Investigation; Methodology; Project administration; Writing—review and editing. **Hong Zheng**: Conceptualization; Methodology; Project administration; Writing—review and editing. **Jiazhao Yang**: Conceptualization; Funding acquisition; Methodology; Project administration; Writing—review and editing. **Haitao Fan**: Funding acquisition; Methodology; Project administration; Writing—review and editing. **Chen Kan**: Conceptualization; Data curation; Supervision; Funding acquisition; Writing—original draft; Project administration; Writing—review and editing.

Source data underlying figure panels in this paper may have individual authorship assigned. Where available, figure panel/source data authorship is listed in the following database record: biostudies:S-SCDT-10_1038-S44318-025-00553-7.

## Disclosure and competing interests statement

The authors declare no competing interests.

# Expanded View Figures

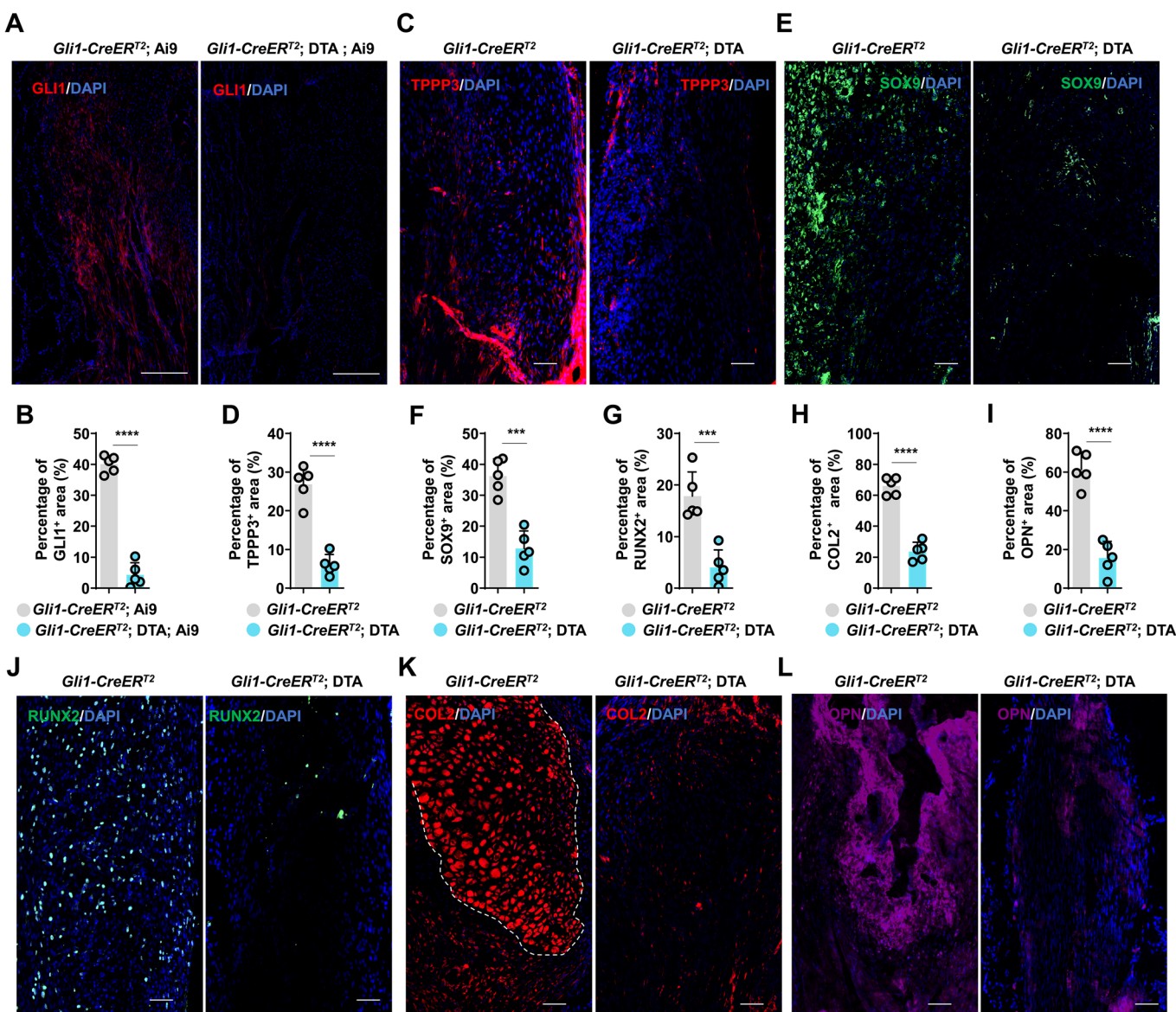

**Figure EV1.** *Gli1*[+] **cell depletion significantly inhibited tendon stem/progenitor population and associated osteochondral differentiation.**

(A, B) Immunofluorescence staining and statistical analysis of the GLI1[+] cells in injured tendons of *Gli1-CreER*[T2]; Ai9 and *Gli1-CreER*[T2]; Ai9; DTA mice at 21 dpi (*n* = 5 per group). ****P = 1.74e-7. Scale bar, 200 µm. (C, D) Immunofluorescence staining and statistical analysis of the TPPP3[+] cells in injured tendons of *Gli1-CreER*[T2] and *Gli1-CreER*[T2]; DTA mice at 7 dpi (*n* = 5 per group). ****P = 2.52e-5. Scale bar, 200 µm. (E, F) Immunofluorescence staining and statistical analysis of the SOX9[+] cells in injured tendons of *Gli1-CreER*[T2] and *Gli1-CreER*[T2]; DTA mice at 7 dpi (*n* = 5 per group). ***P = 1.70e-4. Scale bar, 200 µm. (G–I) Statistical analysis of the RUNX2[+] (G), COL2[+] (H) and OPN[+] (I) cells in injured tendons of *Gli1-CreER*[T2] and *Gli1-CreER*[T2]; DTA mice at 7 dpi, 21 dpi and 63 dpi, respectively. (*n* = 5 per group). *P* values from left to right: ***P = 7.22e-4, ****P = 3.24e-6, ****P = 3.54e-5. (J–L) Immunofluorescence staining of the RUNX2[+] (J), COL2[+] (K) and OPN[+] (L) cells in injured tendons of *Gli1-CreER*[T2] and *Gli1-CreER*[T2]; DTA mice at 7 dpi, 21 dpi and 63 dpi, respectively. Scale bar, 200 µm. Data is presented as mean ± SD. All *P* values were determined by unpaired Student's *t* test.

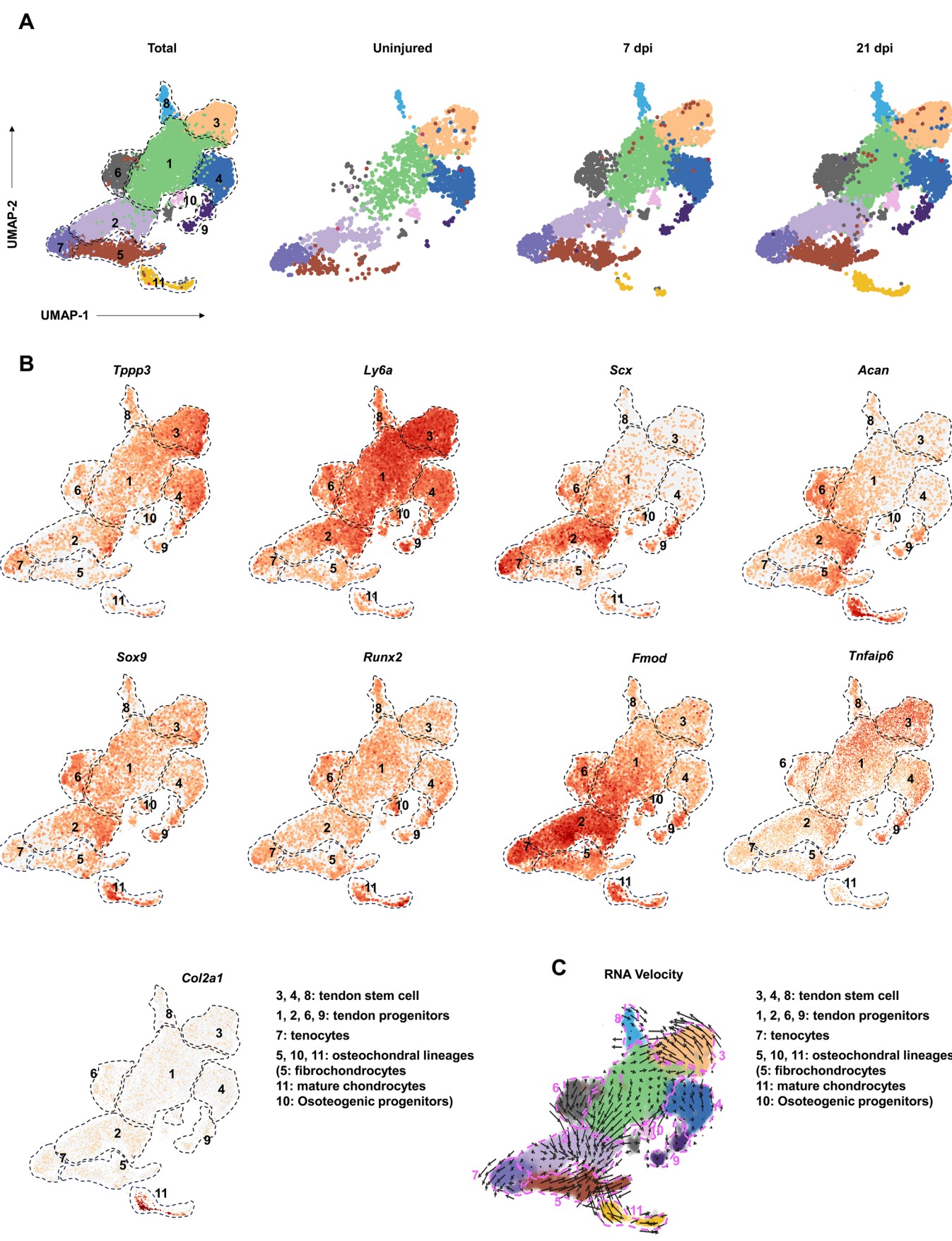

**A**
Total | Uninjured | 7 dpi | 21 dpi

**B**
*Tppp3* | *Ly6a* | *Scx* | *Acan*

*Sox9* | *Runx2* | *Fmod* | *Tnfaip6*

*Col2a1*

3, 4, 8: tendon stem cell
1, 2, 6, 9: tendon progenitors
7: tenocytes
5, 10, 11: osteochondral lineages
(5: fibrochondrocytes
11: mature chondrocytes
10: Osoteogenic progenitors)

**C** RNA Velocity

3, 4, 8: tendon stem cell
1, 2, 6, 9: tendon progenitors
7: tenocytes
5, 10, 11: osteochondral lineages
(5: fibrochondrocytes
11: mature chondrocytes
10: Osoteogenic progenitors)

◀ **Figure EV2.** *Gli1⁺* **tendon sheath progenitors exhibited a multipotent capacity into tenogenic and osteochondrogenic lineages.**

(A) UMAP visualization of *Gli1*⁺ cells at uninjured and different time points post injury (i.e., 7 and 21 dpi). (B) Feature plot images showed the expression of recognized markers for tendon stem cells, progenitors and terminating tenocytes and chondrocytes. (C) RNA velocity showed the tenogenic and osteochondrogenic differentiation trajectory of *Gli1*⁺ tendon sheath progenitors.

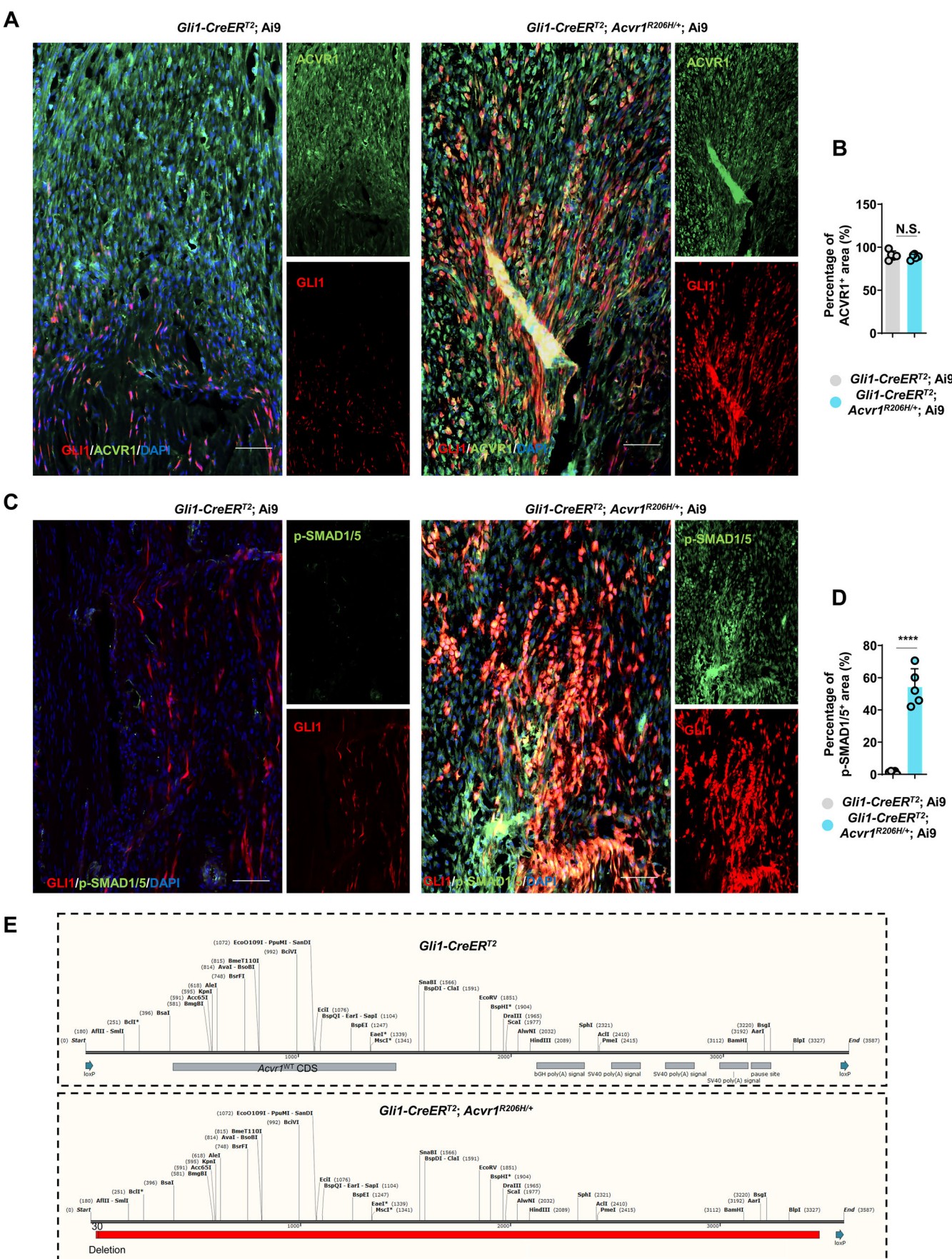

◄ **Figure EV3.  Validation of *Acvr1* expression and *Acvr1*^R206H/+^ mutation following *Gli1-CreER*^T2^ induction.**

(**A**, **B**) Representative immunofluorescence and statistical analysis of GLI1 and ACVR1 in injured site of *Gli1-CreER*^T2^; Ai9 and *Gli1-CreER*^T2^; Ai9; *Acvr1*^R206H/+^ mice at 5 dpi ($n = 5$ per group). $P = 5.18\text{e-}1$. N.S. indicated no significance. Scale bar, 100:μm. (**C**, **D**) Representative immunofluorescence and statistical analysis of GLI1 and p-SMAD1/5 in injured site of *Gli1-CreER*^T2^; Ai9 and *Gli1-CreER*^T2^; Ai9; *Acvr1*^R206H/+^ mice at 5 dpi ($n = 5$ per group). ****$P = 6.89\text{e-}6$. Scale bar, 100 μm. (**E**) Sanger sequencing showed that the STOP cassette preceding the *Acvr1*^R206H/+^ mutation was disrupted. Data is presented as mean ± SD. All *P* values were determined by unpaired Student's *t* test.

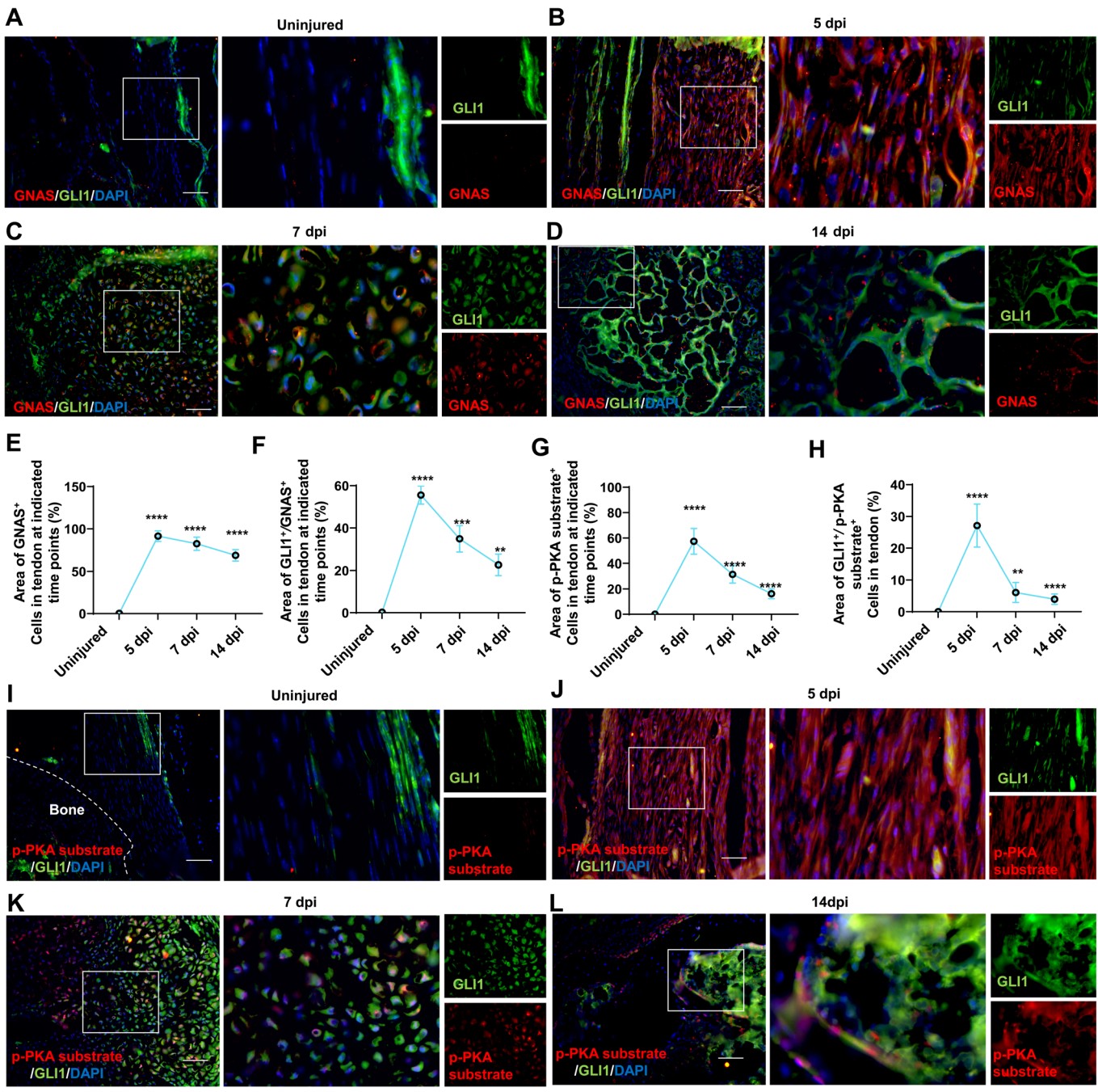

**Figure EV4. GNAS/PKA signaling pathway was activated in osteochondral tendon progenitors of FOP mice.**

(A–D) Representative immunofluorescence images of GNAS expression in uninjured tendons (A) and injured sites of *Gli1-CreER^T2^*; Ai9 mice at 5 (B), 7 (C) and 14 (D) dpi. Scale bar, 50 μm. (E, F) Statistical analysis of GNAS$^+$ region (E) and GNAS$^+$/GLI1$^+$ region (F) in uninjured tendons and injured sites of *Gli1-CreER^T2^*; *Acvr1^R206H/+^* mice at 5, 7 and 14 dpi (n = 5 per group). P values from left to right: ****P = 8.29e-10, ****P = 1.16e-8, ****P = 1.57e-8. ****P = 2.36e-9, ****P = 1.71e-6, ****P = 9.06e-6. (G, H) Statistical analysis of p-PKA substrate$^+$ region (G) and p-PKA substrate$^+$/GLI1$^+$ region (H) in uninjured tendons and injured sites of *Gli1-CreER^T2^*; *Acvr1^R206H/+^* mice at 5, 7 and 14 dpi (n = 5 per group). P values from left to right: ****P = 1.42e-6, ****P = 7.23e-6, ****P = 1.53e-5. ****P = 1.96e-5, **P = 2.63e-3, ****P = 6.44e-4. (I–L) Representative immunofluorescence images of p-PKA substrate expression in uninjured tendons (I) and injured sites of *Gli1-CreER^T2^*; *Acvr1^R206H/+^* mice at 5 (J), 7 (K) and 14 (L) dpi. Scale bar, 50 μm. Data is presented as mean ± SD. All P values were determined by unpaired Student's *t* test.

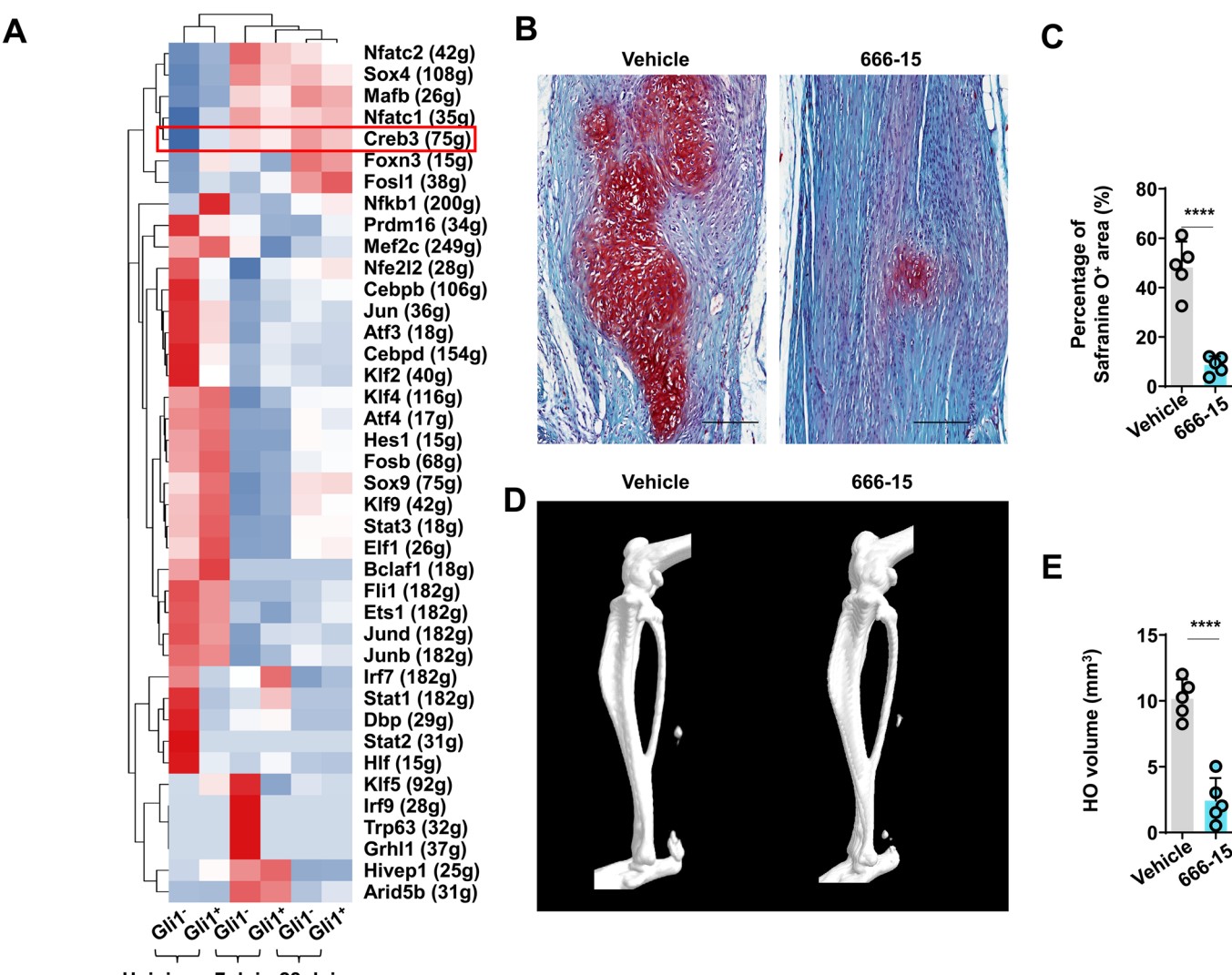

**Figure EV5. 666-15, a CREB inhibitor, prevents tendon HO formation.**

(**A**) Regulon analysis of transcription factors involved in the osteochondrogenic differentiation of *Gli1*+ and *Gli1*+ cells. (**B, C**) Representative safranine O staining (**B**) and statistical analysis (**C**) of chondrocytes in injured site of tenotomized mice treated with vehicle or 666-15 treatment at 21 dpi (*n* = 5 per group). **P = 4.50e-5. Scale bar, 100 μm. (**D, E**) Representative microCT (**D**) and statistical analysis (**E**) of HO volume in injured site of tenotomized mice treated with vehicle or 666-15 treatment at 63 dpi (*n* = 5 per group). **P = 5.93e-5. Data is presented as mean ± SD. All *P* values were determined by unpaired Student's *t* test.

