## [Peer Review File · The EMBO Journal]

GNAS/PKA signaling promotes aberrant osteochondral differentiation of *Gli1*⁺ tendon sheath progenitors

Lijun Chen, Chao Peng, Lanyi Chai, Renjie Zhang, Chenghang Zhu, Hailin Wang, Qirong Cheng, Yan Yan, Cailiang Shen, Hong Zheng, Jiazhao Yang, Haitao Fan, and Chen Kan

Corresponding authors: Chen Kan (chenkan@ahmu.edu.cn), Jiazhao Yang (yangjiazhao@ustc.edu.cn), Haitao Fan (fyyfanhaitao@nbu.edu.cn)

Review Timeline:

Submission Date:	18th Dec 24
Editorial Decision:	11th Jan 25
Appeal:	12th Jan 25
Editor's Correspondence:	24th Jan 25
Editorial Decision:	4th Mar 25
Revision Received:	17th Jun 25
Editorial Decision:	18th Jul 25
Revision Received:	23rd Jul 25
Accepted:	14th Aug 25

Editor: Daniel Klimmeck

Transaction Report:

Dear Dr Kan,

Thank you for submitting your manuscript (EMBOJ-2024-119973-T) to The EMBO Journal. My apologies for the unusual protraction due to the time of the year and detailed discussion in the editorial team. We have considered your study carefully. I am sorry to say that in light of all information at hand we concluded that we cannot offer publication in The EMBO Journal.

We appreciate your results on identification of Achilles heel Gli1+ multipotent progenitors with distinct differentiation potential, and specification of GNAS signaling as targetable pathway to prevent aberrant ossification in this context. At the same time, we noted that Gli1+ cells were demonstrated to be essential for osteochondral differentiation by earlier work of your team and others. Also, GNAs depletion connected to bone heteroplasia and heterotopic ossification. Overall, I am afraid we feel the advance provided in this study is not sufficient for what we have to request for our venue. We have thus concluded not to send your manuscript out for peer-review.

I regret to disappoint you on this occasion and much hope that you will take advantage of our transfer option. I again apologise for the delay.

Kind regards,

Daniel Klimmeck

Daniel Klimmeck, PhD
Senior Editor
The EMBO Journal

** As a service to authors, EMBO Press provides authors with the possibility to transfer a manuscript that one journal cannot offer to publish to another EMBO publication or the open access journal Life Science Alliance launched in partnership between EMBO Press, Rockefeller University Press and Cold Spring Harbor Laboratory Press. The full manuscript and if applicable, reviewers' reports, are automatically sent to the receiving journal to allow for fast handling and a prompt decision on your manuscript. For more details of this service, and to transfer your manuscript please click on Link Not Available. **

Dear Dr. Daniel Klimmeck,

Thanks for the opportunity of considering our manuscript entitled "**GNAS/PKA pathway promotes aberrant osteochondral differentiation of *Gli1*⁺ tendon sheath progenitors (Manuscript EMBOJ-2024-119973-T).**"

On behalf of my co-authors, I would like to lodge an appeal against the rejection of our manuscript.

We don't mean to offend you; we just want to highlight the novelty of our research and hope this study can be reconsidered.

As for reason 1 of rejection, "we noted that *Gli1*⁺ cells were demonstrated to be essential for osteochondral differentiation by earlier work of your team and others", we think this study is not only to show the capacity of *Gli1*⁺ cells. Instead, we demonstrate the identity of *Gli1*⁺ cells, which is a subpopulation of tendon sheath stem/progenitors; this is the first time that this concept has been proposed. As a kind of tendon sheath stem/progenitors, they can undergo aberrant osteochondral differentiation and formed HO, which is most of the consequence of tendon injury.

Indeed, we previously showed that perivascular *Gli1*⁺ cells contribute to muscle HO formation. However, these populations were extremely different from tendon *Gli1*⁺ cells. Moreover, our recent study showed that perivascular *Gli1*⁺ cells is not sufficient to drive muscle HO formation, upon the mutations of *Acvr1R206H/+*, although they have the capacity of osteochondral differentiation.

Additionally, regarding the reason "GNAs depletion connected to bone heteroplasia and heterotopic ossification", we think GNAS deletion indeed led to heterotopic intramembranous bone formation. However, it is completely different from heterotopic endochondral bone formation. We found that *Gnas* is significantly higher during aberrant osteochondral differentiation. So, it is very interesting to explore a different function for GNAS. We also proved that GNAS could lead to PKA activation and consequently promote osteochondral differentiation of *Gli1*⁺ cells. This pathway is also novel for fate determination of tendon stem cells. More importantly, we revealed that PKA could be a therapeutic target for HO treatment.

Overall, we think this paper is a strong candidate for the EMBO Journal, including the quality and reliability of the data; this is also the high standard of the EMBO Journal.

Again, we thank you for the opportunity to appeal. We hope this manuscript can be reconsidered for a chance to review.

Looking forward to your reply.

Sincerely,

Chen

Dear Dr Kan,

Thank you for your note and following up on our recent decision on your manuscript EMBOJ-2024-119973. I have now considered your rebuttal points and also discussed them in the editorial team. We have concluded to still give this work the chance to be evaluated in full depth by external peer-review. Please note however that in light of the expressed reservations, this is a threshold send-out in our view and we will need substantial support by the referees to proceed with the study for EMBO Journal.

I will let you know as soon as we hear from the experts.

Kind regards,

Daniel Klimmeck

Daniel Klimmeck, PhD
Senior Editor | The EMBO Journal
d.klimmeck@embojournal.org

Preliminary point-by-point response

Referee #1:

In this study, Chen et al. conducted an in-depth investigation into the regulatory mechanisms of heterotopic ossification (HO) following tendon injury. By combining genetic lineage tracing and single-cell RNA sequencing (scRNA-seq) analyses, they identified *Gli1*⁺ tendon sheath cells as a key initiator of osteochondral differentiation after injury. Furthermore, they demonstrated that GNAS/PKA signaling is activated in *Gli1*⁺ tendon sheath cells, and that pharmacological or genetic blockade of this pathway significantly ameliorated HO after tenotomy. Overall, this is an interesting study that utilized a series of genetic models to convincingly prove their hypothesis. The human data further emphasize the translational implication of this study. However, a few concerns need to be addressed in a potential revision.

Our response: We thank Referee #1 for the positive comments regarding our manuscript.

Major Points:

1. The trajectory analysis of scRNA-seq data used to predict the downstream differentiation trajectory of *Gli1*⁺ cells is not robust enough. Pseudotime analysis by Monocle2 implies no directionality. RNA Velocity or SlingShot analyses should be a better choice.

Our response: We thank the reviewer for this suggestion. In accordance with this recommendation, we will conduct SlingShot trajectory analysis to investigate the potential developmental relationships among clusters 1, 2, 4, 5, 6, 7, 8, 9, 11, 15, and 16. The results of this analysis will be systematically incorporated into the revised manuscript.

2. Lineage tracing of *Gli1*⁺ cells during development and postnatal growth of tendon in uninjured mice should be added, in order to demonstrate that *Gli1*⁺ cells function as tendon stem/progenitor cells under steady state.

Our response: We sincerely appreciate the reviewer's insightful suggestion regarding lineage tracing of *Gli1*⁺ cells during tendon development. To comprehensively address this important point, we have established a dual transgenic model (*Gli1-CreER*^{T2}; Ai9) and administered tamoxifen via intraperitoneal injection following established protocols for lineage tracing. Tendon samples will be collected at key developmental stages (E14.5, P0, P3, P7) for systematic analysis.

Through triple-label immunofluorescence co-staining of DAPI, FMOD (tenocyte marker), and tdTomato reporter signal, we will rigorously characterize the spatiotemporal distribution patterns of *Gli1*⁺ cells and quantitative correlation between *Gli1*⁺ cell dynamics and tenocyte maturation. These data will be presented in Supplementary Figure with supporting quantification analyses, providing definitive evidence that *Gli1*⁺ cells function as bona fide tendon stem/progenitor cells under

homeostatic conditions.

3. Gli1-expressing cells in the scRNA-seq analyses are not equivalent to Gli1-CreER⁺ cells in the lineage tracing study. Gli1-CreER⁺ cells may down-regulate Gli1 expression upon their activation and differentiation. Therefore, instead of merely analyzing a published dataset, it would be much more informative to perform scRNA-seq on Gli1-CreER-traced cells at different time points post tenotomy to prove their tenogenic and osteochondral differentiation capacities.

Our response: We agree with Referee #1 that *Gli1*-expressing cells in the scRNA-seq analyses are not equivalent to *Gli1-CreER*^{T2+} cells. To rigorously delineate the temporal dynamics of *Gli1*⁺ progenitor cell contributions to heterotopic ossification, we have established a controlled longitudinal cohort comprising *Gli1-CreER*^{T2}; Ai9 reporter mice with tamoxifen induction (n=24, equal sex distribution), stratified into three experimental groups: uninjured controls (n=8) and tenotomy-injured cohorts at 7-, and 21-days post-injury (n=8/group). These samples had been collected and sent out for single-cell RNA-sequencing. Raw sequencing data are being processed through the Cell Ranger pipeline within 7 working days. Upon receipt of the preliminary data, we will promptly initiate comprehensive analyses, such as cell type identification among *Gli1*⁺ cells and pseudotemporal analysis (Monocle3, Slingshot and RNA velocity). We anticipate finalizing these computationally intensive workflows within the 8 weeks, ensuring robust resolution of *Gli1*⁺ lineage plasticity across osteochondrogenic transitions.

4. The Rainbow mice experiment is not convincing. The authors only observed some clonal clusters without staining chondrogenic or osteogenic markers to prove that osteochondral differentiation was clonally derived in vivo. To prove the self-renewal capacity of Gli1⁺ cells, primary and secondary ectopic transplantation should be performed. Alternatively, the author could simply delete this part to avoid confusion or further argument.

Our response: We sincerely appreciate the reviewer's guidance regarding the Rainbow mice experiment. After careful reconsideration of the experimental limitations raised, we fully agree that removing this section will enhance the manuscript's conceptual focus and methodological rigor. Accordingly, we will delete the original **Supplementary Fig. 4** and associated text from both the main text and Supplementary Figures. Additionally, we will temper the identity of *Gli1*⁺ sheath cells in the section of Discussion.

5. It is difficult to identify GLI1⁺/RUNX2⁺ cells in Fig. 2m, and GLI1⁺/SCX⁺/RUNX2⁺ cells in Fig. 3n. Split-channel images are necessary for clarification. Additionally, the proportion of GLI1⁻/SCX⁺/SOX9⁺ cells is significantly higher than GLI1⁺/SCX⁺/SOX9⁺ cells in the injured region, suggesting alternative cellular sources for SOX9⁺ chondrocytes. A similar result is found in RUNX2⁺ osteoblasts.

Our response: We thank reviewer for this suggestion. We will update the

immunostaining images of GLI1/RUNX2 and GLI1/SCX/RUNX2. The Split-channel images will also be added.

Additionally, we fully concur with the reviewer's astute observation regarding the potential contribution of *Gli1*⁻ progenitor subpopulations to osteochondral lineage differentiation, which aligns with emerging evidence of functional heterogeneity within tendon stem/progenitor cell compartments. Building upon our experimental findings that *Gli1*⁺ cell ablation reduces heterotopic ossification, we propose a dual-pathway mechanism to explain this phenotype: 1) Direct depletion of osteochondrogenically primed *Gli1*⁺ progenitors, and 2) Secondary disruption of niche-dependent maintenance in *Gli1*⁻ subpopulations through intercellular crosstalk. To systematically investigate this hierarchical relationship, we are currently performing immunofluorescence analysis of established tendon progenitor markers (TPPP3/SCA1) in both *Gli1-CreER*^{T2} and *Gli1-CreER*^{T2}; DTA models, with particular focus on spatial distribution patterns and proliferation indices of GLI1⁻/TPPP3⁺/SCA1⁺ cells within injury-induced microenvironments. These data will be integrated with scRNA-seq of *Gli1*⁺ and *Gli1*⁻ cells from *Gli1-CreER*^{T2}; Ai9 mice to examine the contribution of *Gli1*⁺ and *Gli1*⁻ cells to HO.

6. The conclusion that "GLI1, GNAS, and p-PKA substrate expression is higher in human tendon HO" lacks convincing evidence. Images of Fig. 8b and 8d show that GLI1 expression levels are similar in normal tendon sheath and HO lesions. Moreover, the manuscript does not clarify the in-situ expression patterns of GNAS and phospho-PKA substrates in uninjured versus HO regions. Is there existing literature supporting the use of CD140a⁺/CD45⁻ as markers for human tendon stem cells? Furthermore, it is unclear whether GNAS and phospho-PKA substrates are activated during the osteogenic differentiation of human tendon stem cells.

Our Response: We sincerely appreciate the reviewer's rigorous evaluation of our human HO analyses. While immunofluorescence intensity quantification shows comparable GLI1 levels between normal tendon sheath and HO lesions, spatial distribution analysis reveals significant expansion of GLI1⁺ cells in HO. We will supplement the statistical analysis for GLI1⁺ area and revise the conclusion to: "GLI1⁺ cells expand significantly in human HO lesions, recapitulating the higher expression of GLI1."

Additionally, we will supplement the immunofluorescence staining assay for GNAS and phospho-PKA substrate in normal tendon and HO lesions, supporting that GNAS/PKA pathway is activated in human HO.

Regarding human tendon stem cell markers, we chose CD140a (*Pdgfra*) according to the scRNA-seq analysis of human tendon (see Adrian R. Kendal et al., (2020) Multi-omic single cell analysis resolves novel stromal cell populations in healthy and diseased human tendon. *Sci Rep* 10, 13939). CD140a⁺ tendon cells consist of

multipotent progenitors, which can differentiate into tenocytes, fibroblast and osteochondrogenic lineages. We will cite this paper when we mention the markers of human tendon stem cells and update the human stem cells into human tendon multipotent cells. Also, we will examine the expression of GNAS and phospho-PKA substrates in CD140a⁺ human tendon multipotent cells with or without induction of osteochondrogenic differentiation.

Minor points:

1. In Fig. 1b, H&E staining should be performed to provide a comprehensive view of specific regions. Additionally, the magnified image at 21 dpi does not align with the statistical data showing a continuous increase in Gli1⁺ cells. Moreover, it remains unclear whether the GLI1 signal represents endogenous expression or lineage tracing results.

Our Response: We thank the reviewer for this suggestion. We will supplement HE staining for normal and injured tendon at different post-injury time points. Also, we will update the magnified images of tendon sheath at 21 dpi. Moreover, we will supplement the immunofluorescence staining of GLI1 in tendon sections from *Gli1-CreER^{T2}*; Ai9 mice with or without tenotomy injury.

2. In Fig. 1h-j, alongside bone mass analysis, the authors should quantify changes in Gli1⁺ cells in both WT and DTA mice to confirm that the improvement in HO is indeed due to Gli1⁺ cell ablation.

Our Response: We will supplement immunostaining of injured tendon section from WT and DTA mice to confirm the availability of DTA mice by examining the endogenous GFP signal. Additionally, we will examine the endogenous GFP signal in injured tendon section from DTA and *Gli1-CreER^{T2}*; DTA mice with tenotomy injury for 21 days and 63 days to prove that decreased HO is due to *Gli1⁺* cell ablation.

3. The authors claim that inhibiting GNAS/PKA signaling prevents tendon HO by impairing osteochondral differentiation in Gli1⁺ tendon sheath cells. However, they do not provide data on how Gli1⁺ SOX9⁺ and Gli1⁺ RUNX2⁺ cell populations change following NF449 treatment or *Prkaca* knockout.

Our Response: We will supplement the immunostaining of SOX9 and RUNX2 in the injured tendon section from tenotomized *Gli1-CreER^{T2}*; Ai9 mice with vehicle or NF449 treatment. Also, the immunostaining of SOX9 and RUNX2 in the injured tendon section from tenotomized *Gli1-CreER^{T2}*; Ai9; *Prkaca^{ff}* mice will be added.

4. The terminology used to describe Gli1⁺ cells is inconsistent, with references to "Gli1⁺ tendon stem cells," "Gli1⁺ tendon sheath cells," and "Gli1⁺ osteochondral tendon progenitors." The authors should adopt a unified and accurate description consistent with the cellular characteristics of Gli1⁺ cells throughout the manuscript.

Our Response: We apologize for the terminology inconsistencies. We will consistently use *Gli1⁺* tendon sheath cells throughout the paper in the revised version.

5. In Fig. 2a (*Gli1* expression) and Fig. 6c (*Gnas* expression), violin plots would provide a more detailed and intuitive visualization of gene expression across different cell populations.

Our Response: We will supplement the violin plots for *Gli1* and *Gnas* expression and update **Fig. 2a** and **6c**.

6. The quality of some immunofluorescence images is poor. For example, in Fig. 3g and 3i, the magnified regions do not match the boxed areas. Additionally, supplementary Fig. 4b lacks proper labeling for the different fluorescent signals.

Our Response: We apologize for these mistakes. We will update the magnified images in **Fig. 3g** and **3i**. Regarding the labeling for GLI1 in Rainbow mice, as mentioned before, we will delete this figure and associated text.

7. Several figure legend errors: In Fig. 4a, "Gli1" should be corrected to "GLI1." In Fig. 4g, "osteogenic progenitor cells" should be corrected to "chondrogenic progenitor cells." Moreover, there are discrepancies regarding sample analysis time points between Fig. 5i and 5j, which need clarification.

Our Response: We appreciate these comments and suggestions. We will correct the *Gli1* into GLI1 in **Fig. 5a** and correct the osteogenic progenitor cells into COL2⁺ chondrocytes in **Fig. 5g**. Also, we will unify the time points in **Fig. 5i** and **5j** as 14 dpi.

8. The description in Line 297 is incorrect. Tamoxifen, not DT treatment, is used to ablate *Gli1*⁺ cells in *Gli1-CreERT2; Acvr1R206H/+; DTA* mice.

Our Response: We sincerely apologize for the inadvertent terminology error and will revise "DT" to "Tamoxifen" in the manuscript.

9. In Fig. 6d (7 dpi group) and Fig. 6i (uninjured group), annotations for the fluorescent signals are incomplete.

Our Response: We will supplement the label for GLI1/GNAS and p-PKA substrate/GLI1 in Fig. 6g (7 dpi group) and 6i (uninjured group).

10. There is an error in the antibody information. "Anti-mouse CD140a-APC" should be corrected to "Anti-human CD140a-APC."

Our Response: We apologize for this mistake. We will correct Anti-mouse CD140a-APC into anti-human CD140a-APC.

11. The figure legend in Fig. 8j contains an error. "Chondrogenic induction" should be corrected to "osteogenic induction" based on the experimental context described in the text.

Our Response: We appreciate the reviewer's meticulous attention to detail; we will correct "chondrogenic induction" to "osteogenic induction" in **Fig. 8j** and perform a thorough terminology audit across all figures to ensure full consistency.

12. Line 397: Reference citation is needed here to support the conclusion that Gli1 labels tendon enthesis progenitors.

Our Response: We thank reviewer for this suggestion. we will cite the *Gli1*⁺ tendon enthesis progenitor reference (Fei Fang et al., Cell Stem Cell, 2022).

13. Whereas an AI software seems to be used to polish the language, several grammatic errors still exist (eg. Lines 29, 34, 412, 418).

Our Response: We will correct the grammatic errors, *i.e.*, correct “remains” to “remain” and “robust acceleration of” to “accelerated”. Also, we will rewrite the sentence of lines 412 and 418 to further discuss the heterogeneity of tendon stem cells and GNAS function in HO.

14. "HO" should be defined when it first appears in the abstract.

Our Response: We will define the full name of HO in the abstract.

Referee #2:

This is an intriguing study, elucidating the role of Gli1 positive cells during apparent osteochondral differentiation of tendon sheath progenitors.

The study is based on previous findings by the authors and others that Gli1 positive cells are involved in heterotopic ossification. Using a number of elegant mouse models, including lineage tracing, cell depletion, and mechanical and genetic alterations of tendon integrity and subsequent heterotopic ossification (HO), the authors show a pivotal signaling role of GNAS in Gli1 positive cells. They first observed in a model of tendon injury that Gli1 cells were found at sites of osteochondral ossification at early and late stages. They observed in published scRNA-Seq data sets an expression of Gli1 with tenocytes and tendon stem cell/progenitor cells. Strikingly, after genetic depletion of Gli1 positive cells, HO was largely abrogated. The predicted trajectories of Gli1 positive cells could be largely reconfirmed by observed co-expression of osteogenic marker genes with GliCreERT2 Ai9 labeled cells. Moreover, using rainbow reporter mice showed that Gli1 driven Cre generated clones and clusters are present throughout the HO lesions.

The authors generate a second model of HO, resembling FOP, by introducing a Cre-loxP mediated introduction of the Acvr1R206H mutation. Additional tendon injury revealed the very rapid generation of osteochondral ossification with a relatively high percentage of Sox9 and lesser Runx2 expressing cells.

Crossing Gli-CreERT2 Ai9 with the Acvr1R206H inducible mutation suggested that most Gli1 cells give rise to FMOD positive tendon progenitor cells. A minor fraction give rise to osteochondroprogenitor cells.

In Cluster 8 from the scRNA-Seq analysis that seems to represent injury associated Gli1 positive progenitor cells being more involved in osteochondral differentiation than stem cells or tendon progenitor cells not involved in HO generation, some signaling molecules were enhanced expressed. The druggable GNAS was followed up and found increased after tendon injury at early time points. This was coincident with phospho-PKA substrates in both traumatic and FOP induced models of HO.

Strikingly, pharmacological inhibition and genetic deletion of GNAS in the two models decreased HO formation.

So in general, an important study, dissecting the role of Gli1 positive cells in HO formation, analyzing in detail the contribution of Gli1 derived cells in HO formation and providing a pharmaceutical target. Many mouse models were involved and justifiable for shedding light into this complex process. Nonetheless, the molecular mechanism of how GNAS signaling triggers Gli1 positive cells to cause HO still remains elusive and should be at least attempted in a revised manuscript.

Our response: We thank Referee #2 for the positive comments regarding our manuscript and appreciate the comments about the molecular mechanism of osteochondral differentiation of *Gli1*⁺ cells. Building upon our preliminary data showing the increase in p-PKA substrates (including CREB phosphorylation) within HO lesions, we will perform temporal profiling of p-CREB in Gli1-tdTomato⁺ cells across key differentiation stages (0/7/21 dpi) using immunofluorescence, coupled with functional validation via the selective CREB inhibitor 666-15 *in vivo* to quantify HO

suppression efficacy.

Additionally, we will perform scRNA-seq for *Gli1*⁺ cells from normal and injured tendon of *Gli1-CreER*^{T2}; Ai9 mice; this analysis (for example regulon analysis) will further identify the target genes of GNAS/PKA pathway, potentially including CREB and other candidates. As mentioned above, we have collected normal and injured tendon samples from *Gli1-CreER*^{T2}; Ai9 mice and sent out for scRNA-seq. Overall, we have confidence to complete these assays to confirm the molecular mechanism of the aberrant osteochondral differentiation of *Gli1*⁺ cells.

Major Points:

1. The authors demonstrate *Gli1* expression in the published scRNA-Seq data set to be restricted to a few clusters of tendon progenitor cells (Fig. 2A). However, the lineage tracing (Fig. 3) revealed that after the tendon injury, the majority of chondro and osteoprogenitors were not derived from *Gli1* positive cells. However, marker genes for later stages were positive in a majority for *Gli1* (*Col2*, *Ocn*). How can this be explained? A direct transdifferentiation from FMOD cells into *Col2* and *Ocn* expressing cells, for example? To address this, an FMOD driven lineage tracing could be considered. On the other hand, is there the possibility that non-*Gli1* expressing progenitors contribute to progenitors only and that they diminish and are overruled by the smaller fraction of *Gli1/Sox9* progenitors? Another possibility, the induction of ectopic ossification could rely on the recruitment of other progenitor cells by *Gli1* positive cells, given that ablation of *Gli1-CreERT2* DTR cells diminishes HO. So a detailed analysis of *Gli1CreERT2-DTA* mice concerning the effects on progenitor and maturing cells of the osteochondral lineage after tendon injury could provide further insights and should be done accordingly.

Our response: We sincerely appreciate the reviewer's insightful observations regarding the cellular dynamics of *Gli1*⁺ populations. We fully concur that *GLI1*⁺/*SOX9*⁺ and *GLI1*⁺/*RUNX2*⁺ cells exhibit lower prevalence compared to their *GLI1*⁻ counterparts, a finding consistent with our hypothesis that *Gli1*⁺ cells represent an early-stage osteochondrogenic subpopulation during initial injury responses, reflecting inherent heterogeneity within tendon progenitor pools.

Importantly, our data demonstrate substantial accumulation of *GLI1*⁺/*COL2*⁺ and *GLI1*⁺/*OCN*⁺ cells within ectopic cartilage and mature ossification zones, supporting their functional engagement in late-stage HO progression. While we agree that *Gli1*⁺-derived *Fmod*⁺ lineages may undergo chondro-osteogenic transdifferentiation at injury sites, practical constraints preclude direct validation via *Fmod-CreER*^{T2} models, as confirmed through extensive consultations with the Jackson Laboratory and multiple transgenic rodent providers in China. To address this concern, we will implement fluorescence-activated cell sorting (FACS) of *Gli1-CreER*^{T2}; Ai9 tendon cells at the steady state and critical post-injury intervals (7 and 21 dpi), enabling comparative single-cell transcriptomic profiling of: *Gli1*⁺/*Sox9*⁺/*Runx2*⁺/*Fmod*⁺ putative tenocytes; *Gli1*⁻/*Fmod*⁺ subsets and *Gli1*⁻ osteochondrogenic progenitors' populations. This

experimental paradigm will rigorously test two hypotheses: i) Whether *Gli1*⁺/*Fmod*⁺ tenocytes acquire chondro-osteogenic fate through transcriptomic reprogramming and ii) If *Gli1*⁻ osteochondrogenic progenitors exhibit temporal depletion correlating with HO maturation.

Furthermore, parallel immunohistochemical analysis of DTA versus *Gli1-CreER*^{T2}; DTA models will quantify dynamic expression of established tendon-derived progenitor markers (TPPP3/SCA1), elucidating how *Gli1*⁺ cell ablation disrupts both autonomous differentiation cascades and niche-dependent maintenance of alternative progenitor pools. Through this multimodal approach, we aim to resolve the proportional contributions of cell-autonomous versus non-autonomous mechanisms to HO suppression in our genetic models.

2. Expression of *Acvr1* and its mutant conversion should be demonstrated.

Our response: We sincerely appreciate the reviewer's critical insight regarding ACVR1 validation. To comprehensively address mutant receptor dynamics, we will leverage immunofluorescence comparing ACVR1 expression patterns in *Gli1-CreER*^{T2}; *Ai9* versus *Gli1-CreER*^{T2}; *Acvr1*^{R206H/+}; *Ai9* mice across uninjured and injured tendons (7 dpi), with particular focus on spatial colocalization of ACVR1 within *Gli1*⁺ lineage cells. Complementing this, fluorescence-activated cell sorting (FACS) of *Gli1*⁺ (tdTomato) cells from injured tendons (7 dpi) from *Gli1-CreER*^{T2}; *Ai9* and *Gli1-CreER*^{T2}; *Acvr1*^{R206H/+}; *Ai9* mice will enable allele-specific Sanger sequencing to quantify mutant allele burden. To functionally validate pathway activation, we will perform phospho-SMAD1/5 (p-SMAD1/5) quantification via immunofluorescence staining, comparing signaling intensity in injury-induced HO niches versus physiological repair zones.

3. The data of GNAS and p-PKA substrate analysis (Fig. 6) suggest that at later time points post-injury, other cells overexpress GNAS and p-PKA. Can the nature of these cells (osteoprogenitors, mature osteoblasts, chondroprogenitors, mature chondrocytes) be revealed? Are they affected by inhibitory treatment?

Our response: We appreciate the reviewer's insightful inquiry into the cellular identity and therapeutic responsiveness of GNAS/p-PKA-activated populations during late-stage injury. To systematically address this, we will integrate trajectory analysis of *Gli1*⁺ and *Gli1*⁻ cell transcriptomic dynamics (from our *Gli1-CreER*^{T2}; *Ai9* scRNA-seq dataset) with immunofluorescence in 7/21 dpi specimens and co-staining for lineage-specific markers: RUNX2 (osteoprogenitors) and SOX9 (chondroprogenitors), alongside GNAS/p-PKA substrate detection. This spatiotemporal profiling will resolve differentiation trajectories of *Gli1*⁻/GNAS⁺ populations through pseudotime ordering and RNA velocity analysis.

To functionally interrogate therapeutic vulnerability, we will administer the inhibitor NF-499 and quantify cell types above mentioned (i.e., GLI1⁻/RUNX2⁺/GNAS⁺, GLI1⁻/RUNX2⁺/GNAS⁺, GLI1⁻/SOX9⁺/GNAS⁺ and GLI1⁻/SOX9⁺/GNAS⁺)

4. The molecular mechanisms of GNAS inhibition that cause less HO formation remain elusive. For example, are the main targets CREB dependently regulated? CREB inhibitors could resolve this fact. The authors could try in addition to identify by RNA-Seq approaches, for example, further direct target genes, and eventually validate in an ex vivo approach that are implicated in the HO effects.

Our response: We sincerely thank the reviewer for highlighting this critical gap in our mechanistic understanding. We will further examine the expression of p-CREB in *Gli1*⁺ cells at different stage and applied CREB inhibitors to evaluate its role in HO prevention, if CREB is activated. However, if GNAS/PKA signaling does not mediate its effects through the canonical CREB pathway, we will further identify the target genes of GNAS/PKA pathway via scRNA-seq (e.g. Regulon analysis) of *Gli1*⁺ cell and confirm the role via ex vivo experiments. As mentioned above, we have collected normal and injured samples of *Gli1-CreER*^{T2}; Ai9 mice (7/21 dpi) and sent them for scRNA-seq. We have confidence to complete these assays to confirm the molecular mechanism of the aberrant osteochondral differentiation of *Gli1*⁺ cells.

Minor Points:

1. Official mouse Jax nomenclature should be used in the MM section.

Our Response: We will supplement the official mouse Jax nomenclature in the section of material and methods, as follows:

Gli1-CreER^{T2} (*Gli1*^{tm3(cre/ERT2)Alj}/J),

Prrx1-Cre (B6.Cg-Tg(Prrx1-cre)1Cjt/J),

Tek-Cre (B6.Cg-Tg(Tek-cre)12Flv/J),

Ai9 (B6.Cg-Gt(ROSA)26Sortm9(CAG-tdTomato)Hze/J),

DTA (B6.129P2-Gt(ROSA)26Sortm1(DTA)Lky/J)

iDTR (C57BL/6-Gt(ROSA)26Sortm1(HBEGF)Awai/J).

2. Line 113: How is the abundance of Gli1⁺ cells in the sheaths of the tendons after injury?

Our Response: We will rewrite this sentence and highlight the abundance of of *Gli1*⁺ sheath cells is increased.

3. Line 132: The term Cre-dependent Gli KO mice is misleading, the authors mean mice with the elimination of Gli1 positive progenitor cells.

Our Response: We apologized for this mistake; we will correct this sentence into “Cre-dependent *Gli1*⁺ cell elimination”.

4. Fig. 2C: It would be helpful for interpretation also to show the expression of Gli1 over the different clusters.

Our Response: We will supplement the violin plot image of *Gli1* in the new **Fig. 2c**.

5. The Figures 3i, m are difficult to read. These legends should be separated from the

y-axis.

Our Response: We will adjust the height of the statistical graphs to ensure clear visibility and legibility of the figure legends.

6. Figures 7r, t are difficult to read. X-axis description is somewhat untidy.

Our Response: We thank the reviewer for this valuable comment. We will also adjust the height of the statistical graphs to ensure clear visibility of the X-axis legends. Additionally, we will revise the presentation of these results to ensure clearer descriptions. The y-axis labels for the measured parameters will be refined to better highlight the purpose and significance of the detected indicators for readers.

Dear Dr Kan,

Thank you again for the submission of your manuscript (EMBOJ-2024-119973) to The EMBO Journal, as well as providing us with a preliminary point-by-point response to the experts' concerns raised. As mentioned, your study was assessed by two reviewers with expertise in systemic cell differentiation analysis and bone fate specification, whose comments are enclosed below.

As you will see from their comments, the referees acknowledge the analysis and potential interest and value of your findings. However, they also express major concerns i.p. regarding i) conclusive support for core claims made on tenogenic and osteochondral differentiation capacities of Gli1-CreER-traced cells (Ref#1, pts. 3; ref#2, pt.1,2); as well as ii) the degree of mechanistic insights into GNAS - p-PKA and downstream targets in this process (Ref#1, pt.6; #2, pt.4). Referee #1 in addition questions the conclusiveness of the Rainbow mice data part of your study (Ref#1, pt.4). Finally, the reviewers also raise a number of issues related to the data presentation, additional controls and improved methods annotation required, as well as and overall discussion of related literature, that would need to be conclusively addressed to achieve the level of robustness and clarity needed for The EMBO Journal.

Given the overall interest stated and broader angle of your findings, we are able to invite you to revise your manuscript experimentally to address the referees' comments along the lines indicated in the preliminary response. I need to stress though that we do require strong support from the referees on a revised version of the study in order to move on to publication of the work, and suggest keeping EMBO Reports in mind as an alternative venue for this study.

Please feel free to contact me if you have any questions or need further input on the referee comments.

When submitting your revised manuscript, please carefully review the instructions below.

Please feel free to approach me any time should you have additional questions related to this.

Thank you for the opportunity to consider your work for publication.

I look forward to your revision.

Kind regards,

Daniel Klimmeck

Daniel Klimmeck, PhD
Senior Editor
The EMBO Journal

Instruction for the preparation of your revised manuscript:

- 1) a .docx formatted version of the manuscript text (including legends for main figures, EV figures and tables). Please make sure that the changes are highlighted to be clearly visible.
- 2) individual production quality figure files as .eps, .tif, .jpg (one file per figure).
- 3) a .docx formatted letter INCLUDING the reviewers' reports and your detailed point-by-point response to their comments. As part of the EMBO Press transparent editorial process, the point-by-point response is part of the Review Process File (RPF), which will be published alongside your paper.
- 4) a complete author checklist, which you can download from our author guidelines ([https://wol-prod-cdn.literatumonline.com/pb-assets/embo-site/Author Checklist%20-%20EMBO%20J-1561436015657.xlsx](https://wol-prod-cdn.literatumonline.com/pb-assets/embo-site/Author%20Checklist%20-%20EMBO%20J-1561436015657.xlsx)). Please insert information in the checklist that is

also reflected in the manuscript. The completed author checklist will also be part of the RPF.

6) It is mandatory to include a 'Data Availability' section after the Materials and Methods. Before submitting your revision, primary datasets produced in this study need to be deposited in an appropriate public database, and the accession numbers and database listed under 'Data Availability'. Please remember to provide a reviewer password if the datasets are not yet public (see <https://www.embopress.org/page/journal/14602075/authorguide#datadeposition>).

7) Our journal encourages inclusion of *data citations in the reference list* to directly cite datasets that were re-used and obtained from public databases. Data citations in the article text are distinct from normal bibliographical citations and should directly link to the database records from which the data can be accessed. In the main text, data citations are formatted as follows: "Data ref: Smith et al, 2001" or "Data ref: NCBI Sequence Read Archive PRJNA342805, 2017". In the Reference list, data citations must be labeled with "[DATASET]". A data reference must provide the database name, accession number/identifiers and a resolvable link to the landing page from which the data can be accessed at the end of the reference. Further instructions are available at .

8) At EMBO Press we ask authors to provide source data for the main and EV figures. Our source data coordinator will contact you to discuss which figure panels we would need source data for and will also provide you with helpful tips on how to upload and organize the files.

Numerical data can be provided as individual .xls or .csv files (including a tab describing the data). For 'blots' or microscopy, uncropped images should be submitted (using a zip archive or a single pdf per main figure if multiple images need to be supplied for one panel). Additional information on source data and instruction on how to label the files are available at .

9) We replaced Supplementary Information with Expanded View (EV) Figures and Tables that are collapsible/expandable online (see examples in <https://www.embopress.org/doi/10.15252/emj.201695874>). A maximum of 5 EV Figures can be typeset. EV Figures should be cited as 'Figure EV1, Figure EV2' etc. in the text and their respective legends should be included in the main text after the legends of regular figures.

11) For data quantification: please specify the name of the statistical test used to generate error bars and P values, the number (n) of independent experiments (specify technical or biological replicates) underlying each data point and the test used to calculate p-values in each figure legend. The figure legends should contain a basic description of n, P and the test applied. Graphs must include a description of the bars and the error bars (s.d., s.e.m.).

We realize that it is difficult to revise to a specific deadline. In the interest of protecting the conceptual advance provided by the work, we recommend a revision within 3 months (2nd Jun 2025). Please discuss the revision progress ahead of this time with the editor if you require more time to complete the revisions.

Referee #1:

In this study, Chen et al. conducted an in-depth investigation into the regulatory mechanisms of heterotopic ossification (HO) following tendon injury. By combining genetic lineage tracing and single-cell RNA sequencing (scRNA-seq) analyses, they identified Gli1+ tendon sheath cells as a key initiator of osteochondral differentiation after injury. Furthermore, they demonstrated that GNAS/PKA signaling is activated in Gli1+ tendon sheath cells, and that pharmacological or genetic blockade of this pathway significantly ameliorated HO after tenotomy. Overall, this is an interesting study that utilized a series of genetic models to convincingly prove their hypothesis. The human data further emphasize the translational implication of this study. However, a few concerns need to be addressed in a potential revision.

Major Points:

1. The trajectory analysis of scRNA-seq data used to predict the downstream differentiation trajectory of Gli1+ cells is not robust enough. Pseudotime analysis by Monocle2 implies no directionality. RNA Velocity or SlingShot analyses should be a better choice.
2. Lineage tracing of Gli1+ cells during development and postnatal growth of tendon in uninjured mice should be added, in order to demonstrate that Gli1+ cells function as tendon stem/progenitor cells under steady state.
3. Gli1-expressing cells in the scRNA-seq analyses are not equivalent to Gli1-CreER+ cells in the lineage tracing study. Gli1-CreER+ cells may down-regulate Gli1 expression upon their activation and differentiation. Therefore, instead of merely analyzing a published dataset, it would be much more informative to perform scRNA-seq on Gli1-CreER-traced cells at different time points post tenotomy to prove their tenogenic and osteochondral differentiation capacities.
4. The Rainbow mice experiment is not convincing. The authors only observed some clonal clusters without staining chondrogenic or osteogenic markers to prove that osteochondral differentiation was clonally derived in vivo. To prove the self-renewal capacity of Gli1+ cells, primary and secondary ectopic transplantation should be performed. Alternatively, the author could simply delete this part to avoid confusion or further argument.
5. It is difficult to identify GLI1+/RUNX2+ cells in Fig. 2m, and GLI1+/SCX+/RUNX2+ cells in Fig. 3n. Split-channel images are necessary for clarification. Additionally, the proportion of GLI1-/SCX+/SOX9+ cells is significantly higher than GLI1+/SCX+/SOX9+ cells in the injured region, suggesting alternative cellular sources for SOX9+ chondrocytes. A similar result is found in RUNX2+ osteoblasts.
6. The conclusion that "GLI1, GNAS, and p-PKA substrate expression is higher in human tendon HO" lacks convincing evidence. Images of Fig. 8b and 8d show that GLI1 expression levels are similar in normal tendon sheath and HO lesions. Moreover, the manuscript does not clarify the in-situ expression patterns of GNAS and phospho-PKA substrates in uninjured versus HO regions. Is there existing literature supporting the use of CD140a+/CD45- as markers for human tendon stem cells? Furthermore, it is unclear whether GNAS and phospho-PKA substrates are activated during the osteogenic differentiation of human tendon stem cells.

Minor points:

1. In Fig. 1b, H&E staining should be performed to provide a comprehensive view of specific regions. Additionally, the magnified image at 21 dpi does not align with the statistical data showing a continuous increase in Gli1+ cells. Moreover, it remains unclear whether the GLI1 signal represents endogenous expression or lineage tracing results.
2. In Fig. 1h-j, alongside bone mass analysis, the authors should quantify changes in Gli1+ cells in both WT and DTA mice to confirm that the improvement in HO is indeed due to Gli1+ cell ablation.
3. The authors claim that inhibiting GNAS/PKA signaling prevents tendon HO by impairing osteochondral differentiation in Gli1+ tendon sheath cells. However, they do not provide data on how Gli1+ SOX9+ and Gli1+ RUNX2+ cell populations change following NF449 treatment or Prkaca knockout.
4. The terminology used to describe Gli1+ cells is inconsistent, with references to "Gli1+ tendon stem cells," "Gli1+ tendon sheath cells," and "Gli1+ osteochondral tendon progenitors." The authors should adopt a unified and accurate description consistent with the cellular characteristics of Gli1+ cells throughout the manuscript.
5. In Fig. 2a (Gli1 expression) and Fig. 6c (Gnas expression), violin plots would provide a more detailed and intuitive visualization of gene expression across different cell populations.
6. The quality of some immunofluorescence images is poor. For example, in Fig. 3g and 3i, the magnified regions do not match the boxed areas. Additionally, supplementary Fig. 4b lacks proper labeling for the different fluorescent signals.
7. Several figure legend errors: In Fig. 4a, "Gli1" should be corrected to "GLI1." In Fig. 4g, "osteogenic progenitor cells" should be corrected to "chondrogenic progenitor cells." Moreover, there are discrepancies regarding sample analysis time points

between Fig. 5i and 5j, which need clarification.

8. The description in Line 297 is incorrect. Tamoxifen, not DT treatment, is used to ablate Gli1+ cells in Gli1-CreERT2; Acvr1R206H/+; DTA mice.

9. In Fig. 6d (7 dpi group) and Fig. 6i (uninjured group), annotations for the fluorescent signals are incomplete.

10. There is an error in the antibody information. "Anti-mouse CD140a-APC" should be corrected to "Anti-human CD140a-APC."

11. The figure legend in Fig. 8j contains an error. "Chondrogenic induction" should be corrected to "osteogenic induction" based on the experimental context described in the text.

12. Line 397: Reference citation is needed here to support the conclusion that Gli1 labels tendon enthesis progenitors.

13. Whereas an AI software seems to be used to polish the language, several grammatical errors still exist (eg. Lines 29, 34, 412, 418).

14. "HO" should be defined when it first appears in the abstract.

Referee #2:

This is an intriguing study, elucidating the role of Gli1 positive cells during apparent osteochondral differentiation of tendon sheath progenitors.

The study is based on previous findings by the authors and others that Gli1 positive cells are involved in heterotopic ossification. Using a number of elegant mouse models, including lineage tracing, cell depletion, and mechanical and genetic alterations of tendon integrity and subsequent heterotopic ossification (HO), the authors show a pivotal signaling role of GNAS in Gli1 positive cells. They first observed in a model of tendon injury that Gli1 cells were found at sites of osteochondral ossification at early and late stages. They observed in published scRNA-Seq data sets an expression of Gli1 with tenocytes and tendon stem cell/progenitor cells. Strikingly, after genetic depletion of Gli1 positive cells, HO was largely abrogated. The predicted trajectories of Gli1 positive cells could be largely reconfirmed by observed co-expression of osteogenic marker genes with GliCreERT2 Ai9 labeled cells. Moreover, using rainbow reporter mice showed that Gli1 driven Cre generated clones and clusters are present throughout the HO lesions.

The authors generate a second model of HO, resembling FOP, by introducing a Cre-loxP mediated introduction of the Acvr1R206H mutation. Additional tendon injury revealed the very rapid generation of osteochondral ossification with a relatively high percentage of Sox9 and lesser Runx2 expressing cells.

Crossing Gli-CreERT2 Ai9 with the Acvr1R206H inducible mutation suggested that most Gli1 cells give rise to FMOD positive tendon progenitor cells. A minor fraction give rise to osteochondroprogenitor cells.

In Cluster 8 from the scRNA-Seq analysis that seems to represent injury associated Gli1 positive progenitor cells being more involved in osteochondral differentiation than stem cells or tendon progenitor cells not involved in HO generation, some signaling molecules were enhanced expressed. The druggable GNAS was followed up and found increased after tendon injury at early time points. This was coincident with phospho-PKA substrates in both traumatic and FOP induced models of HO.

Strikingly, pharmacological inhibition and genetic deletion of GNAS in the two models decreased HO formation.

So in general, an important study, dissecting the role of Gli1 positive cells in HO formation, analyzing in detail the contribution of Gli1 derived cells in HO formation and providing a pharmaceutical target. Many mouse models were involved and justifiable for shedding light into this complex process. Nonetheless, the molecular mechanism of how GNAS signaling triggers Gli1 positive cells to cause HO still remains elusive and should be at least attempted in a revised manuscript.

Major Points:

1. The authors demonstrate Gli1 expression in the published scRNA-Seq data set to be restricted to a few clusters of tendon progenitor cells (Fig. 2A). However, the lineage tracing (Fig. 3) revealed that after the tendon injury, the majority of chondro and osteoprogenitors were not derived from Gli1 positive cells. However, marker genes for later stages were positive in a majority for Gli1 (Col2, Ocn). How can this be explained? A direct transdifferentiation from FMOD cells into Col2 and Ocn expressing cells, for example? To address this, an FMOD driven lineage tracing could be considered. On the other hand, is there the possibility that non-Gli1 expressing progenitors contribute to progenitors only and that they diminish and are overruled by the smaller fraction of Gli1/Sox9 progenitors? Another possibility, the induction of ectopic ossification could rely on the recruitment of other progenitor cells by Gli1 positive cells, given that ablation of Gli-CreERT2 DTR cells diminishes HO. So a detailed analysis of Gli1CreERT2-DTA mice concerning the effects on progenitor and maturing cells of the osteochondral lineage after tendon injury could provide further insights and should be done accordingly.

2. Expression of Acvr1 and its mutant conversion should be demonstrated.

3. The data of GNAS and p-PKA substrate analysis (Fig. 6) suggest that at later time points post-injury, other cells overexpress GNAS and p-PKA. Can the nature of these cells (osteoprogenitors, mature osteoblasts, chondroprogenitors, mature chondrocytes) be revealed? Are they affected by inhibitory treatment?

4. The molecular mechanisms of GNAS inhibition that cause less HO formation remain elusive. For example, are the main targets CREB dependently regulated? CREB inhibitors could resolve this fact. The authors could try in addition to identify by RNA-Seq approaches, for example, further direct target genes, and eventually validate in an ex vivo approach that are implicated in the HO effects.

Minor Points:

1. Official mouse Jax nomenclature should be used in the MM section.

2. Line 113: How is the abundance of Gli1+ cells in the sheaths of the tendons after injury?

3. Line 132: The term Cre-dependent Gli KO mice is misleading, the authors mean mice with the elimination of Gli1 positive progenitor cells.

4. Fig. 2C: It would be helpful for interpretation also to show the expression of Gli1 over the different clusters.
5. The Figures 3i, m are difficult to read. These legends should be separated from the y-axis.
6. Figures 7r, t are difficult to read. X-axis description is somewhat untidy.

To the editor,

Re: EMBOJ-2024-119973R-Q GNAS/PKA pathway promotes aberrant osteochondral differentiation of Gli1⁺ tendon sheath progenitors.

We would like to thank the Reviewer for the particularly detailed and helpful reading of our manuscript. The changes and corrections made in response to the Reviewer's points have been systematically highlighted in yellow within the revised document. These enhancements greatly improved the manuscript. To facilitate discussion, we have addressed each point raised by the Reviewer in order:

Referee #1:

In this study, Chen et al. conducted an in-depth investigation into the regulatory mechanisms of heterotopic ossification (HO) following tendon injury. By combining genetic lineage tracing and single-cell RNA sequencing (scRNA-seq) analyses, they identified Gli1⁺ tendon sheath cells as a key initiator of osteochondral differentiation after injury. Furthermore, they demonstrated that GNAS/PKA signaling is activated in Gli1⁺ tendon sheath cells, and that pharmacological or genetic blockade of this pathway significantly ameliorated HO after tenotomy. Overall, this is an interesting study that utilized a series of genetic models to convincingly prove their hypothesis. The human data further emphasize the translational implication of this study. However, a few concerns need to be addressed in a potential revision.

Our response: We thank Referee #1 for the positive comments regarding our manuscript.

Major Points:

1. The trajectory analysis of scRNA-seq data used to predict the downstream differentiation trajectory of Gli1⁺ cells is not robust enough. Pseudotime analysis by Monocle2 implies no directionality. RNA Velocity or Slingshot analyses should be a better choice.

Our response: We thank the reviewer for this suggestion. In accordance with this recommendation, we have conducted Slingshot trajectory analysis—the only feasible approach applicable to the processed data from GSE126060—to investigate the potential developmental relationships between *Gli1*⁺ cell clusters (1, 2, 4, 5, 6, 7, 8, 9, 15 and 16). Consistent with the Monocle 2 analysis, cluster 8 cells, originating from cluster 5 cells, could differentiate into tenogenic (5→8→2→15), chondrogenic (5→8→7→1→16→18) and osteogenic (5→8→4→9→17) lineage cells. The results of this analysis have been incorporated into the **Appendix Fig. S2C, D**.

2. Lineage tracing of Gli1⁺ cells during development and postnatal growth of tendon in uninjured mice should be added, in order to demonstrate that Gli1⁺ cells function as tendon stem/progenitor cells under steady state.

Our response: We sincerely appreciate the reviewer's insightful suggestion regarding lineage tracing of *Gli1*⁺ cells during tendon development. To comprehensively

address this question, we established a dual-transgenic model by crossing *Gli1-CreER^{T2}* mice with *ZsGreen* reporter mice. Tamoxifen was administered via intraperitoneal injection to pregnant dams at embryonic day 10.5 (E10.5) to induce *Cre*-mediated recombination for lineage tracing. Tendon tissues were harvested at key developmental stages (E14.5, P0, P3, P14) and subjected to systematic analysis following PCR-based genotyping of *Gli1-CreER^{T2}*; *ZsGreen* mice. Our results demonstrate that *Gli1*⁺ cells initially localized to the tendon sheath at E14.5, with subsequent expansion throughout the tendon tissue to encompass both the sheath and midsubstance regions by P0, P3, and P14. These data have been presented in **Appendix Fig. S5**, providing definitive evidence that *Gli1*⁺ cells function as bona fide tendon stem/progenitor cells under homeostatic conditions.

3. *Gli1*-expressing cells in the scRNA-seq analyses are not equivalent to *Gli1-CreER*⁺ cells in the lineage tracing study. *Gli1-CreER*⁺ cells may down-regulate *Gli1* expression upon their activation and differentiation. Therefore, instead of merely analyzing a published dataset, it would be much more informative to perform scRNA-seq on *Gli1-CreER*-traced cells at different time points post tenotomy to prove their tenogenic and osteochondral differentiation capacities.

Our response: We agree with Referee #1 that *Gli1*-expressing cells in the scRNA-seq analyses are not equivalent to *Gli1-CreER^{T2+}* cells. To rigorously delineate the temporal dynamics of *Gli1-CreER^{T2+}* progenitor cell contributions to heterotopic ossification, we have established a controlled longitudinal cohort comprising *Gli1-CreER^{T2}*; Ai9 reporter mice with tamoxifen induction (n=24, equal sex distribution), stratified into three experimental groups: uninjured controls (n=8) and tenotomy-injured cohorts at 7-, and 21-days post-injury (n=8/group). All samples have been collected and subjected to single-cell RNA sequencing. Raw sequencing data are undergoing processing through the 10x Genomics Cell Ranger pipeline (v9.0.0) and have been deposited in GEO under accession number GSE298748.

UMAP-based dimensionality reduction and graph-based clustering identified 11 transcriptionally distinct subpopulations within *Gli1-CreER^{T2+}* cells. Cell type annotation identified: Tendon stem cells (clusters 3, 4 and 8; *Tppp3*^{high}/*Ly6a*^{High}/*Scx*^{low}), tendon progenitors (clusters 1, 2, 6 and 9; *Tppp3*^{mid}/*Ly6a*^{high}/*Scx*^{high}), Tenocytes (cluster 7; *Tppp3*^{low}/*Ly6a*^{low}/*Scx*^{high}/*Fmod*^{high}), osteochondral lineages (cluster 5: Fibrochondrocytes, *Tppp3*^{low}/*Sox9*⁺/*Acan*⁺/*Tnfaip6*^{high}; cluster 11, mature chondrocytes, *Tppp3*^{low}/*Col2a1*^{high}/*Acan*⁺/*Tnfaip6*^{low}; cluster 10: osteogenic progenitors, *Tppp3*^{mid}/*Ly6a*^{high}/*Scx*^{high}/*Runx2*^{high}) (see **new Fig. EV2A, B**).

Furthermore, RNA velocity analysis demonstrated a hierarchical differentiation trajectory from tendon stem cells (clusters 3, 4 and 8) → tendon progenitors (clusters 1, 2, 6 and 9) → tenocytes/osteochondral lineages (clusters 5, 7, 10 and 11) (see **Fig. EV2C**). These dynamic transitions strongly support our central premise that *Gli1-CreER^{T2+}* progenitors adopt divergent differentiation fates during heterotopic ossification.

4. The Rainbow mice experiment is not convincing. The authors only observed some clonal clusters without staining chondrogenic or osteogenic markers to prove that osteochondral differentiation was clonally derived in vivo. To prove the self-renewal capacity of *Gli1*⁺ cells, primary and secondary ectopic transplantation should be performed. Alternatively, the author could simply delete this part to avoid confusion or further argument.

Our response: We sincerely appreciate the reviewer's guidance regarding the Rainbow mice experiment. After careful reconsideration of the experimental limitations raised, we fully agree that removing this section will enhance the manuscript's conceptual focus and methodological rigor. Accordingly, we have deleted **the original Fig. S4** and associated text from both the main text and Supplementary Figures. Additionally, we also temper the identity of *Gli1*⁺ cells in the section of Discussion.

5. It is difficult to identify GLI1⁺/RUNX2⁺ cells in Fig. 2m, and GLI1⁺/SCX⁺/RUNX2⁺ cells in Fig. 3n. Split-channel images are necessary for clarification. Additionally, the proportion of GLI1⁻/SCX⁺/SOX9⁺ cells is significantly higher than GLI1⁺/SCX⁺/SOX9⁺ cells in the injured region, suggesting alternative cellular sources for SOX9⁺ chondrocytes. A similar result is found in RUNX2⁺ osteoblasts.

Our response: We thank the reviewer for this suggestion. We have updated the immunostaining images of GLI1/ RUNX2 and the split-channel images were also added (**see the new Fig. 2N**). To enhance the interpretability of GLI1/ SCX/ RUNX2 co-staining, we have incorporated the full split-channel analysis of these triple-labeled specimens into **new Fig. 3N**.

Additionally, we fully concur with the reviewer's astute observation regarding the potential contribution of *Gli1*⁻ progenitor subpopulations to osteochondral lineage differentiation, which aligns with emerging evidence of functional heterogeneity within tendon stem/progenitor cell compartments. To investigate this possibility, we FACS-sorted *Gli1-CreER*^{T2-} cells and conducted single-cell RNA sequencing, revealing that these progenitors retain capacity for both tenogenic differentiation and osteochondral commitment, as documented in **Appendix Fig. S3**.

While GLI1⁻/SCX⁺/SOX9⁺ cells numerically dominate their GLI1⁺ counterparts, our experimental evidence demonstrating that *Gli1*⁺ cell ablation reduces heterotopic ossification suggests a dual-pathway regulatory mechanism: 1) Direct depletion of osteochondrogenic *Gli1*⁺ progenitors, and 2) Secondary disruption of niche-dependent maintenance in *Gli1*⁻ subpopulations through intercellular crosstalk. To investigate this hierarchical relationship, we are currently performing immunofluorescence analysis of established tendon progenitor markers (TPPP3) and osteochondral markers (SOX9 and RUNX2) in both *Gli1-CreER*^{T2} and *Gli1-CreER*^{T2}; DTA models, with particular focus on spatial distribution patterns of TPPP3⁺ cells within injury-induced

microenvironments. As expected, *Gli1*⁺ cell depletion resulted in a significant reduction of the total TPPP3⁺ tendon progenitor population (see **Fig. EV1A-D**). This was accompanied by corresponding decreases in SOX9⁺ chondrogenic progenitors, RUNX2⁺ osteochondrogenic progenitors, COL2A⁺ mature chondrocytes and OPN⁺ osteoblasts (see **Fig. EV1E-L**). These findings collectively establish *Gli1*⁺ cells as crucial regulators of both cell-autonomous differentiation and non-autonomous progenitor maintenance. We have incorporated these insights in the revised Discussion section (see **line 450, page 17**).

6. The conclusion that "GLI1, GNAS, and p-PKA substrate expression is higher in human tendon HO" lacks convincing evidence. Images of Fig. 8b and 8d show that GLI1 expression levels are similar in normal tendon sheath and HO lesions. Moreover, the manuscript does not clarify the in-situ expression patterns of GNAS and phospho-PKA substrates in uninjured versus HO regions. Is there existing literature supporting the use of CD140a⁺/CD45⁻ as markers for human tendon stem cells? Furthermore, it is unclear whether GNAS and phospho-PKA substrates are activated during the osteogenic differentiation of human tendon stem cells.

Our Response: We sincerely appreciate the reviewer's rigorous evaluation of our human HO analyses. While immunofluorescence intensity quantification shows comparable GLI1 levels between normal tendon sheath and HO lesions, spatial distribution analysis reveals significant expansion of GLI1⁺ cells in HO. We have supplemented the statistical analysis for GLI1⁺ area (**Fig. 8E**) and revise the conclusion to: "GLI1⁺ cells expand significantly in human HO lesions" (see **Line 408, page 16**). Also, we modified the title from "High GNAS and Phospho (p)-PKA substrate expression are associated with HO in humans" to "The spatial density of GNAS and p-PKA substrate is associated with HO in humans".

Additionally, we have supplemented the immunofluorescence staining assay for GNAS and p-PKA substrate in normal tendon and HO lesions, supporting that GNAS/PKA pathway is activated in human HO (**Appendix Fig. S10A-D**).

Regarding human tendon stem cell markers, we chose CD140a (*PDGFRA*) according to the scRNA-seq analysis of human tendon (see **ref 32**, Adrian R. Kendal et al., (2020), Multi-omic single cell analysis resolves novel stromal cell populations in healthy and diseased human tendon. *Sci Rep* 10, 13939). CD140a⁺ tendon progenitor cells consist of multipotent progenitors (including tenogenic progenitor and FAP), which can differentiate into tenocytes, fibroblast and osteochondrogenic lineages. We have cited this paper when we mention the markers of human tendon stem cells and update the human stem cells into human tendon multipotent cells. Also, we have supplemented the expression of GNAS and p-PKA substrates in CD140a⁺ human tendon multipotent cells with induction of osteogenic differentiation in **Appendix Fig. S10E-G**. Collectively, these results demonstrate activation of the GNAS/PKA pathway during aberrant osteochondral differentiation in tendon multipotent cells.

Minor points:

1. In Fig. 1b, H&E staining should be performed to provide a comprehensive view of specific regions. Additionally, the magnified image at 21 dpi does not align with the statistical data showing a continuous increase in Gli1+ cells. Moreover, it remains unclear whether the GLI1 signal represents endogenous expression or lineage tracing results.

Our Response: We thank the reviewer for this suggestion. We have added HE-stained sections showing normal and injured tendons at multiple post-injury time points in **Appendix Fig. S1A**. Also, we have updated the magnified images of tendon sheath at 21 dpi in **Fig. 1B**. Moreover, GLI1 in the **Fig. 1B** represents the lineage tracing results, we have supplemented the description in the legend of **Fig. 1B** (see **line 756, page 27**).

2. In Fig. 1h-j, alongside bone mass analysis, the authors should quantify changes in Gli1+ cells in both WT and DTA mice to confirm that the improvement in HO is indeed due to Gli1+ cell ablation.

Our Response: We have supplemented immunostaining of injured tendon section from *Gli1-CreER^{T2}; Ai9* and *Gli1-CreER^{T2}; Ai9; DTA* mice with tenotomy injury for 7 days and found that *Gli1-Tdtomato⁺* cells in *Gli1-CreER^{T2}; Ai9; DTA* mice was significantly decreased, compared to *Gli1-CreER^{T2}; Ai9* mice (**Fig. EV1A, B**), proving that decreased HO is due to *Gli1⁺* cell ablation.

3. The authors claim that inhibiting GNAS/PKA signaling prevents tendon HO by impairing osteochondral differentiation in Gli1+ tendon sheath cells. However, they do not provide data on how Gli1+ SOX9+ and Gli1+ RUNX2+ cell populations change following NF449 treatment or Prkaca knockout.

Our Response: We have supplemented the immunostaining of SOX9 and RUNX2 in the injured tendon section from tenotomized *Gli1-CreER^{T2}; Ai9* mice treated with vehicle or NF449 treatment. As expected, the populations of GLI1⁺/SOX9⁺ and GLI1⁺/RUNX2⁺ cells were both significantly decreased after NF449 treatment (see **Appendix Fig. S9A-D**).

Additionally, the immunostaining of SOX9 and RUNX2 in the injured tendon sections from tenotomized *Gli1-CreER^{T2}; Ai9* and *Gli1-CreER^{T2}; Ai9; Prkaca^{ff}* mice was documented in **Appendix Fig. S9E-H**, further supporting that GNAS/PKA pathway promotes osteochondral differentiation of *Gli1-CreER^{T2+}* tendon sheath cells.

4. The terminology used to describe Gli1+ cells is inconsistent, with references to "Gli1+ tendon stem cells," "Gli1+ tendon sheath cells," and "Gli1+ osteochondral tendon progenitors." The authors should adopt a unified and accurate description consistent with the cellular characteristics of Gli1+ cells throughout the manuscript.

Our Response: We apologize for the terminology inconsistencies. In the revised version of the paper, we consistently use the term "*Gli1+* tendon sheath progenitors" throughout.

5. In Fig. 2a (Gli1 expression) and Fig. 6c (Gnas expression), violin plots would provide a more detailed and intuitive visualization of gene expression across different cell populations.

Our Response: We have supplemented the violin plots for *Gli1* and *Gnas* expression in new **Fig. 2C** and **Fig. 6C**.

6. The quality of some immunofluorescence images is poor. For example, in Fig. 3g and 3i, the magnified regions do not match the boxed areas. Additionally, supplementary Fig. 4b lacks proper labeling for the different fluorescent signals.

Our Response: We apologize for these mistakes. We have revised the highlighted area in **Fig. 3G**. Additionally, to improve the quality of immunostaining images in **Fig. 3I**, we have updated the higher-resolution images for GLI1 and SCX at 63 dpi.

Regarding the labeling for GLI1 in Rainbow mice, as mentioned before, we have deleted this figure and associated text.

7. Several figure legend errors: In Fig. 4a, "Gli1" should be corrected to "GLI1." In Fig. 4g, "osteogenic progenitor cells" should be corrected to "chondrogenic progenitor cells." Moreover, there are discrepancies regarding sample analysis time points between Fig. 5i and 5j, which need clarification.

Our Response: We appreciate these comments and suggestions. We have implemented the following revisions: In **Fig. 5A** legend, "Gli1" has been corrected to "GLI1"; in **Fig. 5G** legend, "osteogenic progenitor cells" was deleted; additionally, time points in **Fig. 5I, J** were unified at 14 days post-injury (dpi) for temporal consistency.

8. The description in Line 297 is incorrect. Tamoxifen, not DT treatment, is used to ablate Gli1+ cells in *Gli1-CreER^{T2}; Acvr1^{R206H/+}*; DTA mice.

Our Response: We sincerely apologize for the inadvertent terminology error and have revised "DT" to "tamoxifen" in the manuscript (see **line 321, page 12**).

9. In Fig. 6d (7 dpi group) and Fig. 6i (uninjured group), annotations for the fluorescent signals are incomplete.

Our Response: We sincerely appreciate the reviewer's meticulous observation. As recommended, fluorescent signal labels for GLI1/GNAS and p-PKA substrate/GLI1 have been supplemented in **Fig. 6D** (7 dpi group) and **Fig. 6I** (uninjured group).

10. There is an error in the antibody information. "Anti-mouse CD140a-APC" should be corrected to "Anti-human CD140a-APC."

Our Response: We apologize for this mistake. We have corrected Anti-mouse CD140a-APC into anti-human CD140a-APC in the **Reagent table**.

11. The figure legend in Fig. 8j contains an error. "Chondrogenic induction" should be

corrected to "osteogenic induction" based on the experimental context described in the text.

Our Response: We appreciate the reviewer's meticulous attention to detail; we have revised this text.

12. Line 397: Reference citation is needed here to support the conclusion that Gli1 labels tendon enthesis progenitors.

Our Response: We thank reviewer for this suggestion. we have cited the *Gli1*⁺ tendon enthesis progenitor reference (Ref 9, Fei Fang et al., Cell Stem Cell, 2022) on **line 436, page 17**.

13. Whereas an AI software seems to be used to polish the language, several grammatic errors still exist (eg. Lines 29, 34, 412, 418).

Our Response: We have corrected the grammatic errors, *i.e.*, correct “remains” to “remain” and “robust acceleration of” to “accelerate”. Also, we have rewritten the sentence of lines 412 and 418 to further discuss the heterogeneity of tendon stem cells and GNAS function in HO.

14. "HO" should be defined when it first appears in the abstract.

Our Response: We have defined the full name of HO in the abstract.

Referee #2:

This is an intriguing study, elucidating the role of Gli1 positive cells during apparent osteochondral differentiation of tendon sheath progenitors.

The study is based on previous findings by the authors and others that Gli1 positive cells are involved in heterotopic ossification. Using a number of elegant mouse models, including lineage tracing, cell depletion, and mechanical and genetic alterations of tendon integrity and subsequent heterotopic ossification (HO), the authors show a pivotal signaling role of GNAS in Gli1 positive cells. They first observed in a model of tendon injury that Gli1 cells were found at sites of osteochondral ossification at early and late stages. They observed in published scRNA-Seq data sets an expression of Gli1 with tenocytes and tendon stem cell/progenitor cells. Strikingly, after genetic depletion of Gli1 positive cells, HO was largely abrogated. The predicted trajectories of Gli1 positive cells could be largely reconfirmed by observed co-expression of osteogenic marker genes with GliCreERT2 Ai9 labeled cells. Moreover, using rainbow reporter mice showed that Gli1 driven Cre generated clones and clusters are present throughout the HO lesions.

The authors generate a second model of HO, resembling FOP, by introducing a Cre-loxP mediated introduction of the Acvr1R206H mutation. Additional tendon injury revealed the very rapid generation of osteochondral ossification with a relatively high percentage of Sox9 and lesser Runx2 expressing cells.

Crossing Gli-CreERT2 Ai9 with the Acvr1R206H inducible mutation suggested that most Gli1 cells give rise to FMOD positive tendon progenitor cells. A minor fraction give rise to osteochondroprogenitor cells.

In Cluster 8 from the scRNA-Seq analysis that seems to represent injury associated Gli1 positive progenitor cells being more involved in osteochondral differentiation than stem cells or tendon progenitor cells not involved in HO generation, some signaling molecules were enhanced expressed. The druggable GNAS was followed up and found increased after tendon injury at early time points. This was coincident with phospho-PKA substrates in both traumatic and FOP induced models of HO.

Strikingly, pharmacological inhibition and genetic deletion of GNAS in the two models decreased HO formation.

So in general, an important study, dissecting the role of Gli1 positive cells in HO formation, analyzing in detail the contribution of Gli1 derived cells in HO formation and providing a pharmaceutical target. Many mouse models were involved and justifiable for shedding light into this complex process. Nonetheless, the molecular mechanism of how GNAS signaling triggers Gli1 positive cells to cause HO still remains elusive and should be at least attempted in a revised manuscript.

Our response: We sincerely thank Referee #2 for their positive assessment of our manuscript and valuable insights regarding the molecular mechanism underlying osteochondral differentiation in *Gli1*⁺ cells. Building on our preliminary data demonstrating elevated phosphorylation of PKA substrates (including CREB) within heterotopic ossification (HO) lesions, we performed scRNA-seq on *Gli1*⁺ cells isolated from both normal and injured tendons of *Gli1-CreER*^{T2}; Ai9 mice.

Regulon analysis further identified downstream targets of the GNAS/PKA pathway, such as *Creb3* (see **Fig. EV5A**). Complemented by functional validation via *in vivo* administration of the selective CREB inhibitor 666-15, we observed that CREB inhibition significantly attenuated HO formation (see **Fig. EV5B-E**). Collectively, these findings reinforce the GNAS/PKA signaling axis as a core molecular driver of aberrant osteochondral differentiation in *Gli1*⁺ cells.

Major Points:

1. The authors demonstrate *Gli1* expression in the published scRNA-Seq data set to be restricted to a few clusters of tendon progenitor cells (Fig. 2A). However, the lineage tracing (Fig. 3) revealed that after the tendon injury, the majority of chondro and osteoprogenitors were not derived from *Gli1* positive cells. However, marker genes for later stages were positive in a majority for *Gli1* (*Col2*, *Ocn*). How can this be explained? A direct transdifferentiation from FMOD cells into *Col2* and *Ocn* expressing cells, for example? To address this, an FMOD driven lineage tracing could be considered. On the other hand, is there the possibility that non-*Gli1* expressing progenitors contribute to progenitors only and that they diminish and are overruled by the smaller fraction of *Gli1*/*Sox9* progenitors? Another possibility, the induction of ectopic ossification could rely on the recruitment of other progenitor cells by *Gli1* positive cells, given that ablation of *Gli1*-CreERT DTR cells diminishes HO. So a detailed analysis of *Gli1*CreERT2-DTA mice concerning the effects on progenitor and maturing cells of the osteochondral lineage after tendon injury could provide further insights and should be done accordingly.

Our response: We sincerely appreciate the reviewer's insightful observations regarding the cellular dynamics of *Gli1*⁺ populations. We fully concur that *GLI1*⁺/*SOX9*⁺ and *GLI1*⁺/*RUNX2*⁺ cells exhibit relatively lower prevalence compared to their *GLI1*⁻ counterparts, a finding consistent with our hypothesis that *Gli1*⁺ cells represent an early-stage osteochondrogenic subpopulation during initial injury responses, reflecting inherent heterogeneity within tendon progenitor pools.

We acknowledge a critical labeling error in our initial statistical annotations where the data for *GLI1*⁺/*COL2*⁺ versus *GLI1*⁻/*COL2*⁻ cells (and similarly for *OCN*) were inadvertently reversed in **old Fig. S3B and S3C**, which we have now corrected in the revised manuscript with enhanced image clarity to resolve the apparent contradiction (see new **Appendix Fig. S4A-D**); specifically, high-resolution imaging (see **following immunofluorescence images**) confirms that while focal regions (e.g., tendon sheath periphery in magnified insets) exhibit near-total co-localization of *Gli1*⁺ and *COL2*/*OCN*, broader heterotopic cartilage/ossification lesions demonstrate minimal *Gli1*⁺ contribution to *COL2*⁺/*OCN*⁺ areas, thereby aligning with the quantified rarity of *GLI1*⁺/*SOX9*⁺ and *GLI1*⁺/*RUNX2*⁺ populations. For clarity, we updated the *GLI1*-*COL2*/*OCN* co-staining images in the new **Appendix Fig. S4A-D**. We deeply apologize for this oversight and thank the reviewer for prompting a rigorous re-evaluation that strengthens our conclusions.

While we agree that *Gli1*⁺-derived *Fmod*⁺ lineages may undergo chondro-osteogenic transdifferentiation at injury sites, practical constraints preclude direct validation via *Fmod-CreER*^{T2} models, as confirmed through extensive consultations with the Jackson Laboratory and multiple transgenic rodent providers in China. To address this concern, we have implemented fluorescence-activated cell sorting (FACS) of *Gli1-CreER*^{T2}; Ai9 tendon cells at the steady state and critical post-injury intervals (7 and 21 dpi), enabling comparative single-cell transcriptomic profiling of: *Gli1*⁺ and *Gli1*⁻ cells.

UMAP-based dimensionality reduction and graph-based clustering identified 11 transcriptionally distinct subpopulations within *Gli1-CreER*^{T2+} cells. Cell type annotation identified: Tendon stem cells (clusters 3, 4 and 8; *Tppp3*^{high} / *Ly6a*^{high} / *Scx*^{low}), tendon progenitors (clusters 1, 2, 6 and 9; *Tppp3*^{mid} / *Ly6a*^{high} / *Scx*^{high}), Tenocytes (cluster 7; *Tppp3*^{low} / *Ly6a*^{low} / *Scx*^{high} / *Fmod*^{high}), osteochondral lineages (cluster 5: Fibrochondrocytes, *Tppp3*^{low} / *Sox9*⁺ / *Acan*⁺ / *Tnfaip6*^{high}; cluster 11, mature chondrocytes, *Tppp3*^{low} / *Col2a1*^{high} / *Acan*⁺ / *Tnfaip6*^{low}; cluster 10: osteogenic progenitors, *Tppp3*^{mid} / *Ly6a*^{high} / *Scx*^{high} / *Runx2*^{high}) (see Fig. EV2A, B).

Furthermore, RNA velocity analysis demonstrated a hierarchical differentiation trajectory from tendon stem cells (clusters 3, 4 and 8) → tendon progenitors (clusters 1, 2, 6 and 9) → tenocytes/osteochondral lineages (clusters 5, 10 and 11) (see Fig. EV2C). These dynamic transitions strongly support our central premise that *Gli1-CreER*^{T2+} progenitors adopt divergent differentiation fates during heterotopic

ossification. Notably, cluster 2 (tendon progenitors), cluster 5 (fibrochondrocytes) and cluster 11 (mature chondrocytes) exhibited elevated *Fmod* expression, while tendon stem cells (clusters 3, 4 and 8) showed low *Fmod*. RNA velocity analysis further showed that cluster 2, originating from tendon stem cells, could transition into cluster 5 and finally cluster 11, suggesting that *Gli1*⁺ tendon stem cells (*Fmod*^{low} clusters 3, 4 and 8) initially differentiate into *Fmod*⁺ tendon progenitors and finally chondrocytes (see **Fig. EV2**). No direct transdifferentiation from *Fmod*^{high} tenocytes to chondrocytes was observed.

Additionally, to examine the hypothesis that non-*Gli1* expressing progenitors contribute to progenitors only and that they diminish and are overruled by the smaller fraction of *Gli1*⁺/*Sox9*⁺ progenitors, we performed scRNA-seq for *Gli1*⁻ cells. This analysis revealed that *Gli1*⁻ cell comprised tendon stem cells and gave rise to tenocytes and osteochondral lineages (see **Appendix Fig. S3**), consistent with our immunostaining results.

Although *GLI1*⁻ osteochondral lineage cells numerically dominate their *GLI1*⁺ counterparts, our experimental evidence demonstrating that *Gli1*⁺ cell ablation reduces heterotopic ossification suggests a dual-pathway regulatory mechanism: 1) Direct depletion of osteochondrogenic *Gli1*⁺ progenitors, and 2) Secondary disruption of niche-dependent maintenance in *Gli1*⁻ subpopulations through intercellular crosstalk. To investigate this hierarchical relationship, we are currently performing immunofluorescence analysis of established tendon progenitor markers (TPPP3) and osteochondral markers (SOX9 and RUNX2) in both *Gli1-CreER*^{T2} and *Gli1-CreER*^{T2}; DTA models, with particular focus on spatial distribution patterns of TPPP3⁺ cells within injury-induced microenvironments. As expected, *Gli1*⁺ cell depletion resulted in a significant reduction of the total TPPP3⁺ tendon progenitor population (see **Fig. EV1C, D**). This was accompanied by corresponding decreases in SOX9⁺ chondrogenic progenitors, RUNX2⁺ osteochondrogenic progenitors, COL2A⁺ mature chondrocytes and OPN⁺ osteoblasts (see **Fig. EV1E-L**). These findings establish *Gli1*⁺ cells as crucial regulators of both cell-autonomous differentiation and non-autonomous progenitor maintenance, as elaborated in the revised Discussion.

2. Expression of *Acvr1* and its mutant conversion should be demonstrated.

Our response: We sincerely appreciate the reviewer's critical insight regarding ACVR1 validation. To comprehensively address mutant receptor dynamics, we performed immunofluorescence comparing ACVR1 expression patterns in injured tendons of *Gli1-CreER*^{T2}; *Ai9* versus *Gli1-CreER*^{T2}; *Acvr1*^{R206H/+}; *Ai9* mice at 5 dpi, with particular focus on spatial colocalization of ACVR1 within *Gli1*⁺ lineage cells. Critically, no significant difference in ACVR1 expression was observed between groups, excluding the hypothesis that elevated ACVR1 expression (rather than the *Acvr1*^{R206H} mutation) drives heterotopic ossification (HO) in tendons (see **Fig. EV3A, B**).

Acvr1^{R206H/+} mutation classically triggers p-SMAD1/5 pathway activation. To functionally validate pathway activation, we performed phospho-SMAD1/5 (p-SMAD1/5) immunofluorescence comparing signaling intensity in injury-induced HO niches versus physiological repair zones. Strikingly, the majority of *Gli1*⁺ cells in *Gli1-CreER*^{T2}; *Acvr1*^{R206H/+}; Ai9 tendons exhibited robust p-SMAD1/5 activation, whereas *Gli1*⁺ cells in control (*Gli1-CreER*^{T2}; Ai9) tendons showed negligible signaling (**Fig. EV3C, D**).

Complementing this, fluorescence-activated cell sorting (FACS) of *Gli1*⁺ (tdTomato) cells from injured tendons (5 dpi) of both *Gli1-CreER*^{T2}; Ai9 and *Gli1-CreER*^{T2}; *Acvr1*^{R206H/+}; Ai9 mice. Allele-specific Sanger sequencing of sorted cells confirmed successful disruption of the STOP cassette and expression of the mutant allele in *Gli1-CreER*^{T2}; *Acvr1*^{R206H/+} mice (**Fig. EV3E; Appendix Fig. S6**), enabling quantification of mutant allele burden. These results collectively confirm successful *Acvr1*^{R206H/+} mutant receptor expression and BMP pathway hyperactivation, further validating our FOP mouse model establishment.

3. The data of GNAS and p-PKA substrate analysis (Fig. 6) suggest that at later time points post-injury, other cells overexpress GNAS and p-PKA. Can the nature of these cells (osteoprogenitors, mature osteoblasts, chondroprogenitors, mature chondrocytes) be revealed? Are they affected by inhibitory treatment?

Our response: We appreciate the reviewer's insightful inquiry into the cellular identity and therapeutic responsiveness of GNAS/ p-PKA substrate-activated populations during late-stage injury. To address this, we integrated trajectory analysis of *Gli1*⁺ and *Gli1*⁻ cell transcriptomic dynamics (from our *Gli1-CreER*^{T2}; Ai9 scRNA-seq dataset) with immunofluorescence in 7 dpi specimens and co-staining for lineage-specific markers: RUNX2 (osteoprogenitors) and SOX9 (chondroprogenitors), alongside GNAS/p-PKA substrate detection. This spatiotemporal profiling revealed that *Gli1*⁻/ *Gnas*^{high} cells encompassed osteochondrogenic progenitors (cluster 1), tendon progenitors (cluster 3), prefibrochondrocytes (cluster 6), mature chondrocytes (cluster 14) and OPN⁺ mature osteoblasts (**see Appendix Fig. S8A-D**). Likewise, *GLI1*⁻/ p-PKA substrate⁺ cells included SOX9⁺ chondroprogenitors, RUNX2⁺ osteoprogenitors, ACAN⁺ mature chondrocytes and OPN⁺ mature osteoblasts (**see Appendix Fig. S8E-H**).

Moreover, these osteochondrogenic lineage cells were inhibited following NF449 treatment and *Prkaca* knock out in *Gli1*⁺ cells (**see Appendix Fig. S9A-H**).

4. The molecular mechanisms of GNAS inhibition that cause less HO formation remain elusive. For example, are the main targets CREB dependently regulated? CREB inhibitors could resolve this fact. The authors could try in addition to identify by RNA-Seq approaches, for example, further direct target genes, and eventually validate in an ex vivo approach that are implicated in the HO effects.

Our response: We sincerely thank the reviewer for highlighting this critical gap in

our mechanistic understanding. Building on our preliminary data demonstrating elevated phosphorylation of PKA substrates (including CREB) within heterotopic ossification (HO) lesions, we leveraged our scRNA-seq dataset of *Gli1*⁺ cells isolated from both normal and injured tendons of *Gli1-CreER*^{T2}; Ai9 mice to performed Regulon analysis. This analysis identified downstream targets of the GNAS/PKA pathway, such as *Creb3* (see **Fig. EV5A**).

Complementing these findings, functional validation via *in vivo* administration of the selective CREB inhibitor 666-15 revealed that CREB inhibition significantly attenuated HO formation (**Fig. EV5B-E**). Collectively, these results reinforce the GNAS/PKA signaling axis as a core molecular driver of aberrant osteochondral differentiation in *Gli1*⁺ cells.

Minor Points:

1. Official mouse Jax nomenclature should be used in the MM section.

Our Response: We will supplement the official mouse Jax nomenclature in the section of material and methods, as follows:

Gli1-CreER^{T2} (*Gli1*^{tm3(cre/ERT2)Alj}/J),

Prrx1-Cre (B6.Cg-Tg(*Prrx1-cre*)1Cjt/J),

Tek-Cre (B6.Cg-Tg(*Tek-cre*)12Flv/J),

Ai9 (B6.Cg-Gt(ROSA)26Sortm9(CAG-tdTomato)Hze/J),

DTA (B6.129P2-Gt(ROSA)26Sortm1(DTA)Lky/J)

ZsGreen (B6.Cg-Gt(ROSA)26Sortm6(CAG-ZsGreen1)Hze/J)

2. Line 113: How is the abundance of *Gli1*⁺ cells in the sheaths of the tendons after injury?

Our Response: We have rewritten this sentence and highlight the abundance of *Gli1*⁺ sheath cells is increased (see **line 113, page 5**).

3. Line 132: The term Cre-dependent *Gli* KO mice is misleading, the authors mean mice with the elimination of *Gli1* positive progenitor cells.

Our Response: We apologized for this mistake; we have corrected this sentence into “Cre-dependent *Gli1*⁺ cell depletion” (see **line 132, page 5**).

4. Fig. 2C: It would be helpful for interpretation also to show the expression of *Gli1* over the different clusters.

Our Response: We have supplemented the violin plot image of *Gli1* in the new **Fig. 2C**.

5. The Figures 3i, m are difficult to read. These legends should be separated from the y-axis.

Our Response: We have adjusted the height of the statistical graphs to ensure clear visibility and legibility of the figure legends.

6. Figures 7r, t are difficult to read. X-axis description is somewhat untidy.

Our Response: We thank the reviewer for this valuable comment. We have adjusted the vertical scale of the statistical graphs in **Fig. 7R, T** to ensure clear visibility of the X-axis labels. Additionally, specific injury and collection time points have been added to the X-axis descriptors to enhance experimental transparency (see legend of **Fig. 7R, T**).

Again, we thank you for your suggestion and assistance for publishing this study.

Sincerely

Chen Kan, Ph.D.
Department of Pathophysiology
School of Basic Sciences
Anhui Medical University
81 Meishan Road
Hefei, Anhui 230032

Email: chenkan@ahmu.edu.cn

Dear Dr Kan,

Thank you for submitting your revised manuscript (EMBOJ-2024-119973R1) to The EMBO Journal, as well for your patience with our feedback. Your amended study was sent back to the referees for their scientific reassessment, and we have received re-reports from both of them, which I enclose below. As you will see, the referees state that the work has been substantially enhanced by the revisions and they are now broadly in favour of publication.

Thus, we are pleased to inform you that your manuscript has been accepted in principle for publication in The EMBO Journal.

We now need you to take care of a number of issues related to formatting and data presentation as detailed below, which should be addressed at re-submission.

Please contact me at any time if you have additional questions related to below points.

As you might have seen on our web page, every paper at the EMBO Journal now includes a 'Synopsis', displayed on the html and freely accessible to all readers. The synopsis includes a 'model' figure as well as 2-5 one-short-sentence bullet points that summarize the article. I would appreciate if you could provide this figure and the bullet points.

Thank you for giving us the chance to consider your manuscript for The EMBO Journal. I look forward to your final revision.

Again, please contact me at any time if you need any help or have further questions.

Best regards,

Daniel Klimmeck

>> Please add up to five keywords to your study.

>> Adjust the title of the 'Declaration of Interests' section to 'Disclosure and Competing Interests Statement' and move after Acknowledgements.

>> Section order should be corrected as follows: title page with complete author information, abstract, keywords, introduction, results, discussion, methods, data availability section, acknowledgements, disclosure and competing interests statement, references, main figure legends, tables, expanded figure legends.

>> References: adjust reference format to EMBO Journal format, 10 authors et al, and place References after the Discussion, before figure legends.

>> Please cite the earlier study (Kan et al, 2024; PMID: 39308190) in the Methods section.

>> Funding: please enter the following funding information in the list of funders in our online system: " the Postdoctoral Fellowship Program of CPSF under Grant Number GZC20240010, the China Postdoctoral Science Foundation - Anhui Joint Support Program under Grant Number 2024T026AH, the Young Science and Technology Talent Lifting Program of Anhui

Provincial Association for Science and Technology (RCTJ202428), Anhui Provincial Postdoctoral Research Program Funding (Grant NO. 2025A1038), the Basic and Clinical Cooperative Research Incubation Project of Anhui Medical University for The Third Affiliated Hospital (2023sfy016), the Key Project of Ningbo Municipal Natural Science Foundation (2024J031), Anhui Provincial Key Research Program in Natural Sciences for Higher Education Institutions (2023AH053407) and the Natural Science Foundation of Anhui Province (2408085MH197).".

>> Reagents and Tools table for the Methods section: please as a separate .doc file using the existing template in the Guide For Authors, listing key reagents, experimental models, software and relevant equipment.

>> Please provide a completed source data checklist for the study.

>> Data availability section: please update the dataset IDs and html links for the GEO datasets. Make sure, datasets are made publicly accessible.

>>Human samples: include a statement in the Methods section that informed consent was obtained from all subjects and that the experiments conformed to the principles set out in the WMA Declaration of Helsinki and the Department of Health and Human Services Belmont Report.

>> Author Checklist: update the Ethics section on 'authority granting ethics approval' and 'informed consent statement' information.

>> Consider additional changes and comments from our production team as indicated below:

DATA CHECK:

Please note that the specific URLs for 126060 and 298748, S-BSST2091 datasets are not provided in the data availability statement.

- Data citations: no comments
- Figure Legends (main + EV): 1. Please note that information related to n is missing in the legend of figure 2D

Referee #1:

The authors have successfully addressed all of my concerns. The manuscript is now ready to be accepted for publication.

Referee #2:

The authors addressed and responded to my concerns satisfactorily. There might be still some few typos in the text that should be eliminated. I appreciate that the data are presented much more clearly.

The authors addressed all minor remaining editorial requests.

Dear Dr Kan,

Thank you for submitting the revised version of your manuscript. I have now evaluated your amended manuscript and concluded that the remaining minor concerns have been sufficiently addressed.

I am thus pleased to inform you that your manuscript has been accepted for publication in the EMBO Journal.

On a different note, I would like to alert you that EMBO Press offers a format for a video-synopsis of work published with us, which essentially is a short, author-generated film explaining the core findings in hand drawings, and, as we believe, can be very useful to increase visibility of the work. Please see the following link for representative examples and their integration into the article web page:

<https://www.embopress.org/doi/full/10.15252/emj.2019103932>

Best regards,

Daniel Klimmeck

Daniel Klimmeck, PhD
Senior Editor
The EMBO Journal
EMBO
Postfach 1022-40
Meyerhofstrasse 1
D-69117 Heidelberg
contact@embojournal.org

>>> Please note that it is The EMBO Journal policy for the transcript of the editorial process (containing referee reports and your

response letter) to be published as an online supplement to each paper. If you do NOT want this, you will need to inform the Editorial Office via email immediately. More information is available here: https://www.embopress.org/transparent-process#Review_Process